# Maternal pluripotency factors initiate extensive chromatin remodelling to predefine first response to inductive signals

George E. Gentsch [1], Thomas Spruce[2], Nick D.L. Owens[3] & James C. Smith [1]

Embryonic development yields many different cell types in response to just a few families of inductive signals. The property of signal-receiving cells that determines how they respond to inductive signals is known as competence, and it differs in different cell types. Here, we explore the ways in which maternal factors modify chromatin to specify initial competence in the frog *Xenopus tropicalis*. We identify early-engaged regulatory DNA sequences, and infer from them critical activators of the zygotic genome. Of these, we show that the pioneering activity of the maternal pluripotency factors Pou5f3 and Sox3 determines competence for germ layer formation by extensively remodelling compacted chromatin before the onset of inductive signalling. This remodelling includes the opening and marking of thousands of regulatory elements, extensive chromatin looping, and the co-recruitment of signal-mediating transcription factors. Our work identifies significant developmental principles that inform our understanding of how pluripotent stem cells interpret inductive signals.

[1] Developmental Biology Laboratory, The Francis Crick Institute, London NW1 1AT, UK. [2] Centre for Genomic Regulation, Barcelona Institute for Science and Technology, 08003 Barcelona, Spain. [3] Department of Stem Cell and Developmental Biology, Pasteur Institute, 75015 Paris, France. Correspondence and requests for materials should be addressed to G.E.G. (email: george.gentsch@crick.ac.uk) or to J.C.S. (email: jim.smith@crick.ac.uk)

The specification of different cell types during embryonic development is achieved through the repeated use of a small number of highly conserved intercellular signals. The property of a cell that defines the way it responds to such signals (if it responds at all) is known as its competence[1]. Classical experiments with amphibian embryos show that competence is regulated in both space and time. For example, head ectoderm of the tailbud embryo responds to the underlying optic vesicle by forming a lens, while other surface ectoderm is unable to do so[2]. Similarly, on a temporal scale, naïve pluripotent (animal cap) tissue of the early *Xenopus* embryo loses the ability to form muscle in response to Nodal signalling by the mid-gastrula stage[3].

How is competence regulated at the molecular level? A simple mechanism would involve the acquisition or loss of components of a signal transduction pathway, but this fails to explain differences in context-dependent responses to the same signals. In addition, although there are some tissue-specific signal receptor isoforms, transduction to the nucleus is limited to just a few signal mediators with transcriptional activity such as β-catenin (canonical Wnt signalling), Smad2 (Nodal signalling) and Smad1 (BMP signalling). Thus, a more plausible and more frequent way in which tissue-specific competence might arise is through the recruitment of these signal mediators to different gene regulatory sites, which in turn would drive the specification of different cell types.

Cell lineage determinants such as sequence-specific transcription factors (TFs) may play a role in determining competence. For lens induction, the responding cells of the head ectoderm depend on the presence of the eye-regulating TF Pax6[4]. For neural patterning, the appropriate transcriptional response to Sonic hedgehog (Shh) signalling requires the input of the pan-neural TF Sox2[5]. However, although we know that certain TFs can act as competence factors, these TFs have not been identified in a systematic fashion, and their modes of action remain largely unknown. Our understanding of how chromatin interprets inductive signals is especially important because the generation of therapeutically relevant cells like insulin-producing pancreatic β-cells frequently relies on the deployment of signal modulators at different stages of cell differentiation in vitro[6]. Moreover, the initiation and progression of tumours is often associated with a change in their competence which, in some instances, is correlated with the erroneous re-activation of embryonic TFs stimulating excessive cell proliferation[7].

In an effort to analyse systematically the molecular basis of competence, we chose, for the following reasons, to study the first inductive signalling events in the embryo of *Xenopus tropicalis*. First, like most multicellular organisms, *X. tropicalis* begins development with a transcriptionally silent genome (Fig. 1a). After fertilisation, twelve rapid cleavages convert the egg into a mid-blastula embryo with little transcription and little diversification among its 4096 cells[8]. Both the low levels of transcription and the cellular homogeneity assist in the interpretation of whole-embryo and loss-of-function chromatin profiling. Second, TFs involved in zygotic (or embryonic) genome activation (ZGA) are likely to occupy top-level positions within gene regulatory networks, and their binding should thus correlate strongly with gene target expression[9]. This makes them more amenable to study. Third, based on the nuclear accumulation of signal mediators[10] (Fig. 1a), we know a great deal about where and when inductive signalling occurs during ZGA. And finally, in vitro fertilisation yields thousands of synchronously developing embryos, which greatly facilitates temporal chromatin profiling.

Taking advantage of these features of *X. tropicalis*, we have used transcriptional, translational and multi-level chromatin profiling to help identify the earliest regulatory DNA sequences (hereafter called putative *cis*-regulatory modules, pCRMs). From these, we infer the critical maternal activators of the zygotic genome as potential competence factors. We next compare the DNA binding of several maternal/zygotic TFs and signal mediators across the mid-blastula transition (MBT) to reveal the effect of co-expressed TFs on their recruitment and the recruitment of other transcriptional regulators to chromatin in vivo. Finally, we demonstrate how maternal TFs of the pluripotency network structure the chromatin landscape, which in turn determines the signal-mediated regionalisation of gene activity and the specification of the three germ layers: ectoderm, mesoderm and endoderm.

## Results

**pCRM motifs match affinities of frequently translated TFs.** In an effort to understand how early chromatin dynamics influence the recruitment of signal mediators to the genome (Fig. 1b), we first identified ~27,000 pCRMs from the 32-cell to the late gastrula stage by mapping focal RNA polymerase II (RNAPII) depositions on a genome-wide scale by means of ChIP-Seq (Fig. 1c and Supplementary Data 1 and 2). RNAPII has no DNA sequence preference and its chromatin engagement is a reliable and objective indicator of pCRM usage[11]. The number of RNAPII-engaged (RNAPII⁺) pCRMs increased from ~650 at the 32-cell stage to >10,000 at the 1024-cell and later developmental stages (Supplementary Fig. 1a). The largest changes to RNAPII⁺ pCRMs, as calculated by pairwise Spearman correlations, were detected between the 1024-cell stage and the MBT (Fig. 1d), with most pCRMs being engaged only transiently before the MBT (e.g., 6145 from the 128-cell to the 1024-cell stage) and more persistently after the MBT (Supplementary Fig. 1a). The analysis of enriched DNA motifs among RNAPII⁺ pCRMs suggests that pre-MBT recruitment is predominantly directed by members of the FOXH, POU, SOX and T domain TF families (Fig. 1e and Supplementary Fig. 1b).

The discovery of RNAPII⁺ pCRMs is difficult in promoters and gene bodies where extended RNAPII depositions associated with transcript elongation might hamper the correct detection of RNAPII⁺ pCRMs. MBT-staged pCRMs were therefore further characterised for chromatin accessibility and for the enhancer-associated histone mark H3K4me1. In our hands, the high yolk content in early *X. tropicalis* embryos made it impossible to use transposition[12] to probe chromatin accessibility. Instead, we used an approach involving DNase I mediated digestion followed by deep sequencing (DNase-Seq), in which we selected small accessible fragments of DNA (see Methods and Supplementary Fig. 1c for exemplar comparison with other chromatin features). We detected ~17,500 accessible (DNase hypersensitive) pCRMs, ~85% of which showed both RNAPII and flanking H3K4me1 above background (Fig. 1f, Supplementary Fig. 1d and Supplementary Data 1 and 3). About 29% and 31% of accessible pCRMs (compared with ~16% and ~38% of pCRMs detected by RNAPII peak calling) were found in promoters and gene bodies, respectively (Supplementary Fig. 1e).

Enriched DNA motifs among accessible/RNAPII⁺/H3K4me⁺ pCRMs were then correlated with maternally inherited and translated sequence-specific factors identified by egg-staged mass spectrometry[13] and pre-MBT ribosome footprinting[14] to identify members of the TF families that may play a role in the ZGA (Fig. 2a, Supplementary Fig. 2a and Supplementary Data 4). It proved that the binding preferences of the most frequently translated maternal TFs and signal mediators matched the most significantly enriched DNA motifs (Fig. 2b). These were POU-SOX (Pou5f3-Sox3 heterodimer), Krüppel-like zinc finger (ZF; Sp1 and several Klf), POU (Pou5f3), SOX (Sox3), bZIP (Max), FOXH (Foxh1), ETS (Ets2), NFY (NFYa/b/c), SMAD (Smad1/2), T (mVegT, a maternal VegT isoform), TCF (Tcf/β-catenin), basic helix-span-helix domain (bHSH; Tfap2), and OTX (Otx1).

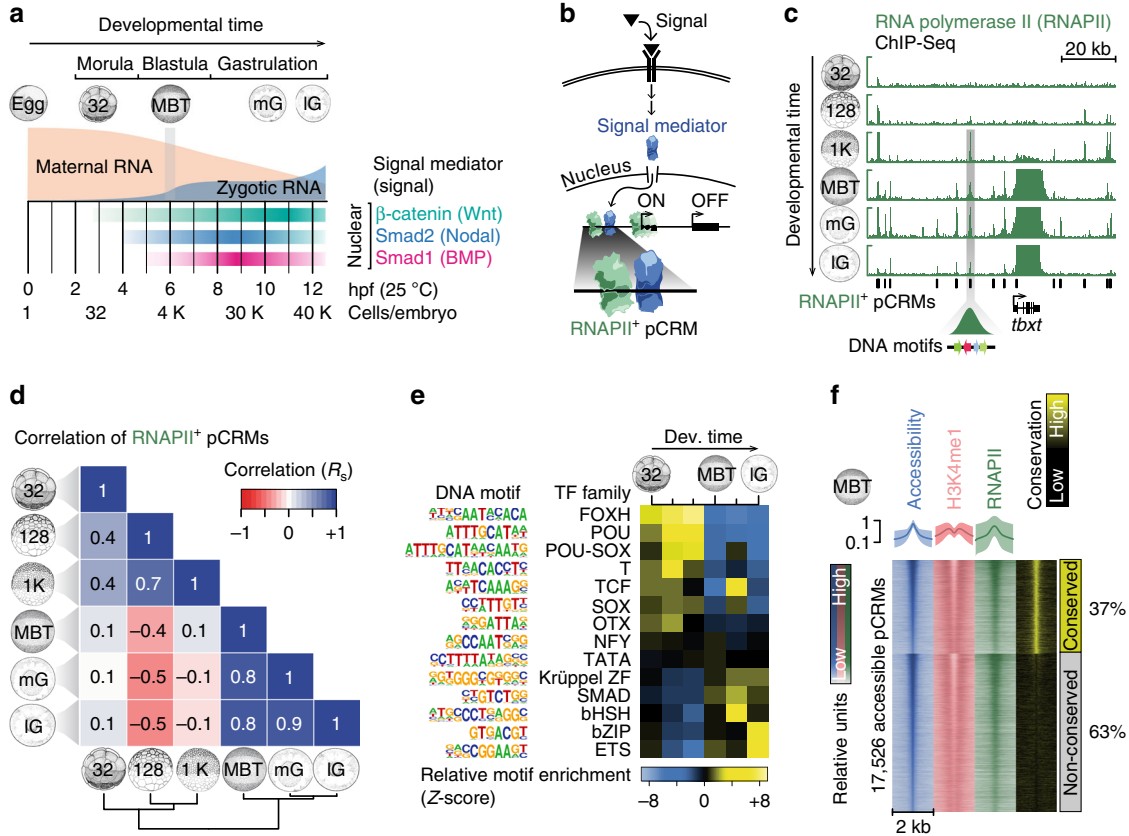

**Fig. 1** Characterisation of pCRMs instructing ZGA. **a** Timeline (hpf, hours post-fertilisation, at 25 °C) of the maternal-to-zygotic transition and earliest signalling events (nuclear accumulation of Wnt, Nodal and Bmp signal mediators β-catenin, Smad2 and Smad1, respectively) during early *X. tropicalis* development up to the late gastrula embryo (12 hpf) with ~40,000 (40 K) cells. **b** Signal transduction pathway causing signal mediators to enter the nucleus and engage with pCRMs (e.g., marked by RNAPII). **c** Snapshot of RNAPII recruitment to pCRMs of the zygotic gene locus *tbxt* from the 32-cell to the late gastrula stage. The underlying DNA sequence of RNAPII+ pCRMs are used to discover enriched DNA motifs de novo (illustrated as coloured arrows for one RNAPII+ pCRM). **d** Spearman correlations ($R_s$) of RNAPII binding levels across ~27,000 pCRMs (Supplementary Data 2) between the indicated developmental stages. **e** Temporal enrichment (Z-score) of consensus DNA motifs known to be recognised by indicated TF families among RNAPII+ pCRMs. **f** MBT-staged heat map of DNase-probed chromatin accessibility, RNAPII binding and H3K4me1 marking (n = 2 biologically independent samples) across ~17,500 pCRMs (Supplementary Data 3) grouped by sequence conservation levels (phastCons) and sorted by the statistical significance of pCRM accessibility. Abbreviations used for the developmental timeline: 32, 128 and 1 K, 32-, 128- and 1024-cell stage; MBT, mid-blastula transition; eG, mG and lG, early, mid- and late gastrula stage

Interestingly, the enrichment level of these motifs was independent of pCRM sequence conservation (based on phastCons scoring) detected among vertebrates (Figs. 1f and 2b).

Of the most frequently translated TFs with cognate pCRM-enriched DNA recognition motifs, we selected Pou5f3, Sox3 and mVegT as potential competence factors of canonical Wnt, Nodal and BMP signalling. Sox3 and presumably also Pou5f3 (based on the spatial distribution of its maternal transcripts *Pou5f3.2* and *Pou5f3.3*[15]) are detected ubiquitously (Fig. 2c), while mVegT is restricted to the vegetal hemisphere[16]. The zygotic isoform of VegT (zVegT) is expressed within the marginal zone (Fig. 2d and Supplementary Fig. 2e). With respect to signal mediators, nuclear β-catenin begins to accumulate weakly on the dorsal side of the embryo (first detected at the 32-cell stage[17]) before spreading more prominently around the upper lip of the forming blastopore after the MBT[10] (Fig. 2d). Signal-induced nuclear accumulation of Smad1 and Smad2 is first detected around the MBT, with Smad1 being preferentially ventral and Smad2 being vegetal and within the marginal zone[10,18] (Fig. 2d and Supplementary Fig. 2h).

**pCRM motifs reflect patterns of chromatin engagement.** Informed by the time of onset of their nuclear accumulation

(Figs. 1a and 2d), we generated genome-wide chromatin profiles across the MBT (Fig. 3a and Supplementary Data 1) to ask whether the recruitment of β-catenin, Smad1 and Smad2 might be influenced by co-expressed maternal TFs like Sox3, mVegT and Foxh1[19] or vice versa (see Fig. 2c and Supplementary Fig. 2b–n for ChIP antibody verification). As an outgroup control to these maternal proteins, we selected the binding profiles of the zygotic TFs Eomes (Eomesodermin)[20], zVegT[20], Tbxt (Brachyury)[20] and Tbx6, all of which contain the T-box DNA binding domain and collectively regulate the neuro-mesodermal cell lineage during and beyond gastrulation[21]. The profiled developmental stages and chromatin factors are colour-coded as illustrated in Fig. 3a and ChIP-Seq peak call coordinates are listed in Supplementary Data 5.

Maternal and zygotic TFs and signal mediators shared many chromatin characteristics: (1) DNA occupancy levels followed a log-normal distribution with <1000 RNAPII-transcribed genes receiving high and super-enhancer-like[22] input (i.e., clusters of occupied pCRMs separated by ≤25 kb). Similar distributions were also observed for chromatin accessibility and RNAPII engagement (see examples, in Fig. 3a and Supplementary Fig. 3a, b and systematic analysis in Supplementary Fig. 4a). (2) From the 1024-cell to the late gastrula stage, TF-bound

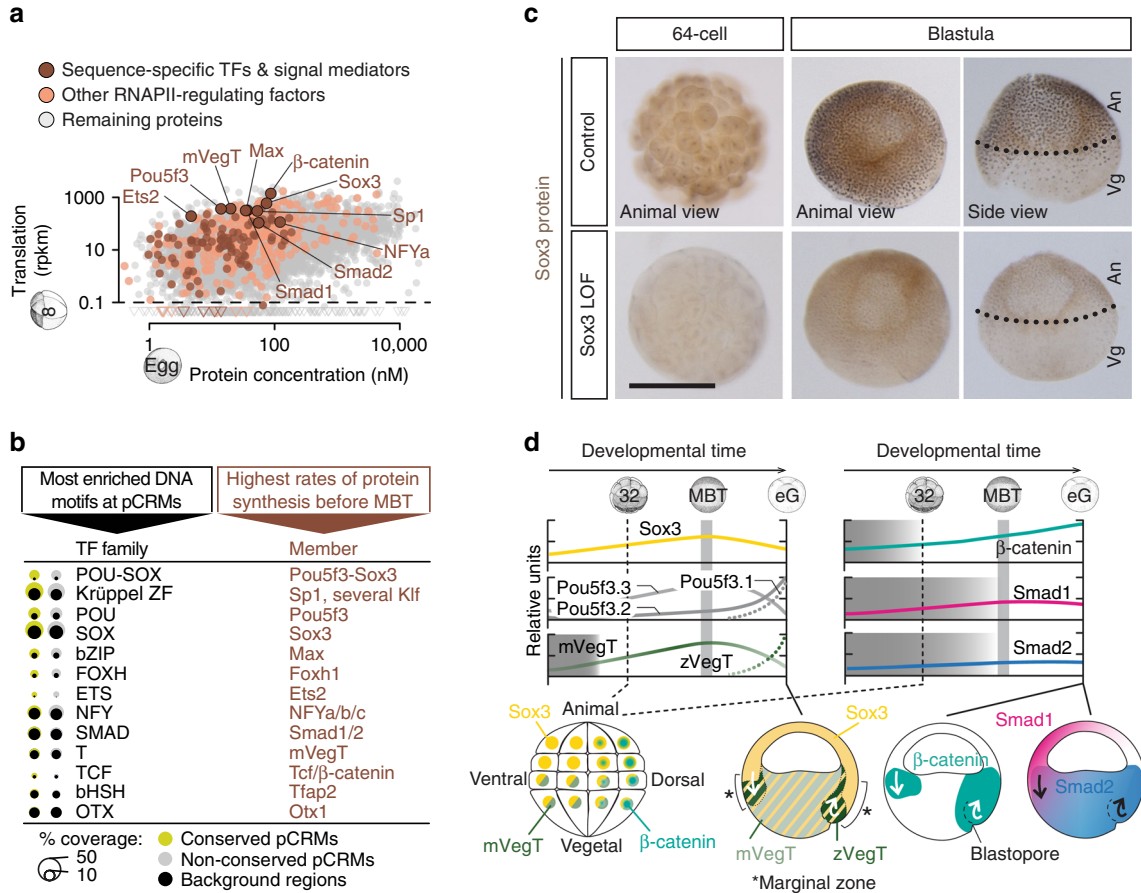

**Fig. 2** Search for ZGA-critical proteins based on their significantly enriched DNA recognition motifs at accessible and engaged (RNAPII[+]/H3K4me1[+]) pCRMs and their high translation frequency before the MBT. **a** Maternal protein concentrations in the egg[13] versus ribosome footprint (translation) levels at the 8-cell stage[14]. Most frequently translated representatives of various TF families are labelled. **b** Matching canonical pCRM-enriched DNA motifs (sorted by statistical significance) with frequently translated TFs and signal mediators. **c** WMIHC of Sox3 protein in control and Sox3 loss-of-function (LOF) embryos at the 64-cell and blastula stage. Nuclear accumulation of Sox3 protein was detected in both the animal (An) and vegetal (Vg) hemisphere of control embryos. Scale bar, 0.5 mm. **d** Graphical illustration of protein levels (derived from mass spectrometry data[42]) and nuclear localisations (mainly derived from WMIHC, see references below) of selected TFs and signal mediators based on our and previously published results: Sox3 (this study and ref. [50]), mPouV (Pou5f3.2 and Pou5f3.3) and (zygotic) Pou5f3.1 (deduced from transcript data[15]), mVegT and zVegT[16], β-catenin[10,17], Smad1 (this study and refs. [10,18]) and Smad2 (this study and refs. [10,18]). Shaded boxes indicate periods of non-nuclear protein localisation. Arrows indicate tissue movements of gastrulation. Abbreviations used for the developmental timeline: 8 and 32, 8-cell and 32-cell stage; MBT, mid-blastula transition; and eG, early gastrula stage

super-enhancers (Supplementary Data 6) were linked through the gene ontologies of nearby zygotic genes (≤5 kb) to early embryonic processes including germ layer and body axis formation (Fig. 3b). (3) The binding sites of TFs and signal mediators were also frequently defined by RNAPII deposition (Fig. 3c). (4) Among all zygotic genes, promoter-proximal regions were more consistently bound than other pCRMs (Supplementary Fig. 4b).

Differences in DNA occupancy levels at different developmental stages, compared by pairwise Spearman correlations and principal component (PC) analysis, suggest that the recruitment to chromatin of signal mediators and TFs is driven both by their individual properties and by the developmental stage (Fig. 3d and Supplementary Fig. 5a). For example, chromatin recruitment of mVegT at stages of pluripotency resembled that of other sequence-specific factors at the same developmental stage but differed from the binding of its zygotic isoform and of related T-box TFs at later developmental stages (highlighted in Supplementary Fig. 5a). The importance of developmental stage in driving patterns of chromatin recruitment was further revealed by the changing DNA binding patterns of sequence-nonspecific RNAPII to pCRMs (Fig. 3e).

The identification of enriched DNA recognition motifs at occupied pCRMs confirmed known TF/signal mediator properties, such as the sequence-specificity of their DNA binding domains, oligomerisation tendencies and protein–protein interactions (Supplementary Fig. 5b). For example, in contrast to other T-box TFs, Tbxt recognises palindromic T motifs due to its propensity to form homodimers[23]; and Smad2 chromatin recruitment is frequently associated with the FOXH motif because Smad2 interacts with Foxh1[24] (Supplementary Fig. 5b). Other DNA motifs such as the POU or POU-SOX motifs were consistently co-enriched from the 1024-cell to the late gastrula stage in most binding profiles. This is indicative of the pluripotent state, a developmental context associated with co-expression of Pou5f (Oct4) and SoxB1 (e.g., Sox2 or Sox3) proteins as previously observed in vitro[25] (Fig. 3g and Supplementary Fig. 5b).

The hierarchical clustering of DNA occupancy levels for the selected factors at different developmental stages revealed specific DNA motif combinations. For example, pCRMs showing 'unique' binding of mVegT (that is, binding that is not shared with the other profiled factors) show a high frequency of T, OTX and SOX motifs (Supplementary Fig. 6).

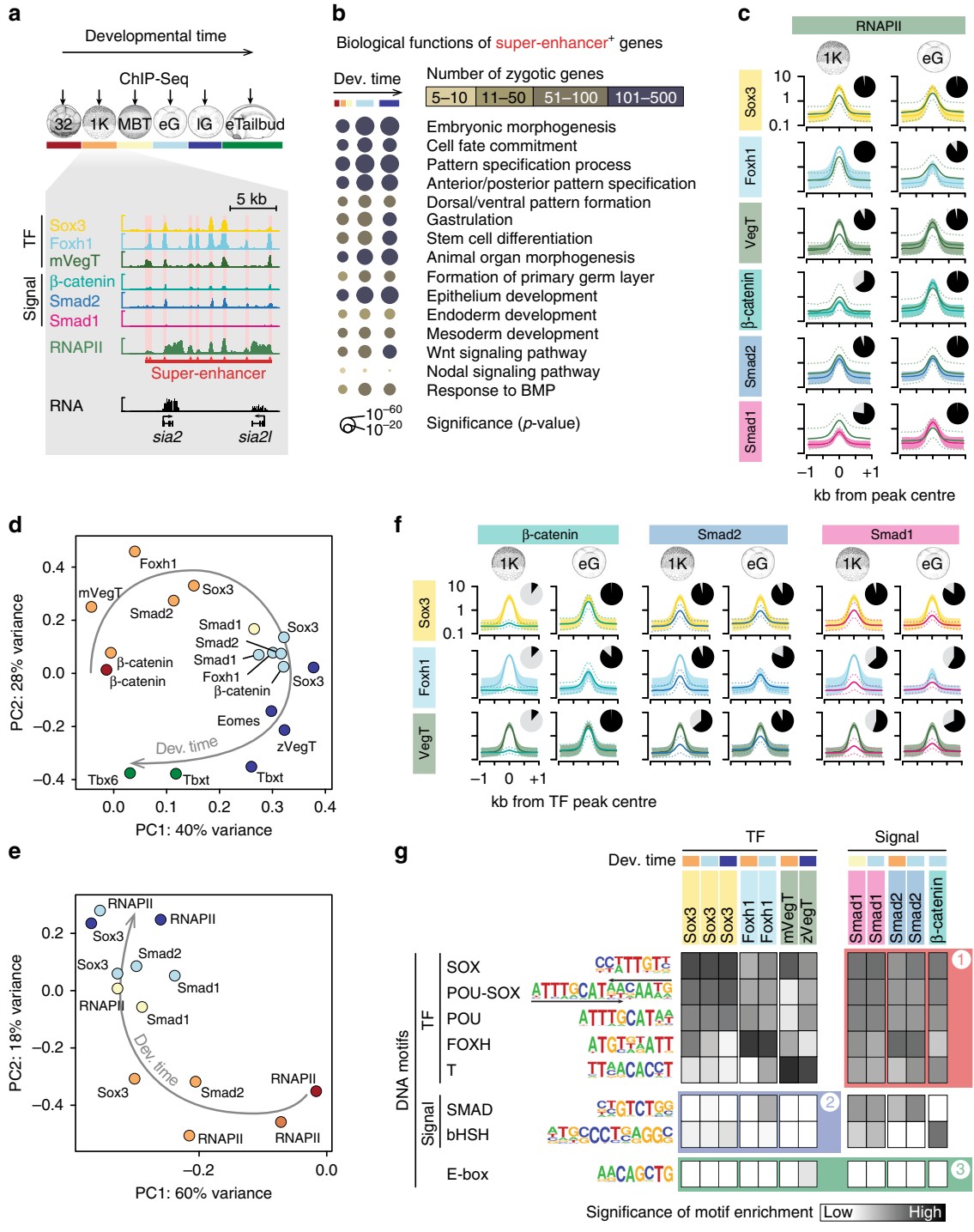

With respect to the chromatin recruitment of TFs versus signal mediators, we note that, first, Smads and/or β-catenin were frequently detected at Sox3, Foxh1 or VegT binding sites (top 2000 peaks shown in Fig. 3f) and, second, Smad- and/or β-catenin-bound pCRMs (top 2000 peaks) were significantly enriched for SOX, POU-SOX, POU, FOXH and T motifs (red field #1 in Fig. 3g) suggesting that corresponding TFs affect the recruitment of these signal mediators. The reverse was much less the case as shown by the low significance of SMAD and bHSH motif enrichments at TF-bound pCRMs (with the exception of Smad2-interactor Foxh1[24]) (blue field #2 in Fig. 3g).

**TF co-expression drives chromatin recruitment dynamics**. To explore the suggested importance of TF co-expression on pCRM engagement, we ectopically expressed an HA-tagged version of the muscle determinant MyoD (MyoD-HA) in animal cap cells and profiled pCRMs for MyoD-HA as well as for endogenous Sox3 and RNAPII at the early gastrula stage (Fig. 4a and Supplementary Data 1 and 7). MyoD was chosen because its canonical E-box recognition motif is normally not significantly enriched before or during gastrulation (green field #3 in Fig. 3g and Supplementary Fig. 5b) so its effect on chromatin engagement ought to be clearly discernible, while Sox3 and RNAPII were selected because they are ubiquitously expressed and

**Fig. 3** Chromatin engagement of TFs and signal mediators during the maternal-to-zygotic transition. **a** Chromatin profiling (ChIP-Seq) of selected TFs and signal mediators from the 32- or 1024-cell stage (32, 1 K) to the early (eG) and late (lG) gastrula and the early tailbud (eTailbud) stage. In all subsequent figure panels, the chromatin factors and developmental stages profiled are consistently colour-coded as illustrated here. The excerpt of multiple chromatin tracks shows the binding of maternal TFs (Sox3, Foxh1 and VegT) and signal mediators ($\beta$-catenin, Smad2 and Smad1) to the *siamois2* (*sia2* and *sia2l*) super-enhancer at the 1024-cell stage (see Supplementary Fig. 3a for the temporal progression of chromatin engagement to the *siamois2* and *ventx* super-enhancers). **b** Bubble plot shows significantly enriched biological processes associated with zygotic super-enhancer+ genes (i.e., genes possessing engaged super-enhancers ≤5 kb from their active TSS at indicated developmental stages). **c**, **f** Meta-plots (mean [solid line] ± s.d. [dotted line or polygon]) summarise the level of RNAPII (**c**) or signal mediator (**f**) engagement across 2000 pCRMs most frequently occupied by the indicated TFs or signal mediators at the 1024-cell and early gastrula stage, respectively. The pie chart next to each meta-plot shows the percentage of these TF+ or signal mediator+ pCRMs bound (ChIP ≥2x input tag density) by RNAPII or signal mediators, respectively. See Supplementary Note 1 for percentage numbers. **d** Biplot of principal component (PC) 1 (accounting for 40% variance) and 2 (28% variance) shows the relationship of TF and signal mediator binding levels across ~12,500 highly engaged pCRMs (compiled from the 2000 pCRMs with the highest DNA occupancy levels detected per protein and developmental stage) over several developmental stages. Note that developmental time (arrow) separates these profiles best. **e** Biplot of PC1 (60% variance) and PC2 (18% variance) for the temporal progression of RNAPII and TF binding levels across the same set of pCRMs as in **d**. **g** Heat map shows the statistical significance (hypergeometric p-value) of finding TF- and signal-specific DNA consensus motifs (*y*-axis) across 2000 pCRMs most frequently occupied by the indicated TFs or signal mediators (*x*-axis)

represent sequence-specific and nonspecific DNA binding factors, respectively. The ectopic expression elevated the Spearman correlations of Sox3 and RNAPII with MyoD-HA (Supplementary Fig. 7a) and shifted the first and second PC of Sox3 and RNAPII toward MyoD-HA (Fig. 4b) suggesting that MyoD-HA altered the binding of both Sox3 and RNAPII. Differential binding analysis on a genome-wide (heat map in Fig. 4c) and locus-specific (pileup track in Fig. 4d) scale confirms that MyoD-HA co-recruits endogenous Sox3 and RNAPII to its gene targets like *actc1* and *myl1*. The E-box motif of MyoD-HA emerged as a significantly enriched motif of Sox3 and RNAPII binding, while MyoD-HA binding itself seemed to be influenced by endogenous TFs as judged by the developmental period (from the 1024-cell to the late gastrula stage) characteristic enrichment of SOX, POU-SOX and FOX motifs at MyoD+ pCRMs (Fig. 4e). However, this opportunistic recruitment to non-canonical binding sites, such as MyoD-HA to functional Sox3 gene targets (e.g., *otx2* and *sox2* activated by Sox3-HA), did not affect transcription in animal caps (Fig. 4f–h).

The influence of TF co-expression on DNA occupancy was further substantiated by profiling chromatin for Sox3 in different anterior–posterior regions of the central nervous system (CNS) (Fig. 5a and Supplementary Data 1 and 8). The analysis of enriched DNA motifs suggested that Sox3 binding was affected by differentially expressed homeodomain TFs, such as orthodenticle homeobox (OTX) in the brain (head) and caudal homeobox (CDX) in the spinal cord (trunk, bud) (Fig. 5c and Supplementary Fig. 7c, d; see Fig. 5d for graphical illustration of TF co-expression at the posterior end of the embryo). This was particularly apparent in Sox3 binding to the colinear *HoxD* cluster defining anterior–posterior cell identity (Fig. 5b). Similar influences on chromatin engagement were observed within anterior and posterior mesoderm marked by Eomes in gastrula embryos and Tbxt/Tbx6 in gastrula and early tailbud embryos, respectively (Fig. 5c, d and Supplementary Fig. 7b, c). On a temporal scale, the influence of FOXH and POU motif-recognising TFs in recruiting Sox3 and T-box TFs was more pronounced in early than in late development (Fig. 5c and Supplementary Figs. 5b and 7c).

**Signal-induced regionalisation of ZGA relies on maternal TFs.** To ask whether signal-mediated ZGA requires maternal TFs, as suggested by the observed chromatin dynamics (Fig. 3), we next compared the effects of loss of maternal Sox3/PouV (Pou5f3.2 and Pou5f3.3; mPouV) or VegT (mVegT), and canonical Wnt, Nodal or BMP signal transduction, on zygotic transcription from the MBT to the late blastula and early gastrula stages (Fig. 6c and Supplementary Data 1). The high translation frequencies (Fig. 2a

and Supplementary Fig. 2a) coupled with fairly steady or even transient (mVegT and Pou5f3.3) protein levels around the MBT (Fig. 2d), suggest that maternal TFs and signal mediators have short half-lives. Consistent with this idea, the injection of antisense morpholino oligonucleotides (MOs) blocking the translation of maternal transcripts, such as *Sox3* or *mVegT* was effective in reducing protein levels (Fig. 2c and Supplementary Fig. 2c, e, f). We used previously-validated MOs to knock down mPouV[26] and $\beta$-catenin[27]. Although MOs can cause off-target mis-splicing and the induction of an immune response, the effects on splicing do not affect mature maternal transcripts and the immune response is only detectable beyond gastrula stages[28]. The nuclear accumulation of Smad1 and Smad2 in response to BMP and Nodal signalling was inhibited with the small molecules LDN193189[29,30] and SB431542[31,32], respectively. The morphological defects of these loss-of-function (LOF) treatments ranged from undetectable (Sox3), to weak (mVegT, BMP), to moderate (mPouV), to severe (mPouV/Sox3, Nodal, $\beta$-catenin) (Fig. 6a and Supplementary Fig. 8a, c). Moderate and severe defects affected gastrulation, while weak ones were only obvious later. All phenotypes were either in line with previous publications ($\beta$-catenin[27], Nodal[33] and BMP[34]) or could be rescued at the morphological (mPouV/Sox3; Fig. 6b) or transcriptional level (mVegT; Supplementary Fig. 8b) by co-injecting cognate mRNA. Interestingly, the role of maternal Sox3 could only be detected by knocking it down together with mPouV. In contrast to single Sox3 or mPouV LOF, double LOF embryos failed to close the blastopore (Fig. 6a and Supplementary Movie 1).

Transcriptome analysis of TF and signal LOF embryos was confined to the 3687 zygotic genes for which ≥two-fold reductions in exonic and/or intronic transcript level (see Online Methods) could be detected following injection of the RNAPII inhibitor $\alpha$-amanitin (Supplementary Data 9). About 83% of these genes had a maternal contribution of ≥1 per 10 million transcripts as detected within first hour post-fertilisation when the genome is quiescent (Fig. 6e). $\alpha$-Amanitin prevented any transcriptional changes and thus blocked the gastrulation movements that are normally initiated by ZGA (Fig. 6d and Supplementary Fig. 8d). The percentage of zygotic genes with reduced transcript levels ranged from ~2% to ~25% caused by BMP and mPouV/Sox3 LOFs, respectively (Supplementary Fig. 11a). We note that the additional loss of Sox3 in mPouV LOF embryos further reduced ZGA of developmentally relevant genes, including those that are expressed ubiquitously, like *mir427*, and those that are activated in a subset of mPouV/Sox3+ cells, like *ventx* and *tbxt* (Fig. 6f, g).

Spatial analysis of reduced gene activation confirmed that many of the 918 mPouV/Sox3-dependent genes showed enriched expression along the animal-vegetal or dorso-ventral

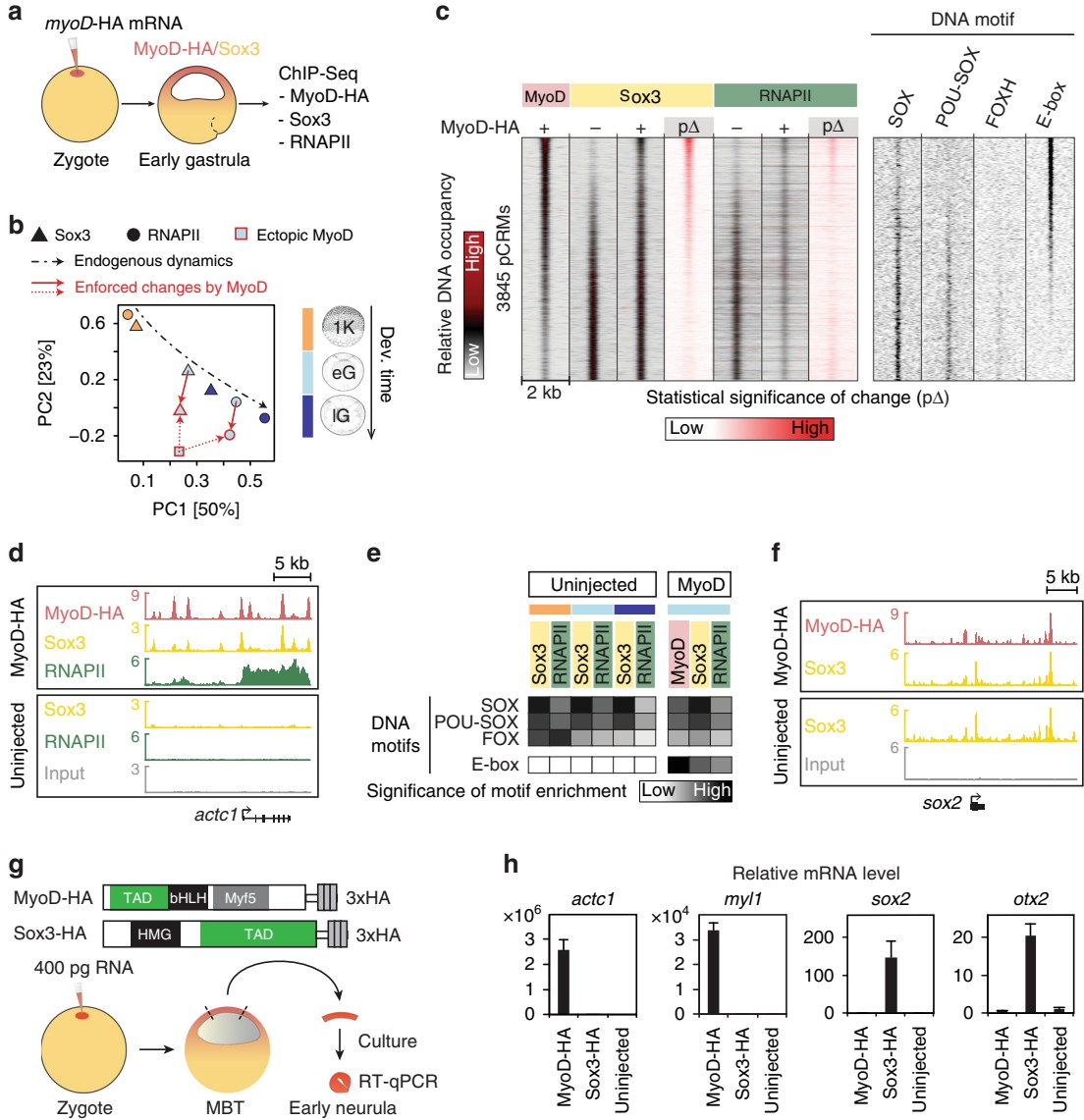

**Fig. 4** Ectopic expression of the muscle determinant MyoD reveals the effect of co-expressed TFs on chromatin engagement and gene expression.
**a** Experimental design: Ectopic expression of the MyoD-HA mRNA construct injected into the animal hemisphere followed by the genome-wide chromatin profiling (ChIP-Seq; $n = 2$ biologically independent samples) of MyoD-HA, Sox3 and RNAPII in early gastrula embryos. **b** Biplot of principal component (PC) 1 (accounting for 50% variance) and 2 (23% variance) shows the relationship of Sox3 (triangle), RNAPII (circle) and MyoD-HA (square) binding levels across MyoD+ and/or Sox3+ pCRMs. Arrows show the normal temporal dynamics of Sox3 and RNAPII binding (black dash-dotted line) and the MyoD-enforced (red dotted line) changes to them (red solid line) at early gastrula stage. Fill colour of symbols represents the developmental stage as indicated, while line colours refer to whether MyoD-HA was expressed (red) or not (black). Abbreviations used for the developmental timeline: 1 K, 1024-cell stage; eG and lG, early and late gastrula stage. **c** Heat map of the DNA occupancies of 3845 pCRMs and enriched DNA motifs sorted by the significance of MyoD-HA-enforced changes (pΔ) to Sox3 binding levels. **d**, **f** Snapshot of chromatin co-recruitment to the super-enhancers of canonical MyoD and Sox3 target genes *actc1* (**d**) and *sox2* (**f**), respectively. **e** Heat map shows the significance of DNA motif enrichments (y-axis) at 10,000 pCRMs most frequently occupied by the indicated proteins in uninjected and MyoD-HA injected embryos (x-axis). **g** Experimental design: Animal cap assay to quantify MyoD- or Sox3-enforced transcription. **h** RT-qPCR results from the animal cap assay. Error bars, mean + s.d. ($n = 2$ biologically independent samples)

axes (Fig. 7a, e). More specifically, comparison with signal LOFs revealed that 268 of the 708 genes induced by Wnt, Nodal or BMP also depended on mPouV/Sox3 (Fig. 7b and Supplementary Figs. 9 and 10b). Similarly, 175 of the 239 mVegT-dependent genes, including such as *foxa1*, *nodal5/6* and *sox17a/b*, were activated in response to signalling (Supplementary Figs. 9 and 10a). Loss of mVegT and mPouV/Sox3 jointly affected 71 genes, 57 of which also depended on Wnt, Nodal or BMP signalling (Supplementary Fig. 10b).

Remarkably, the requirement for particular maternal TFs varied even among related genes with similar expression patterns

that are activated by the same signals. For instance, *tbxt* and *eomes* are both activated by Nodal signalling in the marginal zone between the animal and vegetal hemispheres, but only *tbxt* requires mPouV/Sox3 for its signal responsiveness (Fig. 7a–c). Significantly, DNA occupancy levels did not explain the difference in gene regulation between *tbxt* and *eomes*, because both gene loci showed high levels of Sox3 binding (Supplementary Fig. 10c). Similar dependencies were found for Wnt-responsive genes like *foxb1* and *zic1* (Supplementary Fig. 10d). We confirmed these mPouV/Sox3-dependent signal inductions by treating control and mPouV/Sox3-depleted animal cap tissue

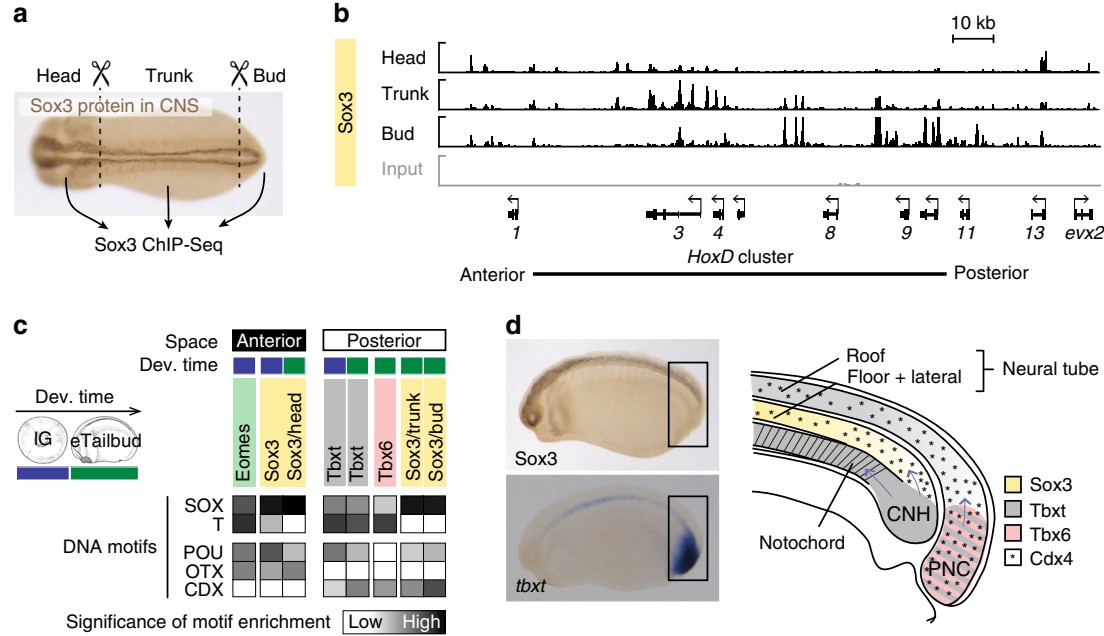

**Fig. 5** Profiling chromatin for Sox3 in different anterior–posterior compartments of the central nervous system (CNS). **a** Experimental design: Genome-wide Sox3 profiling of head, trunk and bud dissected from early tailbud embryos after tissue fixation. **b** Snapshot of Sox3 binding to *HoxD* cluster in the head, trunk and bud. Note the differential binding of Sox3 with 'anterior', 'middle' and 'posterior' Hox genes being preferentially occupied in head, trunk and bud, respectively. **c** Heat map shows the significance of DNA motif enrichments (*y*-axis) across 10,000 pCRMs most frequently occupied in each chromatin profile (*x*-axis). Profiles are grouped depending on whether the cells expressing the TF of interest preferentially contribute to the anterior or posterior compartment. Abbreviations used for the developmental timeline: lG, late gastrula; and eTailbud, early tailbud. **d** Anatomical map of TF expression at the posterior end of an early tailbud embryo to explain the co-enrichment of 'posterior' TF-specific DNA recognition motifs in **c**. Sox3-expressing cells of the posterior neural tube originate from the chordoneural hinge (CNH) and posterior wall of the neurenteric canal (PNC) expressing Brachyury and Brachyury/Tbx6, respectively[21]. Both neural tube and PNC are also exposed to the expression of Cdx such as Cdx4[101].

with Nodal (Activin) and Wnt (CHR99021) agonists (Fig. 7d and Supplementary Fig. 10e). Interestingly, mPouV/Sox3 also facilitated basal low-level expression of (especially) *tbxt* and *foxb1* without Nodal or Wnt stimulation.

Together, our selected maternal TFs and signals activated 1473 of the 3687 (~40%) zygotic genes. These included substantial percentages of genes with preferential expression in the animal (~62%), the vegetal (~89%), the dorsal (~95%) and the ventral (~94%) parts of the embryo, while many ubiquitously expressed genes remained unaffected (Fig. 7e). mPouV/Sox3 and mVegT tend to affect biological functions for which signal-induced regionalisation of ZGA is essential, such as the formation of the main body axes and the segregation of the three germ layers (Supplementary Fig. 11b). We also note that LOF of mPouV/Sox3 caused significant increases in transcript levels of genes encoding gamete-specific biological processes, suggesting that reprogramming towards embryonic pluripotency was compromised (Supplementary Fig. 11a, b).

**Pioneering activity of mPouV/Sox3 predefines signal response.** To discover how maternal TFs allow cell type-specific genes to be signal-induced, we compared various chromatin features from genome-wide accessibility (DNase-Seq; Fig. 8a and Supplementary Fig. 12a) and DNA occupancy (ChIP-Seq for H3K4me1, RNAPII, Smad2 and β-catenin; Fig. 8a) to the high-resolution conformation contacts (next-generation capture-C; Fig. 9a, Supplementary Fig. 12b) of 30 selected promoters (Fig. 9b and Supplementary Data 10) between control and mPouV/Sox3 LOF embryos at the MBT (Supplementary Data 1). We selected mPouV/Sox3 rather than mVegT because of their stronger effect on the ZGA (Supplementary Fig. 11a). Genome-wide analysis showed that mPouV/Sox3 LOF strongly reduced chromatin accessibility as measured by

the significant (FDR ≤10%) loss of DNase cleavages in 6738 of the 16,637 (~41%) pCRMs (Fig. 8b, c and Supplementary Data 11). Sorting of these pCRMs according to the significance of lost accessibility (Fig. 8d) suggests that chromatin opening depends on the pioneering activity of mPouV and Sox3 to recognise their canonical motifs in compacted chromatin (Fig. 8e). By comparison, unaffected pCRMs contained canonical POU/SOX motifs less frequently, and were strongly enriched for promoter-centric motifs of the Krüppel ZF, bZIP and NFY protein families (Fig. 8e, f). At the extreme end of affected loci (e.g., *tbxt*, *foxb1*, *cdc25b* and *zic1*) entire or large proportions of super-enhancers became inaccessible upon mPouV/Sox3 LOF (Figs. 9d and 10a and Supplementary Figs. 13 and 14).

The loss of accessibility triggered further profound changes to chromatin in situ (see arrowheads at bottom of Figs. 9d and 10a and Supplementary Figs. 13 and 14 for strongly affected pCRMs). First, the deposition of H3K4me1, RNAPII and signal mediators Smad2 and β-catenin was substantially reduced (Figs. 8d and 9d and Supplementary Fig. 13 and 14). Second, in contrast to unaffected pCRMs, 32 pCRMs with compromised accessibility— most of which are part of super-enhancers—also showed significantly (FDR ≤10%) reduced promoter contacts (e.g., *tbxt*, *foxb1*, *cdc25b* and *zic1*) (Figs. 9c, d and 10a, Supplementary Figs. 13 and 14 and Supplementary Data 12). Third, at the transcriptional level, lower usage of (clustered) pCRMs coincided with the reduction of coding (Fig. 8g) as well as local non-coding RNA (last column in Fig. 8d and 'low RNA' track in Fig. 9d and Supplementary Figs. 13 and 14). Importantly, the profiling of LOF embryos revealed how chromatin predetermines signal induction and why, for instance, Nodal-induced transcription of *tbxt* was strongly affected by the loss of mPouV/Sox3 and that of *eomes* less so. In contrast to those of *eomes*, all promoter-tied

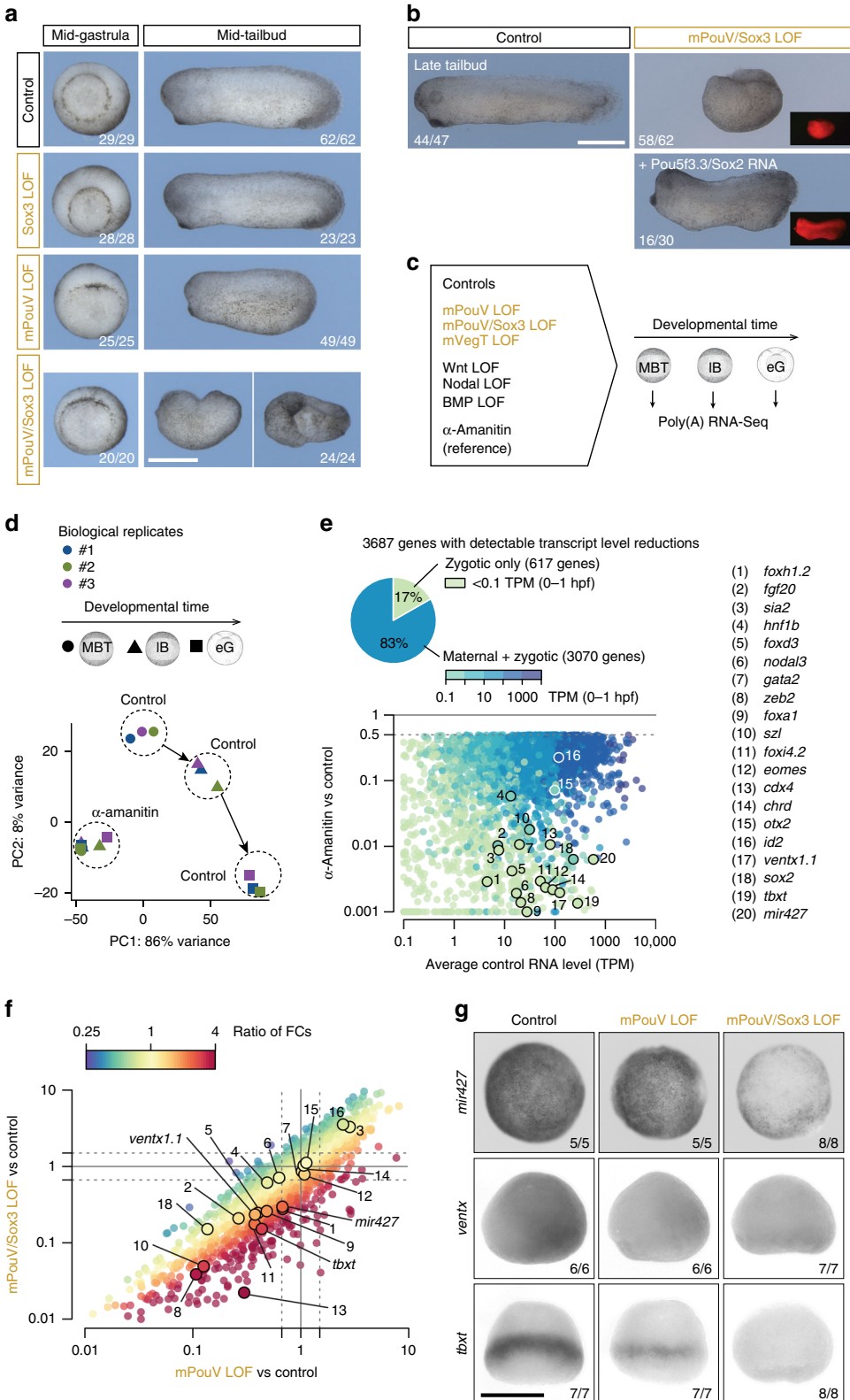

pCRMs (super-enhancer) of *tbxt* contain canonical POU/SOX motifs, so were not accessible to Smad2 in mPouV/Sox3 LOF embryos (Fig. 10a, b). Thus, Smad2 interactions with critical Nodal responsive CRMs of *eomes* remained intact, but those of *tbxt* were impeded by compacted chromatin. Pioneer-initiated competence also applied to Wnt signalling with β-catenin failing to engage with Wnt responsive CRMs of *foxb1* in the absence of

mPouV/Sox3 (Supplementary Fig. 13). Other transcriptional deficiencies in signal-responsive or non-responsive genes that were triggered by mPouV/Sox3 LOF also coincided with significant reductions (FDR ≤10%) in chromatin accessibility (Fig. 10c and Supplementary Figs. 15, 16 and 17), suggesting that mPouV/Sox3 facilitates expression by making critical clusters of CRMs accessible to signal and other transcriptional mediators.

**Fig. 6** Synergistic relationship between maternal Pou5f3 (mPouV) and Sox3. **a** Morphological phenotypes caused by single and combined LOFs of Sox3 and mPouV when control embryos reached mid-gastrula and mid-tailbud stage. See also Supplementary Movie 1. **b** Phenotypical rescue of mPouV/Sox3 LOF embryos by the co-injection of both *X. laevis* Pou5f3.3 and *Sox2* mRNA alongside *mCherry* mRNA as a tracer. **c** Experimental design: Profiling the poly(A) RNA transcriptome (*n* = 3 biologically independent samples) over three consecutive developmental stages under indicated conditions. Abbreviations used for the developmental timeline: MBT, midblastula transition; lB, late blastula; and eG, early gastrula. **d** Biplot of PC1 (accounting for 86% variance) and PC2 (8% variance) shows the relationship of developmental stage-specific poly(A) RNA transcriptomes of control and α-amanitin-injected embryos in biological triplicates (#1–3). **e** Detection of 3687 zygotic genes with reduced transcript levels (≥ 50%, FDR ≤10%) in α-amanitin-injected embryos. These genes were used as reference for all other LOFs. Dots in scatterplot are coloured according to the maternal contribution[56] to the transcript level of each of these zygotic genes. **f** Scatterplot of transcript fold changes (FCs) caused by mPouV and mPouV/Sox3 LOFs with dots coloured according to the ratio of FCs. Numbered dots refer to some developmentally relevant genes listed in **e**. **g** Early gastrula-staged WMISH: *mir427*, animal view; *ventx*, lateral view, ventral side facing right; and *tbxt*, dorsal view. Numbers in the right bottom corner of each image refer to the count of embryos detected with the displayed WMISH staining among all embryos analysed per condition and in situ probe. Scale bars (**a**, **b**, **g**), 0.5 mm

## Discussion

Our results allow us to propose a model (Fig. 10d) of pioneer-initiated chromatin remodelling, or priming, that unlocks context-specific CRMs, some of which contain signal-responsive elements that enhance and regionalise transcription. Our observations are in line with recent reports of CRM priming for tissue-specific gene expression[19,35,36] and signal interpretation[37,38]. Thus, cell type-specific TFs pre-determine signal interpretation by shifting the binding of signal mediators and so causing the induction of different genes. These, ultimately, result in different cell fate transitions. Developmental context is imprinted on chromatin by co-expressed sequence-specific TFs recognising their canonical DNA binding motifs in compacted chromatin. Based on in vitro experiments[39], partial DNA motifs exposed on nucleosome surfaces can be sufficient to initiate chromatin access. Subsequent displacement of nucleosomes may be driven by cooperation among several TFs and the higher affinity of TFs for free genomic DNA. In the context of establishing pluripotency before the MBT, we show that the pioneering activity of maternal Pou5f3 (mPouV) and Sox3 recognising their POU/SOX motifs (Fig. 10d, left branch in top panel) triggers extensive chromatin remodelling in situ. This includes the H3K4me1 marking of CRM-flanking nucleosomes and chromatin looping with promoters. Newly primed CRMs can initiate further canonical as well as opportunistic (sequence-nonspecific) binding. Thus, irrespective of their preference for certain DNA sequences, chromatin factors (e.g., TFs, signal mediators, RNAPII) can appear on accessible CRMs with no canonical DNA binding motifs (Fig. 10d, right branch in top panel). However, the opportunistic recruitment of mPouV and Sox3, at least, seems to have little to no effect on changing the chromatin landscape and its transcriptional readout. This may be also true for other TFs as shown for ectopic MyoD in our study. Such recruitment behaviours can be difficult to detect without multi-level analysis of chromatin under LOF conditions. Overall, the canonical binding of mPouV/Sox3 strongly contributes to the pioneering of two fifths of all accessible pCRMs at the MBT which permits or instructs one quarter of the ZGA (Fig. 8b, g). We often observe that the usage of pCRMs also coincides with the low-level activation of proximal non-coding RNA.

Although we have not performed chromatin profiling at a single-cell level, we suggest for three reasons that this priming occurs in every embryonic cell before the onset of regional Wnt, Nodal or BMP signalling. First, the nuclear accumulation of maternal TFs (e.g., mPouV, Sox3, Foxh1 and mVegT) occurs ubiquitously (or within the vegetal hemisphere in case of mVegT) and precedes that of signal mediators. Second, mPouV/Sox3 LOF affects chromatin similarly at zygotic genes with (e.g., *foxb1*) or without (e.g., *cdc25b*) tissue-specific expression. And finally, dissected animal caps replicate the in vivo response of the marginal zone tissue to Nodal and Wnt signalling. Importantly, and consistent with the idea of context-dependent signal interpretation, our findings suggest that signal mediators have inferior pioneering activity and thus rely on cooperating pioneer factors like mPouV and Sox3 to make their signal response elements accessible. Analysis of the related genes *eomes* and *tbxt*, both of which are induced by Nodal signalling, but with only *tbxt* depending on mPouV/Sox3, reveals that the same context-dependent competence can be differentially encoded in the genome (through the presence or absence of POU/SOX motifs in signal-responsive CRMs) (Fig. 10d, lower panel). Based on transcriptional dependencies shown here and in the spinal cord[40], this regulation of competence is complemented by other TFs like mVegT and also applies to Wnt signalling and other signalling pathways such as BMP and Sonic hedgehog.

Our work has focused on the pioneering activity of mPouV and Sox3. The enrichment of DNA recognition motifs at pCRMs in use before the MBT suggests that only a few additional maternal TF families, including those of T-box, FoxH and Krüppel-like ZF families, are involved in the bulk acquisition of competence at the MBT (Fig. 2b). Indeed, many of these TF family members are rapidly turned over and are most abundant around the MBT[41,42], suggesting dynamic and strong TF activity[43]. Their activity generates a signature of accessible DNA motifs of pluripotency or early cell lineage determination that is conserved among vertebrates and is reminiscent of the chromatin footprints in embryonic stem cells in vitro[44]. Remarkably, as in the human genome[44], POU/SOX motifs populate distal CRMs, while Krüppel ZF motifs are frequently found in promoter-proximal CRMs (Fig. 8f). This creates a functional separation among the pluripotency factors where PouV/SoxB1 and Krüppel-like factors remodel enhancers and promoters, respectively. Collectively, we show that mPouV/Sox3 predetermine cell fate decisions by initiating access to signal-responsive CRMs via POU/SOX motifs. The use of these permissive CRMs is most prominent in pluripotent cells[25,45] and proves to be critical in vivo to the induction of zygotic genes with functions in germ layer formation and primary axis determination.

The findings presented here confirm that core pluripotency TFs contribute substantially to ZGA as previously reported for SoxB1, PouV and Nanog TFs for zebrafish ZGA[46,47]. However, in contrast to zebrafish or humans, a true *nanog* gene has not been found in the *Xenopus* genome, which has led to the suggestion that zygotic Ventx TFs with their Nanog-like DNA binding characteristics could operate as the gatekeepers of pluripotency instead[48]. This notwithstanding, we anticipate that signal competence is controlled across vertebrate embryonic stem cells in a manner resembling that in the early *Xenopus* embryo.

The approach we adopt here can be applied to any other cell type to discover the molecular basis of its competence. Ultimately, this will generate a lexicon of competence that outlines which pioneer factors unlock which (signal-responsive) CRMs and gene

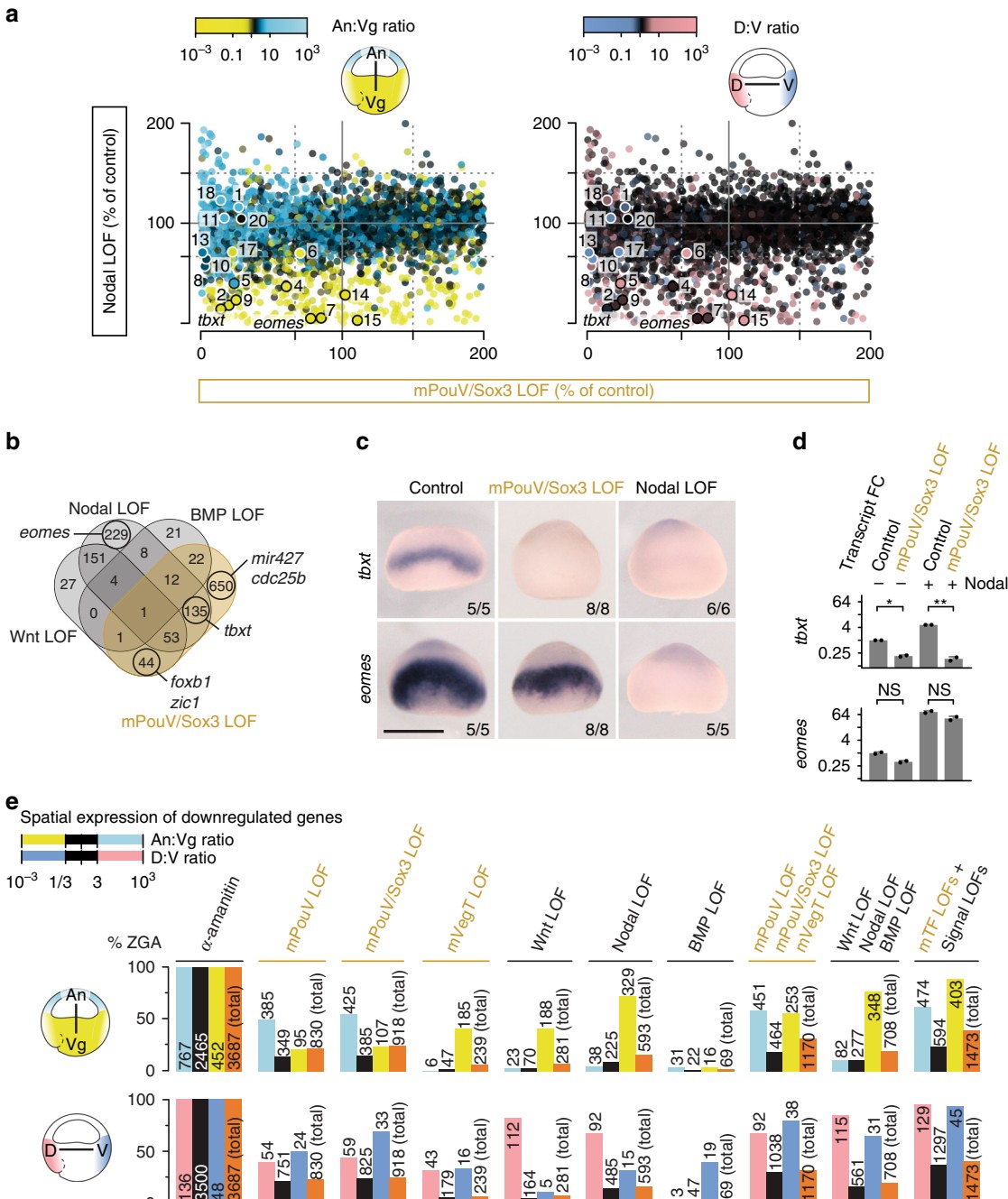

**Fig. 7** Signal-induced regionalisation of ZGA depends on maternal TFs. **a** Transcriptional comparison of zygotic genes between the LOFs of maternal PouV/Sox3 and Nodal signalling. Dots are coloured according to the normal ratio of transcript levels (regional expression[79]) across the animal-vegetal (An:Vg) or dorso-ventral (D:V) axis. Numbered dots refer to genes listed in Fig. 6e. **b** Venn diagram of genes downregulated by the mPouV/Sox3 LOF (orange) or the LOF of single signal transduction pathways (black). **c** Early gastrula-staged WMISH for *tbxt* and *eomes* under indicated LOFs. Scale bar, 0.5 mm. **d** Quantification of *tbxt* and *eomes* transcript levels in control and mPouV/Sox3 LOF animal caps with or without stimulated Nodal signalling. Error bars, mean + s.d. ($n = 2$ biologically independent samples). Two-tailed Student's *t*-test: *$p = 0.014$; **$p = 0.005$; and NS, not significant ($p \geq 0.02$). **e** Bar graphs show the percentage (and number) of downregulated zygotic genes (% ZGA under indicated LOFs) grouped by the normal ratio of transcript levels (regional expression[79]) across the animal-vegetal (An:Vg) and dorso-ventral (D:V) axis

loci. For example, the motif compositions of CRMs later engaged in the *Xenopus* tailbud embryo points at the potential of OTX and CDX TFs in conveying competence in anterior and posterior compartments, respectively (Fig. 5c). Interestingly, various tumours are associated with the mis-expression of pioneer factors[49] and thus may display different competencies from the surrounding host tissue. Conversely, knowing which pioneers are required to permit a specific signal response will increase the success of engineering patient-specific tissue for transplantation therapy. On a broader level, our profiling of chromatin states under LOF conditions discriminates functional from nonfunctional binding and provides a promising avenue for deciphering the non-coding part of the genome for basic and therapeutic research.

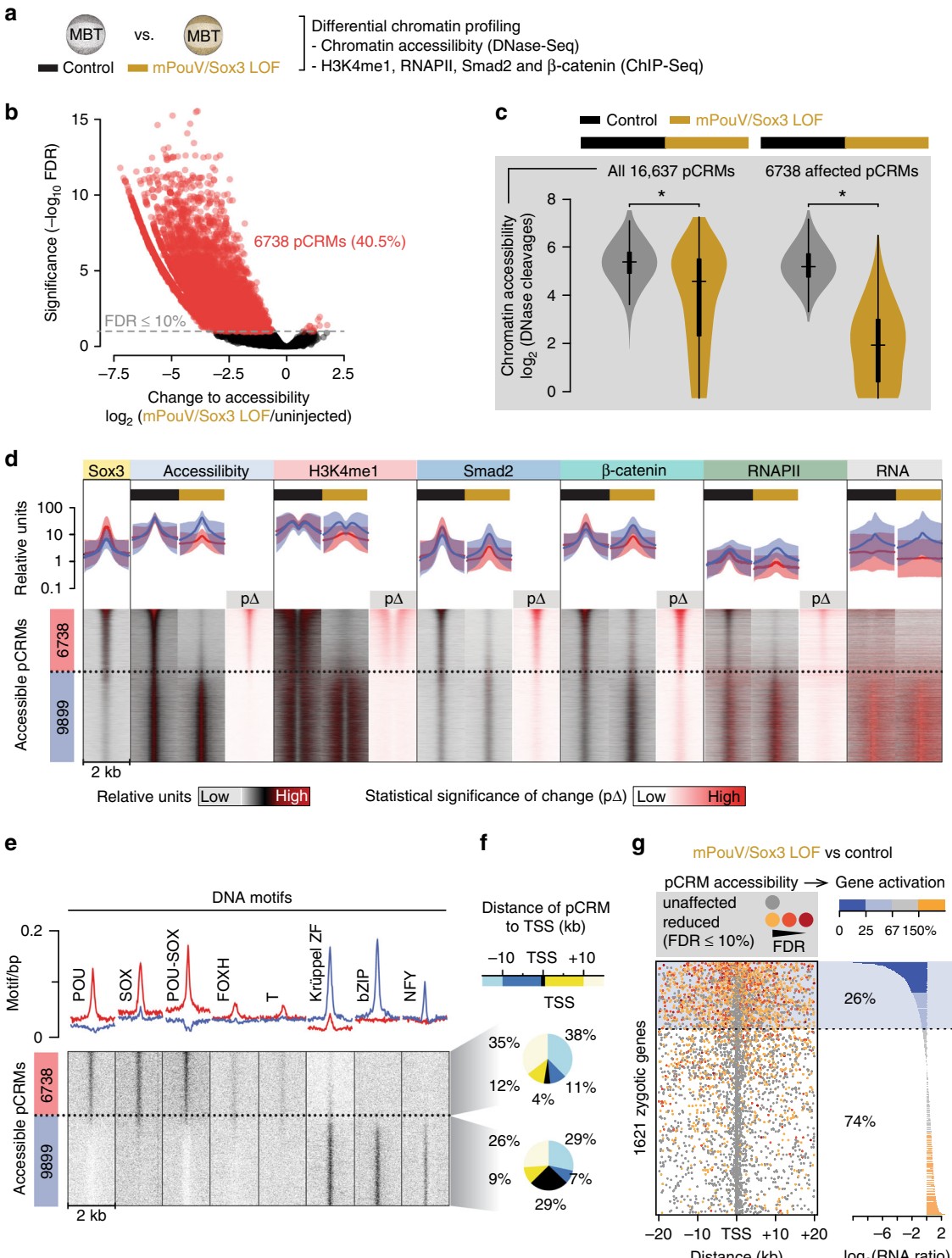

## Methods

**Xenopus manipulations.** Standard procedures were used for ovulation, fertilisation, and manipulation and incubation of embryos[50,51]. Briefly, frogs were obtained from Nasco (Wisconsin, USA). Ovulation was induced by injecting serum gonadotropin (Intervet) and chorionic gonadotropin (Intervet) into the dorsal lymph sac of mature female frogs. Eggs were fertilised in vitro with sperm solution consisting of 90% Leibovitz's L-15 medium (Gibco) and 10% foetal calf serum (Gibco). After 10 min, fertilised eggs were de-jellied with 2.2% (w/v) L-cysteine (Sigma, 168149) equilibrated to pH 8.0. *X. tropicalis* embryos were cultured in 5% Marc's Modified Ringer's solution (MMR)[51] at 21 °C–28 °C. *X. laevis* embryos were cultured in 10% Normal Amphibian Medium (NAM)[51] at 14 °C–22 °C. Embryos

were staged according to Nieuwkoop and Faber[52]. All *Xenopus* work fully complied with the UK Animals (Scientific Procedures) Act 1986 as implemented by the Francis Crick Institute.

**Nucleic acid injections and treatments of embryos.** Microinjections were performed using calibrated needles and embryos equilibrated in 4% (w/v) Ficoll PM-400 (Sigma, F4375) in 5% MMR or 10% NAM. Microinjection needles were generated from borosilicate glass capillaries (Harvard Apparatus, GC120–15) using the micropipette puller Sutter p97. Maximally two nanolitres of morpholino (MO) and/or mRNA were injected into de-jellied embryos at the 1-cell, 2-cell or 4-cell

**Fig. 8** Maternal pluripotency TFs mPouV and Sox3 remodel ~40% of the accessible chromatin landscape to contribute to ~25% of ZGA. **a** Used approach to reveal the effect of mPouV/Sox3 on chromatin accessibility (DNase-Seq) and chromatin composition (ChIP-Seq) around the MBT ($n = 2$ biologically independent samples). **b** Double-logarithmic volcano plot shows massive chromatin accessibility (DNase cleavage) reductions caused by mPouV/Sox3 LOF. pCRMs (dots) with significant accessibility changes (FDR ≤10%) are marked in red. **c** Violin plots show the comparison of chromatin accessibility (DNase cleavages) between uninjected and mPouV/Sox3 LOF embryos at all and affected (FDR ≤10%) pCRMs. Wilcoxon test: $*p < 2.2 \times 10^{-16}$. **d** Normalised meta-plots (top row, mean ± s.d.) and heat maps (bottom row) show the level of chromatin accessibility, chromatin engagements and RNA across accessible pCRMs in uninjected and mPouV/Sox3 LOF embryos. RNA was profiled at and beyond the MBT as shown in Fig. 6c. In the heat map, the pCRMs are sorted and grouped by significantly reduced DNase cleavages under mPouV/Sox3 LOF: red group, affected (FDR ≤10%) and blue group, unaffected (FDR >10%). These groups are represented in the meta-plots. Each heat map under mPouV/Sox3 LOF are followed by a heat map showing the statistical significance (Wald test) of changes (pΔ) caused by mPouV/Sox3 LOF. **e** Heat map shows the occurrence of DNA motifs at accessible pCRMs sorted and grouped as in **d**. **f** Pie charts summarise the distribution of distances (kb) to nearest zygotic TSSs of affected (top pie chart) and unaffected (bottom pie chart) pCRMs. **g** Panel compares the effect of mPouV/Sox3 LOF on chromatin accessibility and RNAPII-mediated gene expression. Plot to the left shows the localisation of accessible pCRMs (affected, dot coloured in orange to red with FDR decreasing from 10%; and unaffected, grey dot) relative to the zygotic TSSs that are active by the MBT[92] and produce enough RNA transcripts to show significant ≥ two-fold reductions upon α-amanitin injection (Fig. 6e). Gene loci are sorted by mPouV/Sox3 LOF-induced transcript fold changes as shown in the log-scaled bar graph to the right

---

stage using the microinjector Narishige IM-300. Injections for the mVegT loss-of-function (LOF) and rescue were targeted to the vegetal hemisphere. All other injections were targeted to the animal hemisphere. Embryos were transferred to fresh 5% MMR or 10% NAM (without Ficoll) once they reached about blastula stage.

The following mRNA amounts were used for ectopic protein expression: 80 pg and 400 pg *MyoD-HA* mRNA into *X. tropicalis* and *X. laevis* embryos, respectively; and 400 pg *Sox3-HA* into *X. laevis* embryos.

The following MO amounts and treatments were used for LOF experiments: 5 ng *Pou5f3.2* MO, 5 ng *Pou5f3.3* MO and 5 ng standard control MO (mPouV LOF); 5 ng *Pou5f3.2* MO, 5 ng *Pou5f3.3* MO and 5 ng *Sox3* MO (mPouV/Sox3 LOF); 5 ng *β-catenin* MO (Wnt LOF); 10 ng m*VegT* MO (mVegT LOF); standard control MO (5–10 ng according to the dose used for the β-catenin or mVegT LOF experiment); and 30 pg α-amanitin (BioChemica, A14850001). To block Nodal (Nodal LOF) or BMP (BMP LOF) signalling, embryos were treated with 100 μM SB431542 (Tocris, 1614) or 10 μM LDN193189 (Selleckchem, S2618) from the 8-cell stage onwards. Control embryos were treated accordingly with DMSO, in which these antagonists were dissolved. MOs were designed and produced by Genetools (Oregon, USA) to block translation. β-catenin MO (5′-TTTCAACAGT TTCCAAAGAACCAGG-3′, underlined base was changed from the original MO[27]), Pou5f3.2 MO (5′-GCTGTTGGCTGTACATAGTGTC-3′, underlined base was changed from the original MO[26]), Pou5f3.3 MO[25] (5′-GTACAGAACG GGTTGGTCCATGTTC-3′), Sox3 MO (5′-GTCTGTGTCCAACATGCTATA CATC-3′), Tbx6 MO (5′-TACATTGGGTGCAGGGACCCTCTCA-3′) and mVegT MO[53] (5′-TGTGTTCCTGACAGCAGTTTCTCAT-3′; see Supplementary Fig. 2e).

For the rescue of mVegT LOF, 25–50 pg z*VegT* mRNA were injected at the 2- or 4-cell stage. For the rescue of mPouV/Sox3 LOF, 75 pg *Pou5f3.3* mRNA and 75 pg *Sox2* mRNA were injected at the 2- or 4-cell stage. For both rescue experiment, 150–300 pg *mCherry* mRNA was co-injected to trace mRNA expression in embryonic cells.

**RNA constructs.** All constructs are products of Gateway (Invitrogen) cloning. First, the coding sequences of interest were amplified from either plasmid using *Pfu*Turbo DNA polymerase (Agilent, 600250) or cDNA using Phusion HF DNA polymerase (NEB, M0530), and unidirectionally cloned into the pENTR/D-TOPO vector (Thermo Fisher Scientific, K2400). Next, these clones were LR-recombined with Gateway-compatible pCS2+ vectors containing no tag or a C-terminal 3xHA tag (see Fig. 4g). The following forward and reverse primers were used for PCR amplification: *X. laevis myoD1a* from *MyoD*2–24 pBluescript plasmid[54], 5′-CACC ATGGAGCTGTTGCCCCCAC-3′, 5′-TAAGACGTGATAGATGGTGCTG-3′; *X. laevis sox3* from the IMAGE:443920 clone (European *Xenopus* Resource Centre), 5′-CACCATGTATAGCATGTTGGACAC-3′, 5′-TATGTGAGTGAGCGGTACAG TG-3′; *X. laevis sox2* from stage 11 cDNA, 5′-CACCATGTACAGCATGATGGAG ACCGA-3′, 5′-TCACATGTGCACAGAGGCAGC-3′; *X. laevis pou5f3.3* from stage 11 cDNA, 5′-CACCATGGACCAGCCCATATTGTACA-3′, 5′-TCAGCCGG TCAGGACCCCCA-3′; *X. tropicalis* z*VegT* from stage 11 cDNA, 5′-CACCATGCA CTCTCTGCCGGATGTA-3′, 5′-TTACCAACAGCTGTATGGAAAGA-3′. pCS2+ 8Nm*Cherry*[55] (34936) was obtained from Addgene.

**In vitro transcription of capped mRNA.** About 10 μg plasmid was linearised by restriction digestion and purified using the QIAquick PCR purification kit (Qiagen, 28104). Capped mRNA was produced from ~1 μg linearised plasmid using mMessage mMachine SP6 (Thermo Fisher Scientific, AM1340) according to the manufacturer's instructions. The following restrictions enzymes (NEB) were used for linearising the pCS2+ constructs: *Asp*781 (*myoD1a*, *sox3*), *Pvu*II (*sox2*), *Psp*OMI (*pou5f3.3*, z*VegT*) and *Not*I (*mCherry*).

**Generation of digoxigenin-labelled RNA probes.** DNA template for generating the *foxb1* in situ hybridisation probe was amplified from *X. tropicalis* stage 19–20 cDNA. KAPA HiFi HotStart polymerase (Kapa Biosystems, KK2602) and the following PCR cycling conditions were used: 98 °C for 45 s followed by 40 cycles (98 °C for 10 s, 63 °C for 10 s, 72 °C for 15 s) and final elongation step of 30 s at 72 °C. Primers were 5′-GTCAGCGCCTATGGAGTACC-3′ and 5′-AACACTGG AGATGCCATGC-3′. Fresh PCR product was zero-blunt cloned into the pCRII-TOPO vector (Thermo Fisher Scientific, K2800). The identity and direction of insert was verified by restriction digest and Sanger sequencing. Plasmids were linearised by restriction digestion and purified using the QIAquick PCR purification kit (Qiagen, 28104). All in situ hybridisation probes were transcribed from ~1 μg linearised plasmid using 1x digoxigenin-11-UTP (Roche, 11277065910), 40 U RiboLock RNase inhibitor (Thermo Fisher Scientific, EO0381), 1x transcription buffer (Roche) and T7 RNA polymerase (Roche, 10881767001) at 37 °C for 2 h. The probe was treated with 2 U Turbo DNase (Thermo Fisher Scientific, AM2238) to remove the DNA template and purified by spin-column chromatography (Clontech). RNA was diluted to 10 ng/μl (10× stock) with hybridisation buffer (50% formamide, 5× SSC, 1× Denhardt's, 10 mM EDTA, 1 mg/ml torula RNA, 100 μg/ml heparin, 0.1% Tween-20 and 0.1% CHAPS) and stored at −20 °C. The following plasmids and restriction enzymes (NEB) were used for plasmid linearisation: *X. laevis eomes* pCRII-TOPO[20], *Bam*HI; *X. tropicalis foxb1* pCRII-TOPO, *Bam*HI; *X. tropicalis mir427* pCS108 (IMAGE:7545411)[56], *Sal*I; *X. laevis ventx2.1* pBluescript SK-[57], *Eco*RI; *X. laevis tbxt* (*Xbra*) pSP73[58], *Bgl*II; and *X. tropicalis zic1* pCMV-SPORT6 (IMAGE:7668846), *Sal*I.

**Animal cap assays.** These assays were carried out for the ectopic expression of mRNA constructs in *X. laevis* and for the MO-mediated LOF in *X. tropicalis*. All animal caps were dissected at the blastula stage (stage 8 to 9) and cultured in 75% NAM containing 0.1% bovine serum albumin (Sigma). Dissections were carried out with 13 μm loop electrodes (Protech International, 13-Y1) connected to a MC-2010 micro cautery instrument (Protech International) operating at power level 2. As illustrated in Fig. 4g, control (uninjected) and MyoD-HA or Sox3-HA expressing animal caps were cultured at 20 °C until sibling embryos reached early neurula stage (stage 13). For the experiment shown in Fig. 7d and Supplementary Fig. 10e, animal caps of control and mPouV/Sox3 LOF embryos were cultured with or without 10 ng/ml recombinant human activin A (Nodal agonist; R&D Systems, 338-AC) or 50 μM CHIR99021 (canonical Wnt agonist; Tocris, 4423) at 22 °C for ~2 h until sibling embryos reached early gastrula stage (stage 10+).

**Extraction of total RNA.** 10–15 embryos (or 15–20 animal caps) were homogenised in 800 (400) μl TRIzol reagent (Thermo Fisher Scientific, 15596018) by vortexing. The homogenate was either snap-frozen in liquid nitrogen and stored at −80 °C or processed immediately. For phase separation, the homogenate together with 0.2× volume of chloroform was transferred to pre-spun 1.5-ml Phase Lock Gel Heavy microcentrifuge tubes (VWR), shaken vigorously for 15 s, left on the bench for 2 min and spun at ~16,000 × *g* (4 °C) for 5 min. The upper phase was mixed well with one volume of 95–100% ethanol and spun through the columns of Zymo RNA Clean and Concentrator 5 or 25 (Zymo Research) at ~12,000 × *g* for 30 s. Next, the manufacturer's instructions were followed for the recovery of total RNA (>17 nt) with minor modifications. First, the flow-through of the first spin was re-applied to the column. Second, the RNA was treated in-column with 3 U Turbo DNase (Thermo Fisher Scientific, AM2238). Third, the RNA was eluted twice with 25 μl molecular-grade water. The concentration was determined on the NanoDrop 1000 spectrophotometer.

**Reverse transcription (RT).** About 0.2–1 μg total RNA was denatured at 75 °C for 5 min before setting up the RT reaction including 40 U M-MLV (RNase H minus, point mutant) RT (Promega, M3681), 500 μM of each dNTP (NEB), and 10 μM

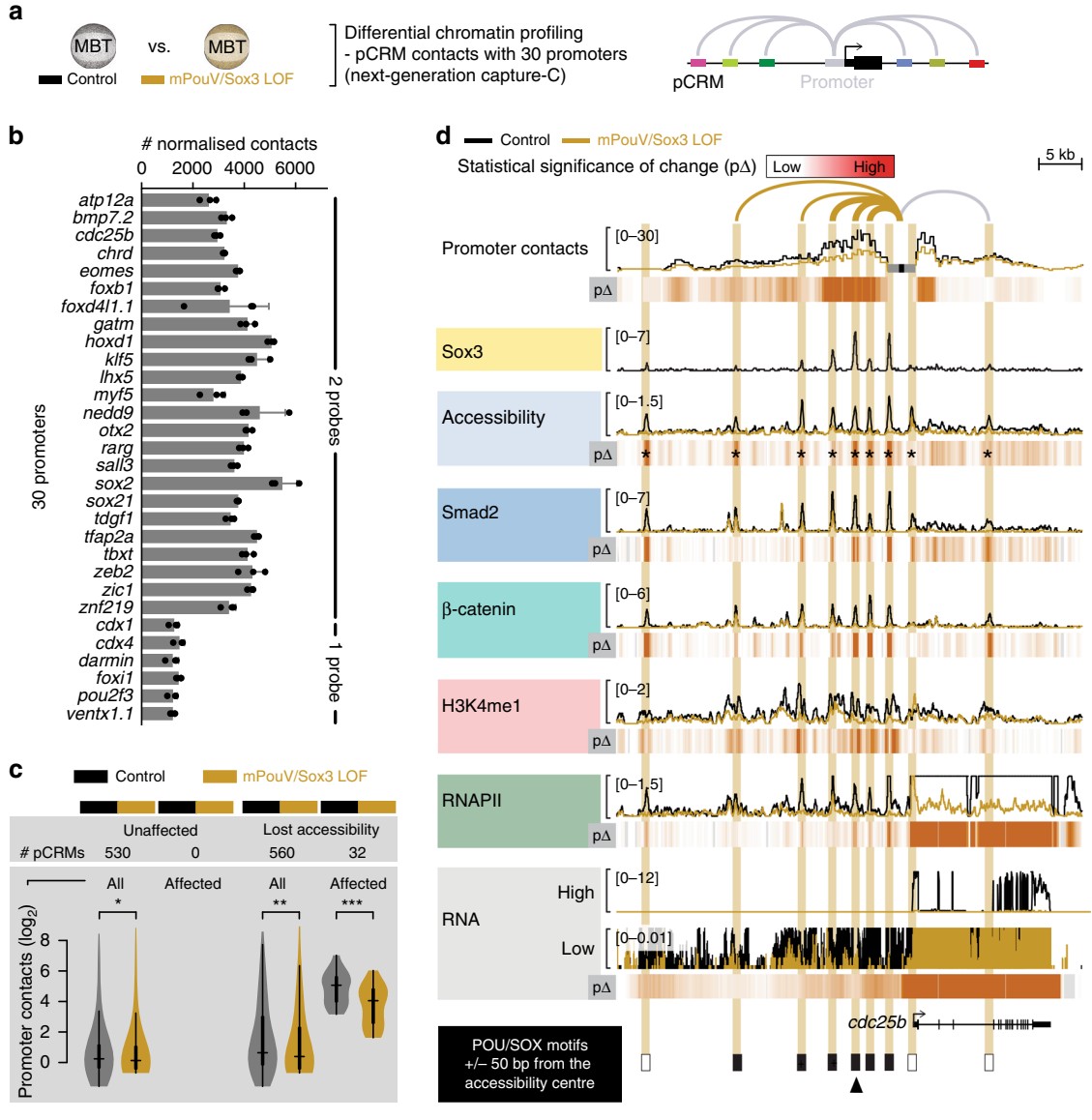

**Fig. 9** Pioneering activity of mPouV/Sox3 initiates extensive chromatin remodelling, such as the chromatin looping of distal pCRMs with promoters. **a** Used approach to reveal the effect of mPouV/Sox3 on chromatin conformation between 30 promoters and distal pCRMs (next-generation capture-C; $n = 3$ biologically independent samples) at the MBT. **b** Bar graph shows the number of normalised promoter contacts (mean + s.d.; $n = 3$ biologically independent samples) derived from non-redundant capture-reporter FLASH[88] reads (Supplementary Fig. 12b) for each promoter captured with one or two probes under control condition (Supplementary Data 10). **c** Violin plots compare the number of promoter contacts with accessible pCRMs between uninjected and mPouV/Sox3 LOF embryos. The comparison is stratified into pCRMs with stable and lost accessibility upon mPouV/Sox3 LOF and shown for both all and affected (FDR ≤10%) promoter contacts. Wilcoxon tests and effect size estimates: *$p = 5 \times 10^{-5}$ and $r_{effect} = 0.12$ (small effect); **$p = 5 \times 10^{-30}$ and $r_{effect} = 0.34$ (medium effect); and ***$p = 5 \times 10^{-10}$ and $r_{effect} = 0.78$ (large effect). **d** Superimposed line tracks show promoter contact frequencies, chromatin accessibilities and DNA occupancies of various chromatin components (Smad2, β-catenin, H3K4me1 and RNAPII) at the *cdc25b* gene locus between control (uninjected) and mPouV/Sox3 LOF embryos. The RNA track is split into a high (0–12) and low (0–0.01) expression window. Note that the low-expression window shows that locally transcribed non-coding super-enhancer RNA depend on mPouV/Sox3 as well as the gene *cdc25b*. Heat maps (pΔ) below each superimposed line plot show the statistical significance (Wald test) of changes caused by mPouV/Sox3 LOF. The footer highlights the occurrences of canonical POU/SOX motifs (black filled rectangles) at accessible pCRMs (±50 bp from the accessibility centre) and one strongly affected pCRM with an arrowhead. Asterisks on the pΔ heat map mark significant (FDR ≤10%) reductions to pCRM accessibility. pCRMs are boxed in and their frequency of contacts with the *cdc25b* promoter are illustrated with an arc of varying strength. Boxes of affected pCRM and arcs of promoter contacts are coloured orange

random hexamers (Sigma) in a final volume of 10 µl. The following incubation settings were used: 25 °C for 15 min, 37 °C for 15 min, 55 °C for 45 min, 85 °C for 15 min and 4 °C for indefinite. The reaction was diluted with 90–190 µl molecular-grade water before using 1–2.5 µl for each qPCR reaction.

**qPCR primers**. The PCR primers were designed to hybridise at ~60 °C ($T_m$) and to generate 75–125 bp amplicons. Where possible, primers were checked for their

specificity in silico using Jim Kent's PCR tool (http://genomes.crick.ac.uk/cgi-bin/hgPcr). All primer pairs were tested to produce one amplicon of the correct size through diagnostic PCR using RedTaq DNA polymerase (Sigma, D4309). Exceptions were previously designed primer pairs for *X. laevis* (L) *actc1*, *odc1* and *otx2*, all of which generate amplicons of 200–250 bp.

Primers (forward, reverse) for *X. laevis* (L) or *X. tropicalis* (T) cDNA: *actc1* (L, 5′-TCCTGTACGCTTCTGGTCGTA-3′, 5′-TCTCAAAGTCCAAAGCCACATA-3′);

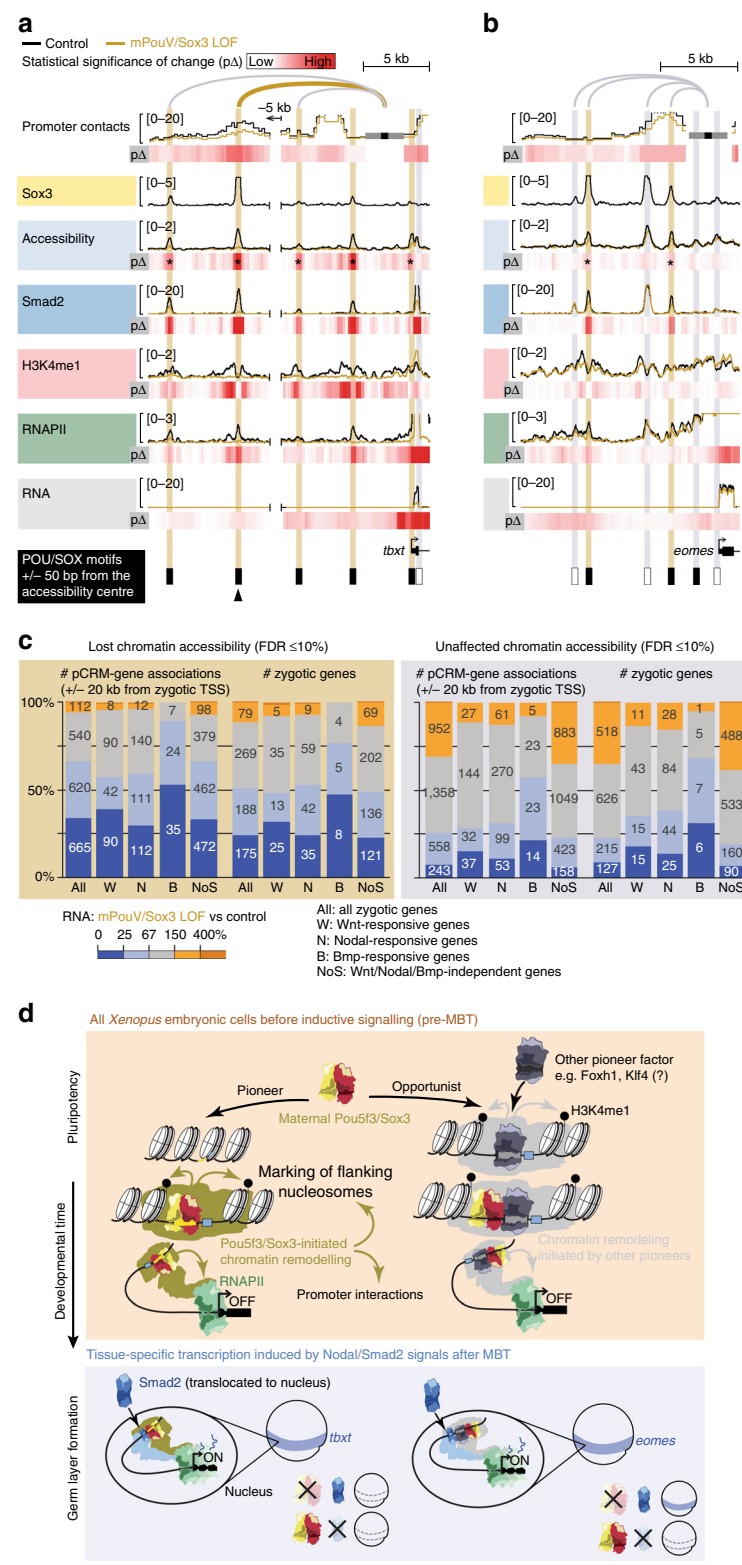

*eomes* (T, 5′-ATTGACCACAACCCATTTGC-3′, 5′-TCTAGGAGAATCCG CAGGAG-3′); *foxb1* (T, 5′-CCGGATGCAGAGTGACAAC-3′, 5′-GAGGCTTC TGGTCGCTGTAG-3′); *gata4* (T, 5′-TCCTCTGATCAAGCCTCAAAG-3′, 5′-AC GGCGCCAGAGTGTAGTAG-3′); *myl1* (L, 5′-GCACCTGCATTTGATCTGT C-3′, 5′-TTGCTGTCTCCTGTCCTGTC-3′); *nodal5* (T, 5′-TTCAAGCCAACAAA CCACGC-3′, 5′-CGCATCTTCACCGGTATGCA-3′); *nodal6* (L, 5′-GCCATTTC GTTGTGAGGGGT-3′, 5′-TCGTACAGCTTGACCAAACTCT-3′); *odc1* (L, 5′- GCCATTGTGAAGACTCTCTCC-3′, 5′-TTCGGGTGATTCCTTGCCAC-3′); *odc1*

(T, 5′-GGGCAAAAGAGCTTAATGTGG-3′, 5′-CATCGTGCATCTG AGACAGC-3′); *otx2* (L, 5′-TTGAACCAGACCTGGACT-3′, 5′-CGGGATGGATT TGTTGCA-3′); *rpl8* (T, 5′-CTGGCTCCAAGAAGGTCATC-3′, 5′-CAGGATGGG TTTGTCAATACG-3′); *sox2* (L, 5′- ACATCACATGGACGGTTGTC-3′, 5′-TCTC TCGGTTTCTGCTCTCG-3′); *sox17b* (T, 5′- GTTTATGGTGTGGGCAAAGG-3′, 5′-TCAGCGATTTCCATGATTTG-3′); *tbx6* (L, 5′-GGGGCACCACGACTCTT AC-3′, 5′-CCCAGAATCCTCTGTGAAGG-3′); and *tbxt* (T, 5′-CCTGTGGATGA GGTTCAAGG-3′, 5′-CACGCTCACCTTTAGAACTGG-3′).

**Fig. 10** Maternal Pou5f3/Sox3-initiated chromatin remodelling to prime the first transcriptional response to inductive signals during ZGA.
**a**, **b** Superimposed line tracks show promoter contact frequencies, chromatin accessibilities and DNA occupancies of various chromatin components (Smad2, H3K4me1 and RNAPII) at the Nodal-responsive mesoderm determinants *tbxt* (**a**) and *eomes* (**b**) between control (uninjected) and mPouV/Sox3 LOF embryos. Heat maps (pΔ) below each superimposed line track show the statistical significance (Wald test) of changes caused by mPouV/Sox3 LOF. The footer highlights the occurrences of canonical POU/SOX motifs (black filled rectangles) at accessible pCRMs (±50 bp from the accessibility centre) and one strongly affected pCRM with an arrowhead. Asterisks on the pΔ heat map mark significant (FDR ≤10%) reductions to pCRM accessibility. pCRMs are boxed in and their frequency of contacts with the promoter are illustrated with an arc of varying strength. Boxes of affected pCRM and arcs of promoter contacts are coloured orange. **c** Stacked bar graphs summarise the correlation of unaffected and significantly reduced (FDR ≤10%) pCRM accessibility with RNAPII-mediated expression of all, signal responsive and non-responsive zygotic genes in mPouV/Sox3 LOF embryos at the MBT. These correlations and corresponding numbers (placed on the stacked bars) are shown for pCRM-gene associations and zygotic genes. TSS-centric maps of reduced chromatin accessibility are shown for signal responsive and non-responsive genes in Supplementary Figs. 15–17. **d** Model of chromatin pioneering and opportunistic engagement to predefine first zygotic responses to inductive signals

Primers (forward, reverse) for *X. tropicalis* genomic DNA: *β-actin* −0.5 kb (5′-TTGTACGCAACACAATGCAG-3′, 5′- AAATGTCAGGACCAAATGCAG-3′); *dlc* −3.2 kb (5′-CATCCTGCAATTAACTGTCTGC-3′, 5′-AGGTGTGACAAATGGCACTG-3′); *gdf3* −0.1 kb (5′-TGGATGAGCACAGAGAGGTG-3′, 5′-GTCAGGAGGGAGGTGTCAAC-3′); *hoxd1* +3.2 kb (5′-TGTTGTAGATGCTGATGCTTATCG-3′, 5′-AACAGAAAATCAAAGGCTTGCA-3′); *mespa* −1.3 kb (5′-CTCCTTTGCCCTCTGAAATG-3′, 5′-CTTCAGGGAATTGGCTTGTG-3′); *msgn1* −0.1 kb (5′-CCAGTCCATTTTCCATGTTG-3′, 5′-GGGGAGGCCCTTTTATACAG-3′); *myf5* −1.1 kb (5′-AAAGAGTTCCTGCACCTTGG-3′, 5′-TCCCAAATCATCAGCACAAC-3′); *nodal6* −0.1 kb (5′-ACATGAGTTTGGCCTTGGTC-3′, 5′-AGGTAACCCCTGAGCTGTCA-3′); *pax3* 5′UTR (5′-CAGATGCCAGAGGAGAGAGC-3′, 5′-GGAACTCTCTGGGTCCCAAC-3′); *pax3* +2.7 kb (5′-ATGAGAATGTGAGCGACACG-3′, 5′-AAACCCAAGGTGAGCAAGTG-3′); *sia1* −0.2 kb (5′-TGGCTTTTAAGTTCGTCTTGG-3′, 5′-CAAAGGGCTTCTCTTTGTGC-3′); *sox3* TSS (5′-GATGGGGCATTAATGGACAG-3′, 5′-CACGGGAGAAAATGAAGCTG-3′); *sox17b* −0.1 kb (5′-GTTAGACTCCGCTCCAGACG-3′, 5′-TCTCCACCTAAGGGGAAACC-3′); *wnt8a* −5.7 kb (5′-GGAAACTGCCTGTGGAAATG-3′, 5′- ACAAAAGCCCACAGGAGATG-3′); *tbxt* −4.0 kb (5′-CAATGGATGCCTCTCTGGAC-3′, 5′-TGGGTTTAGAATGGCAAAGG-3′); and *tbxt* TSS (5′-CACACCTTGGGTTTTGTTCC-3′, 5′-TTCCTCAAGAGAGGGCTTC-3′).

**Whole-mount in situ hybridisation (WMISH)**. WMISH was conducted using digoxigenin-labelled RNA probes. It was based on previously published protocols[51,59]. *X. tropicalis* embryos were fixed in MEMFA (1× MEM and 3.7% formaldehyde) at room temperature for 1 h. The embryos were then washed once in 1× PBS and two to three times in ethanol. Fixed and dehydrated embryos were kept at −20 °C for ≥24 h to ensure proper dehydration before starting hybridisation. Dehydrated embryos were washed once more in ethanol before rehydrating them in two steps to PBT (1× PBS and 0.1% Tween-20). Embryos were treated with 5 μg/ml proteinase K (Thermo Fisher Scientific, AM2548) in PBT for 6–8 min, washed briefly in PBT, fixed again in MEMFA for 20 min and washed three times in PBT. Embryos were transferred into baskets, which were kept in an 8 × 8 microcentrifuge tube holder sitting inside a 10 × 10 slot plastic box filled with PBT. Baskets were built by replacing the round bottom of 2-ml microcentrifuge tubes with a Sefar Nitex mesh. This container system was used to readily process several batches of embryos at once. These baskets were maximally loaded with 40 to 50 *X. tropicalis* embryos. The microcentrifuge tube holder was used to transfer all baskets at once and to submerge embryos into subsequent buffers of the WMISH protocol. Next, the embryos were incubated in 500 μl hybridisation buffer (50% formamide, 5× SSC, 1× Denhardt's, 10 mM EDTA, 1 mg/ml torula RNA, 100 μg/ml heparin, 0.1% Tween-20 and 0.1% CHAPS) for 2 h in a hybridisation oven set to 60 °C. After this pre-hybridisation step, the embryos were transferred into 500 μl digoxigenin-labelled probe (1 ng/μl) preheated to 60 °C and further incubated overnight at 60 °C. The pre-hybridisation buffer was kept at 60 °C. The next day embryos were transferred back into the pre-hybridisation buffer and incubated at 60 °C for 10 min. Subsequently, they were washed three times in 2× SSC/0.1% Tween-20 at 60 °C for 10 min, twice in 0.2 × SSC/0.1% Tween-20 at 60 °C for 20 min and twice in 1× maleic acid buffer (MAB) at room temperature for 5 min. Next, the embryos were treated with blocking solution (2% Boehringer Mannheim blocking reagent in 1× MAB) at room temperature for 30 min, and incubated in antibody solution (10% lamb or goat serum, 2% Boehringer Mannheim blocking reagent, 1× MAB and 1:2000 Fab fragments from polyclonal anti-digoxigenin antibodies conjugated to alkaline phosphatase) at room temperature for 4 h. The embryos were then washed four times in 1× MAB for 10 min before leaving them in 1× MAB overnight at 4 °C. On the final day of the WMISH protocol, the embryos were washed another three times in 1× MAB for 20 min and equilibrated to working conditions of alkaline phosphatase (AP) for a total of 10 min by submerging embryos twice into AP buffer (50 mM MgCl₂, 100 mM NaCl, 100 mM Tris-HCl pH 9.5 and 1% Tween-20). At this stage, the embryos were transferred to 5-ml glass vials for monitoring the progression of the AP-catalysed colorimetric reaction. Any residual AP buffer was discarded before adding 700 μl staining solution (AP buffer, 340 μg/ml nitro-blue tetrazolium chloride, 175 μg/ml 5-bromo-4-chloro-3′-indolyphosphate). The colorimetric reaction was developed at room temperature in the dark. Once the staining was clear and intense enough, the colour reaction was stopped by two washes in 1× MAB. To stabilise and preserve morphological features, the embryos were fixed with Bouin's fixative without picric acid (9% formaldehyde and 5% glacial acetic acid) at room temperature for 30 min. Next, the embryos were washed twice in 70% ethanol/PBT to remove the fixative and residual chromogens. After rehydration to PBT in two steps, the embryos were treated with bleaching solution (1% H₂O₂, 0.5× SSC and 5% formamide) at 4 °C in the dark overnight. Finally, the embryos were washed twice in PBS before imaging them in PBS on a thick agarose dish by light microscopy.

**Antibody generation**. Affinity-purified rabbit polyclonal antibodies were generated by sdix (Delaware, USA) against different amino acid (aa) sequences of *X. tropicalis* Tbx6 avoiding its highly conserved DNA binding domain (aa 95–280) among T-box proteins: aa 1–94 (#4596), aa 304–403 (#5061) and aa 418–517 (#5107). Antibody #5107 did not work (data not shown). Antibodies #4596 and #5061 were suitable for immunoprecipitation and Western blotting to detect endogenous Tbx6 in both *X. tropicalis* and *X. laevis* embryos (Supplementary Fig. 2k–m). Antibody #4596 complied with ENCODE guidelines[60] and thus was suitable for ChIP (Supplementary Fig. 2n).

**Immunoprecipitation (IP)**. For each IP, 40 de-jellied embryos were collected at the developmental stage of interest. They were homogenised in 400 μl ice-cold IP buffer (50 mM Tris-HCl pH 7.5, 150 mM NaCl, 5 mM MgCl₂, 1 mM EDTA and 0.25 % Igepal CA-630) supplemented with an EDTA-free protease inhibitor cocktail (Roche, 11873580001). The extract was kept on ice for 5 min. Subsequently, the extract was mixed with 400 μl 1,1,2-trichloro-1,2,2-trifluoroethane (Sigma) by vigorously inverting the tube and cleared by centrifugation at 16,000 × g (4 °C) for 5 min. An aliquot of 20 μl supernatant was saved as input. The remaining supernatant was incubated with 0.5 μl antibody (anti-Sox3[61], anti-VegT[16] or anti-Tbx6; all rabbit polyclonal antibodies) at 4 °C for 1 h. Subsequently, each embryonic extract was supplemented with 20 μl washed magnetic Dynabeads M280 conjugated to sheep anti-rabbit IgG (Thermo Fisher Scientific, 11205D) and kept on a tube rotator at 4 °C for another 2 h. Next, the beads were washed three times with IP buffer. The immunoprecipitate was eluted off the beads with 20 μl 1× SDS loading buffer (50 mM Tris-HCl pH 6.8, 2% SDS, 6% glycerol and 5% β-mercaptoethanol). Western blotting was carried out with TrueBlot (Rockland Immunochemicals) and normal horseradish peroxidase (HRP)-conjugated secondary antibodies for IP and input samples, respectively.

**Western blotting**. Protein samples were denatured in 1× (final concentration) SDS loading buffer at 70 °C for 5 min. Denatured samples were run alongside a standard protein ladder into pre-cast gels (Any kD Mini-PROTEAN TGX, BioRad) at constant 200 V for 1 h. Size-separated proteins were immediately transferred onto hydrophobic Immobilon-P PVDF transfer membranes (Millipore) at constant 100 V for 30 min using standard protein electrophoresis equipment. The membranes were blocked with 5% milk powder in PBS/0.1% Tween-20 (MPBTw) at room temperature for 30 min. Next, the membranes were incubated at room temperature for 1 h with the primary antibodies diluted in MPBTw: 1:5,000 of mouse monoclonal anti-α-tubulin (Sigma, T5168), 1:2000 rabbit polyclonal anti-Sox3[61], 1:2000 rabbit polyclonal anti-VegT[16] or 1:2,000 rabbit polyclonal anti-Tbx6 (#5061). The membranes were washed three times with PBST for 10 min before applying 1:1,000 TrueBlot HRP-conjugated anti-rabbit IgG (Rockland Immunochemicals, 18-8816-31) or 1:2000 normal goat HRP-conjugated anti-mouse IgG (Thermo Fisher Scientific, 31430) in MPBTw at room temperature. After 1 h, the membranes were washed three times in PBTw for 10 min and once in PBS for 5 min. Finally, the membranes were treated with UptiLight US HRP WB reagent (interchim, 58372) for 1 min in the dark. HRP-catalysed chemiluminescence was detected on the ChemiDoc XRS+ system (BioRad). Uncropped Western blots are shown in Supplementary Fig. 18.

**Whole-mount immunohistochemistry (WMIHC)**. Embryos were fixed with MEMFA (0.1 M MOPS, 2 mM EGTA, 1 mM MgSO$_4$ and 3.7% formaldehyde) in capped 5-ml glass vials at room temperature for 2 h, dehydrated and stored in absolute ethanol at $-20\,°C$ for ≥24 h. For sagittal sections, fixation was stopped after 30 min. Embryos were placed on a piece of flattened Blu-Tack (Bostik) in a large Petri dish filled with PBS to bisect them with a scalpel. Following a three-step rehydration (50%, 75% and 100% PBS) the embryos were bleached with bleaching solution (1% H$_2$O$_2$, 0.5× SSC and 5% formamide) on a light box at room temperature for 2 h or in the dark at 4 °C overnight. Bleached embryos were washed three times with PBS/0.3% Triton X-100 (PBT) and pre-incubated in blocking solution (PBS, 20% goat or donkey serum, 2% BSA and 0.1% Triton X-100) at 4 °C for 6 h. The embryos were then transferred to 2-ml round-bottom microcentrifuge tubes before discarding all remaining blocking solution. The embryos were incubated at 4 °C for 1–3 days in 50 µl blocking solution containing the primary antibody at the following dilutions: 1:1,000 rabbit polyclonal anti-Sox3[61], 1:500 rabbit polyclonal anti phospho-Smad1/5/8 (Cell Signaling, 9511) or 1:500 goat polyclonal Smad2/3 (R&D Systems, AF3797). Afterwards, the embryos were transferred back to capped 5-ml glass vials, washed three times with RIPA buffer (50 mM HEPES pH 7.5, 0.5 M LiCl, 1 mM EDTA, 1% Igepal CA-630 and 0.7% sodium deoxycholate) for 1 h and once with PBT for 5 min. Next, the embryos were incubated with the secondary HRP-conjugated antibody diluted in blocking solution at 4 °C overnight: 1:400 goat anti-rabbit IgG-HRP (Thermo Fisher Scientific, G-21234) or 1:200 donkey anti-goat IgG-HRP (Santa Cruz Biotechnology, sc-2020). The washes of the previous day were repeated before pre-incubating the embryos in 400 µl inactive 3,3'-diaminobenzidine (DAB) tetrahydrochloride with cobalt (Sigma, D0426). After 1–2 min, the inactive solution was replaced with the H$_2$O$_2$-activated DAB solution. The HRP reaction was stopped after 40 s by washing the embryos several times with PBT. In Figs. 2c and 5a, d, embryos were dehydrated in three steps to absolute methanol and cleared with Murray's clear (2:1 benzyl benzoate/benzyl alcohol) on a glass depression slide.

**Deep sequencing and quality filter**. All deep sequencing libraries were quality controlled: The DNA yield and fragment size distribution were determined by fluorometry and chip-based capillary or polyacrylamide gel electrophoresis, respectively. Libraries were sequenced on the Illumina platforms GAIIx or HiSeq by the Advanced Sequencing Facility of the Francis Crick Institute to produce single or paired-end reads of at least 40 bases. Next-generation capture-C libraries were sequenced on MiSeq with a read length of 150 bases. Sequencing samples and read alignment results are summarised in Supplementary Data 1. The metrics of paired-end alignments such as insert size mean and standard deviation were determined by Picard (CollectInsertSizeMetrics) from the Broad Institute (USA).

**Chromatin immunoprecipitation (ChIP)**. ChIP was carried out as detailed previously[62]. Briefly, de-jellied *X. tropicalis* embryos were treated with 1% formaldehyde (Sigma, F8775) in 1% MMR at room temperature for 15–45 min to cross-link chromatin proteins to nearby genomic DNA. Duration of fixation was determined empirically and depended mainly on the developmental stage and antibody epitopes[62]. Fixation was terminated by rinsing embryos three times with ice-cold 1% MMR. Where required, post-fixation embryos were dissected to select specific anatomical regions in ice-cold 1% MMR. Fixed embryos were homogenised in CEWB1 (10 mM Tris-HCl pH 8.0, 150 mM NaCl, 1 mM EDTA, 1% Igepal CA-630, 0.25% sodium deoxycholate and 0.1% SDS) supplemented with 0.5 mM DTT, protease inhibitors and, if using phospho-specific antibodies, phosphatase blockers (0.5 mM orthovanadate and 2.5 mM NaF). To solubilise yolk platelets and separate them from the nuclei, the homogenate was left on ice for 5 min and then centrifuged at $1000 \times g$ (4 °C) for 5 min. Homogenisation and centrifugation was repeated once before resuspending the nuclei containing pellet in 1–3 ml CEWB1. Nuclear chromatin was solubilised and fragmented by isothermal focused or microtip-mediated sonication. The solution of fragmented chromatin was cleared by centrifuging at $16,000 \times g$ (4 °C) for 5 min. Where required, ~1% of the cleared chromatin extract was set aside for the input sample (negative control). ChIP-grade antibodies were used to recognise specific chromatin features and to enrich these by coupling the antibody-chromatin complex to protein G magnetic beads (Thermo Fisher Scientific, 10003D) and extensive washing. These steps were carried out at 4 °C. The beads were washed twice in CEWB1, twice in WB2 (10 mM Tris-HCl pH 8.0, 500 mM NaCl, 1 mM EDTA, 1% Igepal CA-630, 0.25% sodium deoxycholate and 0.1% SDS), twice in WB3 (10 mM Tris-HCl pH 8.0, 250 mM LiCl, 1 mM EDTA, 1% Igepal CA-630 and 1% sodium deoxycholate) and once in TEN (10 mM Tris-HCl pH 8.0, 150 mM NaCl and 1 mM EDTA). ChIP was eluted off the beads twice with 100 µl SDS elution buffer (50 mM Tris-HCl pH 8.0, 1 mM EDTA and 1% SDS) at 65 °C. ChIP eluates were pooled before reversing DNA-protein cross-links. Input (filled up to 200 µl with SDS elution buffer) and ChIP samples were supplemented with 10 µl 5 M NaCl and incubated at 65 °C for 6–16 h. Samples were treated with proteinase K (Thermo Fisher Scientific, AM2548) and RNase A (Thermo Fisher Scientific, 12091021) to remove any proteins and RNA from the co-immunoprecipitated DNA fragments. The DNA was purified with phenol:chloroform:isoamyl alcohol (25:24:1, pH 7.9) using 1.5-ml Phase Lock Gel Heavy microcentrifuge tubes (VWR) for phase separation and precipitated with 1/70 volume of 5 M NaCl, 2 volumes of absolute ethanol and 15 µg GlycoBlue (Thermo Fisher Scientific, AM9516). After centrifugation, the DNA pellet was air-dried and dissolved in 11 µl elution buffer (10 mM Tris-HCl pH 8.5). The DNA concentration was determined on a fluorometer using high-sensitivity reagents for double-stranded DNA (10 pg/µl to 100 ng/µl).

The following antibodies were used for ChIP: rabbit polyclonal anti-β-catenin (Santa Cruz Biotechnology, H-102), rabbit polyclonal anti-H3K4me1 (Abcam, ab8895), rabbit polyclonal anti phospho-Smad1/5/8 (Cell Signaling, 9511), goat polyclonal anti-Smad2/3 (R&D Systems, AF3797), rabbit polyclonal anti-Sox3[61], rabbit polyclonal anti-Tbx6 (#4596) and rabbit polyclonal anti-VegT[16].

**Quantitative PCR (qPCR)**. DNA was amplified in technical duplicates with SYBR Green I master mix (Roche, 04707516001) on a Light Cycler 480 II (Roche) cycling 55-times between 94 °C, 60 °C and 72 °C with each temperature step running for 10 s and switching at $+4.8\,°C/s$ and $-2.5\,°C/s$. At the end qPCR reactions were heated from 65 °C to 97 °C with a gradual increase of 0.11 °C/s (melting curve) to ensure only fluorescence was collected from one specific amplicon. ChIP-qPCR results were based on absolute quantification using an eight-point standard curve of three-fold dilutions of ~1% input DNA (Supplementary Fig. 2d, g, i, j, n). RT-qPCR results were normalised to the housekeeping gene *odc1* (Figs. 4h and 7d and Supplementary Figs. 10e) or *rpl8* (Supplementary Fig. 8b), and scaled relative to control embryos using the $2^{-\Delta\Delta Ct}$ method[63]. The threshold cycle ($C_t$) was derived from the maximum acceleration of SYBR fluorescence (second derivative maximum method).

**ChIP-Seq library preparation**. 0.5–10 ng ChIP DNA or 5 ng input DNA were used to prepare single (only Tbx6 ChIP-Seq) or indexed paired-end libraries using the KAPA Hyper Prep Kit[62,64–66].

**Post-sequencing analysis of ChIP-Seq**. Single reads of maximal 50 bases were processed using trim_galore v0.4.2 (Babraham Institute, UK) to trim off low-quality bases (default Phred score of 20, i.e. error probability was 0.01) and adapter contamination from the 3' end. Processed reads were aligned to the *X. tropicalis* genome assembly v7.1 (and v9.1 for some ChIP-Seq data) running Bowtie2 v2.2.9[67] with default settings (Supplementary Data 1). Alignments were converted to the HOMER's tag density format[68] with redundant reads being removed (make-TagDirectory -single -tbp 1 -unique -mapq 10 -fragLength 175 -totalReads all). Only uniquely aligned reads (i.e., MAPQ ≥10) were processed. We pooled all input alignments from various developmental stages. This created a comprehensive mappability profile that covered ~400 million unique base pair positions. All chromatin profiles were position-adjusted and normalised to the effective total of 1 million aligned reads including multimappers (counts per million aligned reads, CPM). Agreeing biological replicates according to ENCODE guidelines[60] were subsequently merged. For stage 10$^+$ (early gastrula stage) β-catenin and Smad2 ChIP-Seq, external datasets[26,69,70] were used as biological replicates. HOMER's peak caller was used to identify transcription factor binding sites by virtue of ChIP-enriched read alignments (hereafter called peaks): findpeaks -style factor -minDist 175 -fragLength 175 -inputFragLength 175 -fdr 0.001 -gsize 1.435e9 -F 3 -L 1 -C 0.97. This means that both ChIP and input alignments were extended 3' to 175 bp for the detection of significant (0.1% FDR) peaks being separated by ≥175 bp. The effective size of the *X. tropicalis* genome assembly v7.1 was set to 1.435 billion bp, an estimate obtained from the mappability profile. These peaks showed equal or higher tag density than the surrounding 10 kb, ≥3-fold more tags than the input and ≥0.97 unique tag positions relative to the expected number of tags. To detect focal RNAPII recruitment to pCRMs and avoid calling peaks within broad regions of RNAPII reflecting transcript elongation, the threshold of focal ratio and local enrichment within 10 kb was elevated to 0.6 and 3 (-L 3), respectively. To further eliminate any false positive peaks, we removed any peaks with <0.5 (TFs including signal mediators) or <1 (RNAPII) CPM and those falling into blacklisted regions showing equivocal mappability due to genome assembly errors, gaps or simple/tandem repeats. Regions of equivocal mappability were identified by a two-fold lower (poor) or three-fold higher (excessive) read coverage than the average detected in 400-bp windows sliding at 200-bp intervals through normalised ChIP input and DNase-digested naked genomic DNA. All identified regions ≤800 bp apart were subsequently merged. Gap coordinates were obtained from the Francis Crick mirror site of the UCSC genome browser (http://genomes.crick.ac.uk). Simple repeats were masked with RepeatMasker v4.0.6[71] using the crossmatch search engine v1.090518 and the following settings: RepeatMasker -species "xenopus silurana tropicalis" -s -xsmall. Tandem repeats were masked with Jim Kent's trfBig wrapper script of the Tandem Repeat Finder v4.09[72] using the following settings: weight for match, 2; weight for mismatch, 7; delta, 7; matching probability, 80; indel probability, 10; minimal alignment score, 50; maximum period size, 2,000; and longest tandem repeat array (-l), 2 [million bp]. The multi-genome sequence conservation track (phastCons) for *X. tropicalis* genome assembly v9.1 was obtained from Xenbase[73]. The following eleven vertebrate species were used to evaluate sequence similarity: *X. tropicalis, X. laevis, Nanorana parkeri* (High Himalaya Frog), *Fugu rubripes* (Japanese Pufferfish), *Chrysemys picta* (Painted Turtle), *Gallus gallus* (Chicken), *Anolis carolinensis* (Green Anole lizard), *Monodelphis domestica* (Grey Short-tailed Opossum), *Canis lupus familiaris* (dog),

mouse and human. pCRMs with a phastCons average ≥0.4 were considered 'conserved'.

**Poly(A) RNA-Seq.** Libraries were made from ~1 µg total RNA by following the low-sample protocol of the TruSeq RNA sample preparation guide version 2 (Illumina) with a few modifications. First, 1 µl cDNA purified after second strand synthesis was quantified on a fluorometer using high-sensitivity reagents for double-stranded DNA (10 pg/µl to 100 ng/µl). By this stage, the yield was ~10 ng. Second, the numbers of PCR cycles were adjusted to the detected yield of cDNA to avoid products of over-amplification such as chimera fragments: 7 (~20 ng), 8 (~10 ng), 9 (~5 ng) and 12 (~1 ng).

**RNA-Seq read alignment.** Paired-end reads were aligned to the *X. tropicalis* genome assembly v7.1 using STAR v2.5.3a[74] with default settings (Supplementary Data 1) and a revised version of gene models v7.2[56] to improve mapping accuracy across splice junctions. The alignments were sorted by read name using the sort function of samtools v1.3.1[75]. Exon and intron counts (-t 'exon;intron') were extracted from unstranded (-s 0) alignment files using VERSE v0.1.5[76] in featureCounts (default) mode (-z 0). Intron coordinates were adjusted to exclude any overlap with exon annotation. For visualisation, genomic BAM files of biological replicates were merged using samtools v1.3.1 and converted to the bigWig format. These genome tracks were normalised to the wigsum of 1 billion excluding any reads with mapping quality <10 using the python script bam2wig.py from RSeQC v2.6.4[77].

**Differential gene expression analysis.** Differential expression analysis was performed with both raw exon and intron counts excluding those belonging to ribosomal and mitochondrial RNA using the Bioconductor/R package DESeq2 v1.16.1[78]. In an effort to find genes with consistent fold changes over time, p-values were generated according to a likelihood ratio test reflecting the probability of rejecting the reduced (~developmental stage) over the full (~developmental stage + condition) model. Resulting *p*-values were adjusted to obtain false discovery rates (FDR) according to the Benjamini-Hochburg procedure with thresholds on Cook's distances and independent filtering being switched off. Equally, regional expression datasets[79] without time series were subjected to likelihood ratio tests with reduced (~1) and full (~condition) models for statistical analysis. Fold changes of intronic and exonic transcript levels were calculated for each developmental stage and condition using the mean of DESeq2-normalised read counts from biological replicates. Both intronic and exonic datasets were filtered for ≥10 DESeq2-normalised read counts detected at least at one developmental stage in all uninjected or DMSO-treated samples. Gene-specific fold changes were removed at developmental stages that yielded <10 normalised read counts in corresponding control samples. Next, the means of intronic and exonic fold changes were calculated across developmental stages. The whole dataset was confined to 3687 genes for which at least 50% reductions (FDR ≤10%) in exonic (default) or intronic counts could be detected in α-amanitin-injected embryos. Regional expression was based on exonic read counts by default unless the intronic fold changes were significantly (FDR≤10%) larger than the exonic fold changes (Supplementary Data 9).

**Analysing ribosome footprinting and mass-spectrometry data.** The ribosome footprinting reads were trimmed 5′ by 8 bases and 3′ by as many bases overlapping with the adapter sequence 5′-TCGTATGCCGTCTTCTGCTTG-3′ from the 5′ end. All trimmed reads between 27 and 32 bases in length were aligned first to ribosomal RNA as listed in the SILVA rRNA database[80] using Bowtie v1.0.1[81] with the following parameters:--seedlen 25 (seed length) --seedmms 1 (number of mismatches allowed in the seed) --un (unaligned reads were reported). All non-aligned (rRNA-depleted) reads were mapped to the gene model 6.0 of *X. laevis* using Tophat v2.0.10[82] --no-novel-juncs (spliced reads must match splice junctions of gene model 6.0) --no-novel-indels (indel detetion disabled) --segment-length 15 (minimal length of read fragment to be aligned) –GTF v6.0.gff3 --prefilter-multi-hits (reads aligned first to whole genome to exclude reads aligning >10 times) --max-multihits 10. Alignment files were converted to the HOMER's tag density format[68] before retrieving reads for each CDS per kilobase and one million mapped reads (rpkm) using HOMER's perl script analyzeRNA.pl. This read count table was merged with a published list of estimated protein concentrations (nM) in the *X. laevis* egg using mass-spectrometry[13] (Supplementary Data 4). Chromatin-associated proteins regulating RNAPII-mediated transcription were filtered based on human (version 2.0, 09/2014) and Xenbase-released gene ontology (GO) associations. The genes associated with any of the following GO terms were verified with UniProt (UniProt Consortium, 2015) whether chromatin binding is supported by functional evidence: chromatin (GO:0000785), DNA binding (GO:0003677), DNA-templated regulation of transcription (GO:0006355), sequence-specific DNA binding and TF activity (GO:0003700) and DNA replication (GO:0006260). We separated specific TFs from all other chromatin binding proteins to form three categories: (1) sequence-specific DNA binding factors, (2) other RNAPII-regulating factors and (3) remaining genes, whereby genes associated with DNA repair (GO:0006281) and RNAPIII (GO:0006383) were moved to the third category.

**DNase-Seq based on solid phase immobilisation (SPI).** In our hands, the high yolk content in vegetal blastomeres made it impossible to employ ATAC-Seq[12] on whole *Xenopus* embryos before the onset of gastrulation. Therefore, DNase-Seq was adapted to early *Xenopus* embryos with an alternative approach to select small DNase-digested DNA fragments. Ultracentrifugation- or gel electrophoresis-mediated size selection[83,84] was replaced by two rounds of SPI to remove long inaccessible DNA from short accessible DNA. Wide-bore pipette tips were used for the resuspensions and the transfers of samples from the second homogenisation step until after SPI to minimise fragmentation of high-molecular DNA. Approximately 250 de-jellied mid-blastula embryos were collected in 2-ml round-bottom microcentrifuge tubes and homogenised in 2 ml ice-cold LB-DNase buffer (15 mM Tris-HCl pH 8.0, 15 mM NaCl, 60 mM KCl, 1 mM EDTA, 0.5 mM EGTA and 0.5 mM spermidine) supplemented with 0.05% Igepal CA-630. This lysate was left on ice for 3 min before centrifuging at 1000 × g (4 °C) for 2 min. The supernatant was discarded without disrupting the pellet. The pellet was gently resuspended in 2 ml ice-cold LB-DNase buffer (without Igepal CA-630) before centrifuging again at 1000 × g (4 °C) for 2 min. After discarding the supernatant, the pellet was resuspended at room temperature in 600 µl LB-DNase buffer (without Igepal CA-630) with 6 mM CaCl₂. The sample was distributed equally to two 1.5-ml microcentrifuge tubes, one for probing chromatin accessibility with DNase and the other one for creating a reference profile of DNase-digested naked genomic DNA. Approximately 0.1 U recombinant DNase I (Roche, 04716728001) was added to one aliquot. Both samples were incubated at 37 °C for 8 min before adding 300 µl STOP buffer (0.1% SDS, 100 mM NaCl, 100 mM EDTA and 50 mM Tris-HCl pH 8.0) including 80 µg RNase A (Thermo Fisher Scientific, 12091021), 333 nM spermine and 1 µM spermidine. The tubes were inverted gently to mix samples before incubating them at 55 °C for 15 min. Next, 200 µg proteinase K (Thermo Fisher Scientific, AM2548) were added and the tubes were inverted gently to mix the samples again. After 2 h at 55 °C, the digests were transferred to pre-spun 1.5-ml Phase Lock Gel Heavy microcentrifuge tubes (VWR). 600 µl phenol:chloroform:isoamylalcohol (25:24:1, pH 7.9) were added to the digests. The tubes were shaken gently and then centrifuged at 1500 × g (room temperature) for 4 min. The top phase was transferred to a fresh 2-ml microcentrifuge tube and mixed with 60 µl 3 M sodium acetate (pH 5.2) and 1.2 ml absolute ethanol to precipitate the genomic DNA. The precipitation was stored at −20 °C overnight and centrifuged at 16,000 × g (4 °C) for 30 min. The supernatant was discarded and the DNA pellet washed by adding 500 µl ice-cold 80% ethanol and centrifuging at 16,000 × g (4 °C) for 3 min. The ethanol was discarded and the DNA pellet was air-dried at room temperature for 10 min. After that, the DNA pellet was left on ice to dissolve in 27 µl elution buffer (10 mM Tris-HCl pH 8.5) for 20 min. To remove any residual RNA, 10 µg RNase A were added to the DNA. 1 µl was used to determine the DNA fragment size distribution by gel electrophoresis. On a 0.6% TAE agarose gel, a smear of very high molecular DNA was visible as expected from previous DNase experiments[83,85]. Genomic DNA to the amount of 20 untreated mid-blastula embryos was digested with 0.3 U DNase I at 37 °C for 5 min following the same steps as described above, which generated a low-molecular smear of DNA fragments. Next, 70 µl AMPure XP beads (Beckman Coulter, A63880) per DNase sample were transferred to a pre-spun 1.5-ml Phase Lock Gel Heavy tube. 162 µl elution buffer and 300 µl phenol:chloroform:isoamylalcohol (25:24:1, pH 7.9) were equilibrated to room temperature for 10 min. 22.5 µl AMPure XP beads (0.9× of the volume) were added to the DNA sample without pipetting up and down achieving a final polyethylene glycol concentration of ~9.5%. After 3 min, by which time high-molecular DNA causes beads to coalesce, the tubes were clipped into a magnetic stand for microcentrifuge tubes. After 3 min or until the beads were separated from the supernatant, the latter was transferred to a 96-well microplate (350-µl round wells with V-shaped bases). 47.5 µl elution buffer and 43 µl AMPure XP beads were added sequentially and mixed gently by slowly pipetting up and down. After 3 min, the plate was transferred to a magnetic stand for 96-well plates. Once the beads have completely separated from the suspension, the supernatant was transferred to a pre-spun 1.5-ml Phase Lock Gel Heavy tube. 162 µl elution buffer and 300 µl phenol:chloroform:isoamylalcohol (25:24:1, pH 7.9) were added. The tubes were shaken gently and then centrifuged at 1500 × g (room temperature) for 4 min. The top phase was transferred to a 1.5-ml low-retention microcentrifuge tube and mixed with 30 µl 3 M sodium acetate (pH 5.2) and 900 µl absolute ethanol to precipitate the DNA as outlined above. The DNA pellet was dissolved in 12 µl elution buffer. The concentrations of the DNA samples were determined on a fluorometer using high sensitivity reagents for dsDNA (10 pg/µl to 100 ng/µl). Libraries were generated using the KAPA Hyper Prep Kit (Roche) according to the manufacturer's instructions except that all cleaning steps were executed with 0.2× more AMPure XP beads.

**Post-sequencing analysis of DNase-Seq.** Single and paired-end reads of maximal 50 bases were processed using trim_galore v0.4.2 from the Babraham Institute (UK) to trim off low-quality bases (default Phred score of 20, i.e., error probability was 0.01) and adapter contamination from the 3′ end. Processed reads were aligned to the *X. tropicalis* genome v7.1 and v9.1 using Bowtie2 v2.2.9[67] with default settings apart from -X (fragment length), which was reduced to 250 bp for paired-end reads. Alignments were sorted by genomic coordinates and only those with a quality score of ≥10 were retained using samtools v1.3.1[75]. Duplicates were removed using Picard (MarkDuplicates) from the Broad Institute (USA). Paired-end alignments were dissociated using hex flags (-f 0 × 40 or 0 × 80) of samtools view. Single alignments were converted to HOMER's tag density format[68]

(makeTagDirectory -single -unique -fragLength 100 -totalReads all). DNase hypersensitive sites were identified using the following or otherwise default parameters of HOMER's peak calling: findpeaks -style factor -minDist 100 -fragLength 100 -inputFragLength 100 -fdr 0.001 -gsize 1.435e9 -F 3 -L 1 -C 0.97. This means that alignments of DNase-digested chromatin fragments (or naked genomic DNA fragments considered here as 'input') were extended 3′ to 100 bp from the DNase cleavage site to detect significant (0.1% FDR) DNase-enriched read alignments (hereafter called peaks) being separated by ≥100 bp. The effective size of the *X. tropicalis* genome assembly v7.1 was set to 1.435 billion bp, an estimate obtained from the mappability of ChIP input reads. These peaks showed equal or higher tag density than the surrounding 10 kb, at least three-fold more tags than the input and ≥0.97 unique tag positions relative to the expected number of tags. Peaks falling into blacklisted regions (see Post-sequencing analysis of ChIP-Seq) were removed. The profiles of DNase-probed chromatin accessibility were position-adjusted and normalised to the effective total of 1 million aligned reads including multimappers. Genomic coordinates of accessible pCRMs are listed in Supplementary Data 3 and 10 (mPouV/Sox3 LOF experiment).

**Next-generation capture-C**. About 500 mid-blastula embryos per condition were collected in 9-ml capped glass vials and briefly washed once with 1% MMR. The embryos were then fixed with 1% formaldehyde (Sigma, F8775) in 9 ml 1% MMR at room temperature for 40 min. The fixation reaction was terminated by briefly rinsing the embryos three times with ice-cold 1% MMR. The embryos were aliquoted into 2-ml round-bottom microcentrifuge tubes in batches of ~250 embryos (filling the volume of ~250 μl water). The supernatant was removed before equilibrating embryos in 250 μl ice-cold HEG buffer (50 mM HEPES pH 7.5, 1 mM EDTA and 20% glycerol). Once the embryos settled to the bottom of the tube as much liquid as possible was discarded. The aliquots were snap-frozen in liquid nitrogen and stored at −80 °C.

For each chromatin conformation capture (3C) experiment, 10 ml chromatin extraction buffer CEB-3C (10 mM Tris-HCl pH 8.0, 150 mM NaCl, 1 mM EDTA, 1% Igepal CA-630, 0.25% sodium deoxycholate and 0.2% SDS) were supplemented with a mini protease inhibitor tablet (Roche, 11873580001) and 0.05 mM DTT (CEB-3C*). Both aliquots of ~250 embryos of each condition were homogenised in 2 ml ice-cold CEB-3C*. The tubes were kept on ice for 5 min before centrifuging at 1000 × *g* (4 °C) for 2 min. The supernatants were discarded and the pellet resuspended in 0.5 ml ice-cold CEB-3C*. The resuspensions of each condition were pooled and then equally divided to two 50-ml conical tubes and each filled up with CEB-3C (without protease inhibitors and DTT) to 50 ml. The embryonic extracts were incubated at 37 °C for 1 h in a hybridisation oven with the tubes rotating inside a hybridisation bottle. The tubes were then centrifuged at 1000 × *g* (room temperature) for 5 min. The supernatants were discarded and the pellets resuspended in 50 ml double-distilled water. The tubes were centrifuged again at 1000 × *g* (room temperature) for 5 min. The supernatants were discarded and each pellet was resuspended in 300 μl double-distilled water. The resuspensions containing cross-linked nuclei of each condition were pooled in a 1.5-ml microcentrifuge tube. The samples were then digested with 400 U *Dpn*II (NEB, R0543) in a total volume of 800 μl containing 1× *Dpn*II reaction buffer. The digest was incubated overnight in a thermomixer set to 37 °C and 1400 rpm. Of note, any residual yolk platelets in the solution did not interfere with the restriction digest. The next day, the digest was supplemented with 200 U *Dpn*II and incubated at 37 °C and 1,400 rpm for another 6–8 h. After that, *Dpn*II was heat-inactivated at 65 °C for 20 min.

To verify the degree of chromatin digestion, an aliquot of 50 μl from each condition was transferred into separate 1.5-ml microcentrifuge tubes and processed as follows: First, any remaining RNA was degraded by incubating the aliquot with 20 μg RNase A (Thermo Fisher Scientific, 12091021) at 37 °C for 30 min. Second, proteins were digested by incubating the aliquot mixed with 50 μl SDS elution buffer (50 mM Tris pH 8.0, 1 mM EDTA and 1% SDS) and 20 μg proteinase K (Thermo Fisher Scientific, AM2548) at 65 °C in a hybridisation oven overnight. Third, the DNA was purified and ethanol precipitated: 300 μl TE pH 8.0 were added to the digest before transferring it to pre-spun 1.5-ml Phase Lock Gel Heavy microcentrifuge tubes (VWR). The diluted digests were mixed vigorously with 400 μl (1 volume) phenol:chloroform:isoamylalcohol (25:24:1, pH 7.9) before centrifuging at 16,000 × *g* (room temperature) for 1 min. The top (aqueous) phases were transferred into 1.5-ml microcentrifuge tubes and mixed with 40 μl sodium acetate (pH 5.2) and 800 μl absolute ethanol and 15 μg GlycoBlue (Thermo Fisher Scientific, AM9516). The DNA precipitations were kept at −80 °C for 30 min before centrifuging at 16,000 × *g* (4 °C) for 30 min. The supernatants were discarded without disturbing the DNA pellets which were subsequently washed once with 400 μl ice-cold 80% ethanol. The tubes were centrifuged at 16,000 × *g* (4 °C) for 30 min before discarding the supernatants. The DNA pellets were air-dried for 10 min and resuspended in 10 μl elution buffer (10 mM Tris-HCl pH 8.5). The DNA concentrations were determined with a fluorometer and the DNA fragment distributions were visualised on a 0.6% TAE agarose gel. The digestion of cross-linked chromatin with *Dpn*II caused extensive fragmentation of the genomic DNA such that no high-molecular DNA fragment bands (>10 kb) were visible.

The chromatin digests were processed for proximity ligation by adding 240 U T4 DNA ligase (Thermo Fisher Scientific, EL0013) in a total volume of 1.2 ml containing 1× T4 DNA ligase buffer including 5 mM ATP. The ligation reactions

were incubated at 16 °C and 1400 rpm for ≥16 h before centrifuging at 16,000 × *g* (4 °C) for 1 min. The supernatants were discarded. 200 μl elution buffer and 40 μg RNase A were added to the pellets to degrade any residual RNA at 37 °C and 1400 rpm for 30 min. Proteins were digested by adding 200 μl SDS elution buffer and 160 μg proteinase K and incubating at 65 °C in a hybridisation oven overnight. The digest was transferred to pre-spun 1.5-ml Phase Lock Gel Heavy tubes and processed as described in the previous paragraph to purify and precipitate ligated genomic DNA. The air-dried DNA pellet was dissolved in 20 μl elution buffer and quality controlled as above. Proximity ligation of *Dpn*II-digested chromatin fragments massively increased the size of DNA fragments to the range of high molecular weight (>10 kb) representing the 3C library. The concentration was measured on a fluorometer with broad range concentration (5–500 ng/μl) reagents.

About 10–15 μg of the 3 C libraries were diluted with elution buffer to the total volume of 130 μl and transferred to a designated glass vial (microTUBE) for isothermal focused sonication (Covaris). The following settings of the focused ultrasonicator Covaris S220 were used to shear the libraries to an average DNA fragment length of ~200 bp: duty cycle, 10%; intensity, 5; cycles per burst, 200; duration, 60 s in frequency sweeping mode. Sonication was run with 6 cycles in batch mode. The degree of DNA fragmentation was verified by gel electrophoresis and the DNA concentration was measured on a fluorometer with broad range concentration reagents.

Approximately 2–4 μg of the DNA fragments were converted into next-generation paired-end libraries using the enzymes of the KAPA Hyper Prep Kit (Kapa Biosystems, KR0961) and TruSeq adaptors (Illumina). The end-repair and A-tailing reactions were set up according to the manufacturer's instructions and incubated at 20 °C for 1 h followed by 30 min at 65 °C. End-repaired and A-tailed DNA fragments were ligated to 150 pmol TruSeq adaptors of index 6 and 12, respectively, at 20 °C for 1 h. The DNA was purified using 0.8× volume of AMPure XP beads (Beckman Coulter, A63880) and amplified in a total volume of 100 μl using 50 pmol of each TruSeq PCR primer, KAPA HiFi HotStart ReadyMix (Kapa Biosystems, KK2602) and the following PCR conditions: 98 °C for 45 s followed by 3 cycles (98 °C for 15 s, 98 °C for 30 s and 72 °C for 30 s) and a final elongation step of 1 min at 72 °C. The lid temperature was set to 105 °C. The PCR reactions were cleaned up with 100 μl AMPure XP beads and eluted in 14 μl elution buffer. The integrity of the library DNA was verified on a chip-based capillary electrophoresis system, an 8% TBE polyacrylamide gel or a 2% TAE agarose gel. DNA concentration was measured on a fluorometer with broad range concentration reagents. The DNA yield was 2–4 μg.

To capture the genomic regions of interest (viewpoints), 5′-biotinylated oligonucleotides of 120 bases (hereafter called probes) (Supplementary Data 10) were designed for each viewpoint as follows: The viewpoints were *Dpn*II fragments of 300–3300 bp which were overlapping gene promoters or located <1 kb from them. Each probe matched the terminal sequence of the same DNA fragment strand including the *Dpn*II restriction site GATC. The probe sequence was examined using BLAT to check whether it is unique and did not partially match any other genomic regions. Furthermore, CENSOR v4.2.29[86] and RepeatMasker v4.0.6[71] programs were run to discard any probes that contained repeats. Because of these design restrictions, six of the selected 30 gene promoters were captured with only one probe. All the probes were purchased as desalted oligomers (4 nmol) from IDT (ultramer technology) and reconstituted in molecular-grade water to 10 μM. The probes were mixed in equimolar quantities and diluted to 2.9 nM such that 4.5 μl contained 13 fmol of oligomers.

The capture was performed with the SeqCap Hybridisation and Wash Kit (Roche, 05634253001) and SeqCap HE-Oligo Kit A (Roche, 06777287001) as follows: Exactly 1 μg of each TruSeq library with index 6 and 12 was mixed with 1 nmol TruSeq universal blocking oligonucleotide and 0.5 nmol blocking oligonucleotides specific to TruSeq index 6 and 12 in a 1.5-ml low-retention microcentrifuge tube. Of note, Cot-1 DNA commonly used to mask repetitive DNA proved to be unnecessary here in reducing non-specific hybridisation. The mixture of library and blocking oligonucleotides was dried in a vacuum centrifuge before adding 7.5 μl 2× hybridisation buffer (vial 5) and 3 μl hybridisation component A (vial 6). This mixture was vortexed for 10 s, centrifuged at 16,000 × *g* for 10 s and incubated at 95 °C for 10 min. In the meantime, 4.5 μl (13 fmol) of the equimolar probe mixture were transferred to a PCR tube and incubated at 47 °C in a PCR machine with the volume and the lid temperature set to 15 μl and 57 °C, respectively. Upon denaturation, the libraries and blocking oligonucleotides were added to the probes without removing either tubes from the heating block or the PCR machine. The hybridisation mixture was incubated at 47 °C for 64–72 h. The wash buffers were prepared according to the manufacturer's instructions to make 1× working solutions for one single capture experiment. The stringent wash buffer and wash buffer I were pre-warmed to 47 °C and 50 μl M270 streptavidin-conjugated magnetic beads (Thermo Fisher Scientific, 65305) were transferred into a 1.5-ml low-retention microcentrifuge tube to let them equilibrate to room temperature for 10 min. The beads were washed twice with 200 μl bead wash buffer. Immediately after the final wash, the hybridisation mixture was directly added to the beads and vortexed for 10 s. The sample was incubated for 45 min in a thermomixer set to 47 °C and 1100 rpm. The beads were washed by adding 100 μl pre-warmed wash buffer I and vortexing for 10 s. The tube was placed into a magnetic rack. Once the liquid was clear, the supernatant was discarded, and 200 μl pre-warmed stringent wash buffer were added to the beads. In an effort to avoid any temperature drop, it is important to work quickly according to the

manufacturer's instructions. The beads in stringent wash buffer were incubated in a thermomixer set to 47 °C and 1100 rpm for 5 min. The wash with pre-warmed stringent wash buffer was repeated once. Next, 200 µl pre-warmed wash buffer I were added to the beads and vortexed at 1400 rpm (room temperature) for 2 min. After removing the respective wash buffer, beads were vortexed at 1400 rpm in 200 µl wash buffer II for 1 min and 200 µl wash buffer III for 30 s. After the final wash, as much liquid as possible was removed and the beads were resuspended in 40 µl molecular-grade water.

The captured DNA was directly amplified from the beads using KAPA HiFi HotStart master mix and 25 pmol of each TruSeq PCR primer in two separate 50-µl PCR reactions. The PCR conditions were the same as outlined above except that 14 cycles were used for amplification. The PCR reactions were pooled and the DNA was purified using 100 µl AMPure XP beads. The DNA was eluted with 11 µl elution buffer. The eluted DNA was subjected to another round of probe-mediated capture with the hybridisation timespan reduced to 24 h. Furthermore, after washing the beads, the captured fragments were amplified using only 10 cycles of PCR.

**Post-sequencing analysis of next-generation capture-C data.** The analysis was carried out in accordance with the original description of the method[87] with some modifications. Paired-end reads were processed using trim_galore v0.4.2 from the Babraham Institute (UK) to trim off low-quality bases (default Phred score of 20, i.e., error probability was 0.01) and adapter contamination from the 3' end. Only complete mate pairs were processed further to reconstruct single reads from overlapping paired-end sequences using FLASH v1.2.11[88] with interleaved output settings for non-extended reads (flash --interleaved --output --max-overlap 150). FLASH reads were split in silico at DpnII restriction sites using a designated perl script (https://github.com/telenius/captureC/releases) before aligning them to the X. tropicalis genome assembly 7.1 using Bowtie2 v2.2.9[67]. The alignment was run with default settings and one thread only to keep the order of the reads. The view function of samtools v1.3.1[75] was used to retain alignments with a quality score of ≥10. The alignments were analysed further using a suite of perl scripts (https://github.com/Hughes-Genome-Group/captureC/releases) modified to process both chromosome and scaffold coordinates. Viewpoint coordinates included a 1-kb proximity exclusion range. Restriction fragments were classified as capture, proximity-exclusion or reporter. Restriction fragments were classified as capture, proximity-exclusion or reporter. PCR duplicates were removed. The interaction map was based on the number of unique paired-end reads per restriction fragment. Windows of 2 kb incrementing by 200 bp were used to consolidate interactions, which were normalised to 10,000 interactions per viewpoint. Promoter contacts are listed in Supplementary Data 12.

**Analysis of genomic profiles, DNA motifs and super-enhancers.** Unless otherwise stated, Spearman correlation ($R_s$), relative tag density (meta-plots), principal component analysis (PCAs) and enrichment of DNA motifs were determined using a limited number of detected peaks per chromatin profile: top 2000 peaks (Figs. 3c–g and 4c and Supplementary Figs. 5, 6 and 7d), top 5000 peaks (Supplementary Fig. 1b) and top 10,000 peaks (Figs. 4b, e and 5c and Supplementary Fig. 7a–c). $R_s$ and PCA were calculated using DiffBind v2.4.8[89] in R (Figs. 1d, 3d, e and 4b and Supplementary Figs. 5a, 7a, b and 12a). Z-scores of motif enrichments at RNAPII peaks were calculated using ISMARA v1.2.2[90] (https://ismara.unibas.ch/fcgi/mara) in expert mode (Fig. 1e). Heat maps of relative tag densities were hierarchically clustered (Ward's method) using the R package seriation[91] except for those in Figs. 4c and 8d, where genomic coordinates were sorted upon significant changes in DNA occupancy or chromatin accessibility (see Figure legends). In Supplementary Fig. 6, cluster leaves were additionally ordered with the Gruveaus-Wainer's algorithm. The matrices of normalised tag densities, p-values and motif instances were generated with HOMER as follows: annotatePeaks.pl -size 2000 (or 200 for Supplementary Fig. 6) -hist 25 (bin size) -ghist. In heat maps, all genomic profiles were scaled to the means of the highest bin values of the top 100 (or 1000 in Figs. 4c and 8d) peaks. In addition, to visualise low RNAPII binding and RNA expression, their top means were divided by 5 or 500, respectively (Fig. 8d). Super-enhancers were assembled by stitching together peaks that are ≤25 kb apart using: findPeaks -style super -fragLength 175 -inputFragLength 175 -fdr 0.001 -gsize 1.435e9 -F 3 -L 1 -localSize 10000 -C 0.97 -superSlope 1 -minDist 25000. Subsequently, we combined these super-enhancers according to the developmental stage of profiling (see Supplementary Data 6 for genomic coordinates). Super-enhancers ≤5 kb from TSS apart were associated with zygotic genes defined by RNAPII-mediated transcription elongation[92].

**Analysis of enriched gene ontology (GO) terms.** Over-represented GO terms were found by applying hypergeometric tests of the Bioconductor/R package GOstats v2.42.0[93] on gene lists. The process was also supported by the Bioconductor/R packages GSEABase v1.38.1[94] and GO.db v3.4.1[95]. The gene universe was associated with GO terms by means of BLAST2GO[64,96,97].

**Differential analysis of chromatin features.** The significance of differential chromatin accessibility between control and mPouV/Sox3 LOF embryos was assessed at pCRMs using the Bioconductor/R package DiffBind v2.4.8[89] and the DESeq2[78] algorithm. The size of all pCRMs was fixed to 500 bp. Only the very 5'

end of each aligned DNase-Seq read, which represents the DNase cleavage site, was considered for this analysis. Read counts from mapping DNase-digested naked genomic DNA were subtracted from chromatin accessibility readings at each pCRMs. Any changes to DNase-mediated cleavages at pCRMs with FDR ≤10% were determined as statistically significant (Fig. 8b). For visualisation, the orientations of pCRMs were aligned to the direction of transcription of the closest zygotic gene. Matrices of normalised and raw tag counts were generated at pCRMs for various chromatin features and RNA (Fig. 8d). Their rows were sorted according to the statistical significance of differential chromatin accessibility. Raw tag counts in sliding (+200 bp) windows of 400 bp (accessibility, H3K4me1, RNAPII, Smad2 and β-catenin) or 2 kb (RNA and promoter contacts) across the genome were processed to estimate p-values of chromatin changes (pΔ) using DESeq2 v1.16.1[78]. Read count dispersions were locally fitted. The shrinkage of $log_2$ fold changes, threshold on Cook's distances and independent filtering were all switched off. For the comparison of local interactions with promoters at affected and unaffected pCRMs, the size of the pCRMs was widened to 1.2 kb and filtered for ≥5 reporter fragments (Fig. 9c). Normalised line and bar graph tracks of control and mPouV/Sox3 LOF embryos were superimposed in the IGV genome browser v2.3.92[98] (Figs. 9d and 10a, b and Supplementary Figs. 13 and 14).

**Quantification and statistical analysis.** No statistical method was used for determining sample size. We followed the literature to select the appropriate sample size. The experiments were not randomised. Due to the nature of experiments, the authors were not blinded to group allocation during data collection and analysis. Only viable embryos and embryonic tissues were included in the analysis. Due to space constraints, data from the trials of establishing and optimising protocols (ChIP-Seq, DNase-Seq and next-generation capture-C) were excluded. Frequencies of shown morphological phenotypes and WMISH patterns are included in every image. Significances of normally or non-normally distributed data points were calculated using a paired Student's t-test (one-tailed or two-tailed) or paired Wilcoxon test (two-tailed), respectively. The effect size r was estimated from the standard normal deviate of the Wilcoxon p-value (p) according to Rosenthal[99], $r_{effect} = Z/\sqrt{N}$, where $Z = qnorm(1-p/2)$ is the standardised Z-score and N is the number of observations. Significance of motif enrichments or GO terms are based on hypergeometric tests. For RNA-Seq, biological triplicates were used to account for transcriptional variability between clutches. Each LOF experiment has its own control embryos collected in parallel from the same mothers: exp. #1 (α-amanitin), uninjected embryos; exp. #2 (BMP or Nodal LOF), DMSO-treated embryos; exp. #3 (Wnt LOF), uninjected embryos; exp. #4 (mPou5f3/Sox3 LOF), uninjected embryos; exp. #5 (mVegT LOF), uninjected embryos. The gene expressions of control MO-injected embryos of exp. #2 and #5 were normalised to their corresponding uninjected embryos. The mean of these normalisations and conservative FDR estimations (i.e., higher FDR of the two likelihood ratio tests) were used for the comparison with LOF conditions. RNA-Seq libraries from each experiment were generated simultaneously to mitigate any batch effects. The significance of transcriptome-wide differential expression was adjusted for multiple comparisons using the Benjamini-Hochburg procedure.

**External datasets.** The following external datasets were used for this study: 5mC at stage 9, and ChIP for H3K4me3 and H3K27me3 at stage 9 (GSE67974), β-catenin ChIP at stage 10+ (GSE72657), Smad2 ChIP at stage 10+ (GSE30146 and GSE53654), Foxh1 ChIP at stage 8 and 10+ (GSE85273 and GSE53654), total RNA recorded at 30-min intervals from unfertilised eggs to 23.5 hpf (GSE65785), ribosome footprinting from stage 3, 9 and 12+ (GSE52809), and regional expression along the animal-vegetal and dorso-ventral axis at stage 10+ (GSE81458).

**Reporting summary.** Further information on research design is available in the Nature Research Reporting Summary linked to this article.

## Data availability

Sequencing data are deposited in the Gene Expression Omnibus (GEO) database with accession number GSE113186. All other relevant data supporting the key findings of this study are available within the article and its Supplementary Information files or from the corresponding authors upon reasonable request.

## Code availability

The source data underlying Figs. 1d–f, 2b, 3c–g, 4b, c, e, 5c, 6e, f, 7a, b, e, 8b–g and 9c, Supplementary Figs. 1a, b, d, 5, 7a–d, 9, 10a, b, 11, 12a and 15–17 and Supplementary Data 9, 11 and 12 are available on GitHub at https://github.com/gegentsch/SignalCompetence. Post-alignment analysis and graphical illustration were performed using samtools v1.3.1[75], bedtools v2.25.0[100], HOMER v4.8.3[68] and tools in R v3.4.1/Bioconductor v3.5, Perl v5.18.2 or Python v2.7.12.

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

## Acknowledgements

We thank Abdul Sesay, Leena Bhaw, Harsha Jani, Deborah Jackson and Meena Anissi for deep sequencing; Mike Klymkowsky, Manolis Gialitakis and John Gurdon for supplying antibodies; Stefan Hoppler and Yukio Nakamara for discussing β-catenin ChIP protocol; Jim Hughes and Damien Downes for sharing pre-publication results of an improved next-generation capture-C method; Ying Wang for providing sequence conservation tracks; Mareike Thompson, Rita Monteiro and Clara Collart for critical reading of the manuscript; and the Smith lab for discussions and advice. G.E.G and J.C.S. were supported by the Medical Research Council (program number U117597140) and are now supported by the Francis Crick Institute, which receives its core funding from Cancer Research UK (FC001-157), the UK Medical Research Council (FC001–157), and the Wellcome Trust (FC001–157).

## Author contributions

Conceptualisation, G.E.G.; Methodology, G.E.G.; Software, G.E.G.; Formal analysis, G.E.G. and N.D.L.O.; Investigation, G.E.G. and T.S.; Writing—Original Draft, G.E.G. and J.C.S.; Writing—Review & editing, G.E.G and J.C.S.; Funding acquisition, G.E.G. and J.C.S.; Supervision, G.E.G. and J.C.S.

## Additional information

**Competing interests:** The authors declare no competing interests.

