## [Peer Review File · Nature Communications]

Reviewers' comments:

Reviewer #1 (Remarks to the Author):

This paper follows the theme that the competence of embryonic cells to respond to signals depends on whether maternal factors have already opened the chromatin. On initial reads, I got the impression that there was going to be something particularly special about PouV and Sox3, but it would be good to see a more clear discussion of this, since there are other competence factors that can act on different genes from Pou5 and Sox3. Is there really a rational distinction between these sets of genes- especially with regard to pluripotency? It would also be good to put the paper in the context of the frog work on FoxH1, which seems to have a similar message, ie Foxh1 activates different sequences in chromatin to be competent to respond to signaling mediators. While it is clear that this paper has a great deal more data and examines many more TFs, It would be useful to list what is explicitly new here in principle and in detail, and why Pou5, Sox3 and VegT are chosen for emphasis. However, the authors do analyze several maternally encoded factors, including FoxH1, which is covered in this paper, and others, that regulate competence, but these factors do not get as much attention as Sox3/Pou5. Perhaps more could be analyzed in the structure of chromatin near brachyury and eomesodermin- brachyury requires the Pou5 /SoxB pair, so what does Eomes need?

Overall, this is an impressive piece of work, analyzing many variables in chromatin and transcription, and putting them together. It suffers from the volume of data, such that the paper is dense, and not helped by the sometimes odd sentence constructions (Uncharacteristic of the usually clear writing, and grammatical attention of Sir Jim).

It would also be useful to put this work in the context of the fish work, which did not go so far in analysis of chromatin, but analyzed ZGA with respect to loss of Sox, Pou and "nanog". Indeed, the absence of nanog here is notable- Perhaps there should be some explicit mention here that nanog is not universal, and genes that have been called nanog homologues may sometimes not be true homologues.

As I allude to below, the choice of what is in the paper and what is in the supplement seems artificial, with important points not supported by data in the main text. Also, even in the supplement, where presumably there is no space limit, the opportunity could be taken to head each part of the legend with a brief conclusion, and some minimal description of what has been done.

As an important analysis, and reference work, the paper definitely should be published, since it illustrates the previously documented roles of PouV/SoxB in Zygotic gene activation, but adds the

crucial analysis at the level of chromatin. However, it does suffer from its compression, and many of the experiments are not presented for the general audience to understand how the conclusions were drawn. This may be hard to resolve in the compressed format of Nature, (also with its utilitarian but inadequate partial references).

Major and minor detailed comments (no need to publish these I hope!):

“Thus, a more plausible way in which tissue-specific competence might arise is through the recruitment of these signal mediators to different genomic sites. “

What is this more plausible than? Different sites from what?

“Second, ENCODE studies highlight the high gene regulatory capacity of maternal or top-level TFs, which makes them more amenable to study8”

More so than zygotic TF's? This statement assumes a lot, and doesn't have enough information to know what the author means.

“these TFs have not been identified in a systematic fashion, and their modes of action remain largely unknown. “

But there have been very nice studies that identified such things as Shh competence factors in the limb versus the nervous system. Is the pax6 example really a good one?

“using transcriptional, translational and multi-level chromatin profiling, we first identified the earliest active regulatory DNA sequences (cis-regulatory modules, CRMs) in *X. tropicalis*, “

While these sequences identified from the various methods are likely to be CRMs it would probably be best to say explicitly that these sites are assumed to be CRMs at the beginning, since there is no data reviewed to show that these CRMS are sufficient to support transcriptional control.

“effect of TF co-expression on chromatin recruitment in vivo. “

Don't the authors mean the effect of co-expression of transcription factors on their recruitment to chromatin in vivo? Or do they really just mean generic chromatin recruitment.

page 3 “Based on this orthogonal correlation, “

But the orthogonal correlation does not refer to what was just discussed? i.e CRM conservation. The authors are referring (this) to an earlier comparison and might lose the reader here.

“Based on this orthogonal correlation, we selected Pou5f3, Sox3 and mVegT as potential competence factors of canonical Wnt, Nodal and BMP signaling. “

It is not clear why the “orthogonal correlation” chose these, and not the others, such as FoxH1. And is this the right use of orthogonal? It would be good to be explicit about what the two correlated measures were, if this really selected PouV/Sox3. Otherwise, its fine to concentrate on these, but the

reader gets the impression that there is something special and different about these factors, and I do not see that there is.

Context-dependent TF co-expressions affect the dynamics of chromatin recruitment

The documentation of the characteristics of the TFs and signaling mediators is said to show that they share special characters, but the special characters are not compared to outgroups.

“DNA occupancy levels followed a log-normal distribution with only a few hundred RNAPII-transcribed zygotic genes receiving high and super-enhancer-like²¹ input “

Odd construction is used in the supplement where a “Chromatin landscape of [endogenous transcription factor] to super-enhancer of the *mesp* gene. “ While a string of code words can be cracked, it might be useful to have a more coherent sentence, maybe with a verb for clarity.

Was fig 2b early tailbud from dissected embryos? Or is this drawn this way because this is the predominant region of expression in the tailbud? A reference, eg to Xenbase or the literature might clarify this.

The “Super enhancer associated binding correlates withDNA occupancy of TFs. Isn’t this circular? Maybe this sentence needs to be re-ordered or clarified?

in c, It is not immediately clear what the plot shows. What does normalized mean- is this between samples or within samples?

“From the 1,024-cell to the late gastrula stage, frequent chromatin recruitment occurred on average to ~800 super-enhancers which had zygotic genes nearby (<5kb) that are primarily involved in early embryonic processes including germ layer and body axis formation (Supplementary Fig. 2d) “

That shared property was hard work- isn’t the shared property more about the embryonic processes, rather than the frequent chromatin recruitment. Shouldn’t the main issue be in the main sentence rather than a sub clause? And compared to what condition?

“Among all zygotic genes, promoter-proximal regions were most consistently bound “

Compared to what other condition?

“factor-intrinsic versus developmental context dependent. What is factor intrinsic? Are developmental regulators not controlled by DNA binding proteins such as FoxH1 with intrinsic binding ability? Or does factor intrinsic just mean they are not developmentally regulated?”

In the supplement, the legend “PCA of chromatin engagement” is pretty terse. Presumably this is of the target sequences?

Re Context-dependent TF co-expressions affect the dynamics of chromatin recruitment

It took me a long time to find the orange frame, and now that I found it, I do not understand the point being made for orange and black frames. Nor do I understand the basis for “This observation supports context-dependent signal interpretation (top panel in Fig. 2c): “ Perhaps it would be useful

to explain how the results would have differed in the boxes if there was not context dependent signal interpretation.

“The discovery and co-clustering of DNA motifs at occupied CRMs indicate that factor-intrinsic elements include the sequence- specificity of DNA binding domains, oligomerization tendencies (e.g. Brachyury binding palindromic motifs due to its propensity to form homodimers²²) and protein-protein interactions, while the developmental context reflects co-expressed TFs “

There has to be a clearer way of saying this, perhaps starting with better subjects for the sentence that fit with the verbs used? And I don't understand the second part of the sentence.

Ref 18 belongs in the introduction as well as here- isn't the approach and conclusion somewhat similar, and therefore reflects the state of knowledge that should be introduced?

Figure 2e is not described until much later, but even then it is not described in the legend or text enough to know what it means. Furthermore fig 2f appears to only have one significant box- where Sox3 is recruited to E boxes by MyoD. The recruitment of polII is expected from the previously documented effect of MyoD on transcription.

I do not follow why “MyoD-HA binding itself was influenced by the endogenous TF expression profile” though admittedly I expect a major point like this should be in the main text, perhaps it is in the supplement. (Which I read later)- raising the question, Shouldn't the reader be able to follow the main paper without reference to the supplement? And raises the question of why so many essential figures are in the supplement? This one is hard enough to follow without having to jump to the supplement all the time.

Would it be good to explain the colors earlier, since people tend to read the legend from the beginning to the end.- Is yellow a wise choice in h?

In this and the MyoD case, where Sox3 is a follower to another protein (MyoD), what impact does this have on the argument that Sox3 is a pioneer that establishes a competent chromatin state for signaling?

Signal-induced regionalization of ZGA depends on maternal TFs

I'm not seeing any transient expression of TFs in figure 1h, as referred to in the text.

If brachyury is spelled out, then shouldn't eomesodermin also be spelled out?

“By comparison, unaffected CRMs rarely contained canonical POU/SOX motifs, and were strongly enriched for promoter-centric motifs of the Krüppel ZF, bZIP and NFY protein family “

It is surprising that the Fox motifs are not in these classes- do all Fox sites coincide with Sox/Pou, or are they just at regions that are less accessible?

Discussion

“Our results allow us to propose a model (Fig. 6) of pioneer-initiated chromatin remodeling, or priming, that unlocks context-specific CRMs, some of which contain signal responsive elements to enhance and regionalize transcription. “

Isn't this an enhancement of the previous models, and specifically that for FoxH1

It would be nice to have an explicit documentation of the pluripotency gene aspect- are Sox3 and Pou really pluripotency activities in this context, or just two of a set of various maternal pioneer factors that activate the zygotic genome?

On the competence angle, perhaps it would be worth mentioning the similar principle from the effect of the Zn finger protein that acts to terminate ectodermal competence to respond to mesoderm induction, but only appears to act, as here, through a subset of genes, specifically those that have p53 inputs.

Reviewer #2 (Remarks to the Author):

Gentsch et al. investigate the role that maternal transcription factors play in establishing the cell type specific responses to inductive signals during early embryonic development. They tackle this important developmental question using the powerful *Xenopus tropicalis* model system. The authors use genomics, bioinformatics and developmental biology approaches to investigate how context-dependent signaling transduction is achieved. Starting with motif analysis within earliest activated CRMs, the authors identified and characterized the roles of maternal TFs, specifically mPouV/Sox3, in unlocking signal-responsive CRMs during *Xenopus* ZGA.

A major strength of the manuscript is the very large number of original and relevant experiments (ChIP-seq, RNA-seq, Capture C, DNase-seq) done in primary *Xenopus* tissues at one or more developmental stages as well as after several perturbation experiments. They anchor their study at the mid-blastula embryo (4096 cells) which shows high levels of homogeneity, little transcription and diversification.

A major weakness of this paper is that it is dense and challenging to read. Specifically, the paper could benefit from a simpler, more systematic approach to data integration and data summary.

I found the manuscript challenging to read and hence review. At an initial and superficial, yet practical level, I would have preferred a more conventional format, rather than a mocked-up

publication format with tiny multi-panelled figures. For example, a conventional format makes it easier to identify specific lines, pages for comments (For this reason I will include quotations in my review to highlight the text I am referring to).

While it was clear to me that great effort was made to produce visually appealing figures in both the main text and supplemental material, there are many figure panels where reading the figure legend and looking at the figure was not enough to understand what was being shown.

Moreover, when reading the main text that referred to the figures, it was often challenging to sift through the jargon and understand the underlying logic, details, and caveats that led the authors to so confidently state their conclusions. In my opinion, the authors often seem to state their conclusions and then give reference to a Figure panel or Supplemental figure panel without providing enough details or context to appreciate how their experiments lead them to their conclusions (I will try and provide some examples below however this occurs throughout the paper).

I found myself having to rely on the methods to understand what analyses were performed. In many cases there was not enough information to assess whether the data the authors presented supported their conclusions. While I appreciated the detailed experimental methodology, I found that the bioinformatics analyses were not sufficiently described.

Specific Comments:

At the beginning of the results section the authors mention that RNAPII is a reliable and objective indicator of CRM activity. I agree with this. However, I don't think one can simply identify CRMs using RNAPII ChIP-seq and then say that these RNAPII regions themselves are, by definition, active CRMs. I could not find details about the number of these putative 'CRMs' (i.e. the average length, and evidence of where they are located, their overlap with other data from Mod ENCODE or the literature was not given). How many, and by what criteria, were putative CRMs assigned as being obscured by RNAP2 occupancy at promoters and gene bodies? Since these CRMs form the basis of much of the analyses in the paper I think these details need to be more transparent.

It was mentioned that largest changes in RNAPII recruitment to CRMs occur between 1024 stage and MBT. While the main text and supplemental exemplary genome browser views suggest the data is of good quality, little information is given for the data comparisons made between developmental stages.

Fig 1d shows a summary of the differential motif enrichments at the different developmental stages. It is not clear to me how this analysis was done. I see that “Z-scores of motif enrichments at RNAPII peaks were calculated using ISMARAv1.2.281 (<https://ismara.unibas.ch/fcgi/mara>) in expert mode (Fig. 1d). and “Heatmaps of relative tag densities were hierarchically clustered (Ward’s method) using the R package seriations”. Without knowing which peaks went into the analysis one cannot reproduce this analysis easily or properly understand what was done. Looking into ISMARA, expert mode requires the user to put in specific intervals/gene names as well as motif matrices. It is not clear what exactly was used to derive the motifs (e.g. were all matrices from a collection used?).

The supporting Supp Fig 1b provides additional information. However, Supp fig1b legend does not appear to explain what the green and black circles mean in terms of number of CRMs and background. I see in the methods section that Top N peaks were used in some analyses however there does not seem to be information about what was done for Supp Fig 1b.

The supplementary data Table 1 was the only place where the experimental design with the number of biological replicates was presented. I think the authors should mention in the main text as well the number of biological replicates used for each experiment (this is done for the LOF experiments). I did not request/obtain the token to access and verify the GEO submitted data, however given the minimal analyses and description of the data generated, I think it will be helpful for the authors to deposit both raw, and aligned data, as well as the identity of the various CRMs so that others can easily look at the data and reproduce the analyses.

In the second paragraph of the results, the authors acknowledge the limitations of using RNAPII for de novo CRM discovery. They go on to use DNase hypersensitivity and H3K4me1 ChIP-seq to show the regions with RNAPII activity within genes at the MBT stage are in accessible chromatin. Are there DNase plus H3K4me1 enriched regions that gain RNAP2 later in development? To what extent would performing DNase-seq at the pre-MBT stage inform about the pioneering activity of the maternal TFs (e.g. would the authors expect open chromatin shared between pre-MBT and MBT that shows a gain of RNAP2 activity at the MBT stage (or later) to be examples of chromatin opened by maternal pioneer factors?

It would be useful to integrate the ChIP-seq data from the 1024 cell stage (I see β -catenin ChIP, Sox3 ChIP, VegT ChIP data from 1000 cell stage listed in Table 1). To what extent are the pioneer factor bound regions already bound by inductive TFs at the 1000 cell stage? What fraction of Sox3 or VegT bound regions are not considered to be part of RNAP2 defined CRMs at the 1000 cell stage? Do

RNAP2 negative Sox3/VegT bound regions at the 1000 cell stage become sites of later TF or RNAP2 usage? The answers to these questions should help support the authors definition of CRMs.

I think the use of ribosome footprinting and proteomics to identify relevant transcription factors was well justified and explained. Given the literature, would the authors have expected to have concluded that Pou5f3, Sox3 and mVegT would be the pioneer factors to study further?

At first glance the model in Figure 1h is visually appealing. However, for this reviewer the components of the figure were not sufficiently explained. The figure was said to be derived from the authors previously published data derived from 6 references. Which data from each reference were used is not clear. Perhaps this kind of figure would make sense in the discussion or review article, assuming it was properly explained.

The authors tend to present their conclusion/interpretation without clearly describing their observation/results first, which is often confusing. Figures like Fig. 2b appeared multiple times in the manuscript, so it may be worthwhile the authors to take the time to explain it when it first appears. Given the several orange elements in the figure it took me a moment to notice the gold box which I presume is the orange frame that is being referred to. I have a similar comment about the black frame. This may simply be a function of the small figure size. Fig. 2b seems like an expected result where most TF bound regions were enriched for their expected motif. I would also expect that signal mediator TFs would bind regions bound by the other maternal TFs. It would be helpful for the authors to refer to expectations coming from previous studies and highlight what was already known, what was expected and what was novel.

For example, for Fig. 2b, which is one of the key figures that support the author's argument, the authors directly brought up their interpretation that "maternal TFs affected the recruitment of signal mediators (see orange frame in Fig. 2b), but (with the exception of Foxh1, which interacts directly with Smad2) the reverse was much less the case". The logic could be clearer if the author provides more description and context for these results. For example, "Enrichment of maternal TF motifs were observed within the binding sites of signal mediators, however the reverse was not seen. This indicates that maternal TFs are likely to affect the recruitment of signal mediators, just as developmental lineage-specific TFs are known to recruit signal mediators later in development (insert representative references from the literature)".

This paragraph is dense and hard to follow:

“Maternal TFs and signal mediators shared many chromatin characteristics: (1) DNA occupancy levels followed a log-normal distribution with only a few hundred RNAPII-transcribed zygotic genes receiving high and super-enhancer-like²¹ input (Supplementary Fig. 2a-c). (2) From the 1,024-cell to the late gastrula stage, frequent chromatin recruitment occurred on average to ~800 super-enhancers which had zygotic genes nearby (<5kb) that are primarily involved in early embryonic processes including germ layer and body axis formation (Supplementary Fig. 2d). (3) Among all zygotic genes, promoter-proximal regions were most consistently bound (Supplementary Fig. 2e). All these characteristics were also observed for chromatin accessibility and RNAPII engagement at CRMs (Supplementary Fig. 2c,e).”

The following paragraph is also hard to follow. It seems like the findings here are important for the paper but I struggle to make sense of the writing. I think the factor-intrinsic and developmental context-dependent jargon which emerges here generates some ambiguity. Throughout the manuscript TFs are classified using several terms. It would be helpful to have these terms defined clearly once, and then consistently used throughout. For example: pioneer factors, co-expressed maternal TFs, factor-intrinsic TFs, development context-dependent TFs, and Signal TFs.

“The comparison of DNA occupancy levels among and between signal mediators and TFs suggests that the recruitment of both depends on factor-intrinsic and developmental context-dependent elements (Supplementary Fig. 3a,b).”Conversely, signal-initiated recruitment fails to generate cell type-specific responses (bottom panel in Fig.2c).

When I look at Supplementary Fig 3a, b and the associated legend this conclusion is not immediately apparent. Which data and comparisons of 'DNA occupancy levels' went into this conclusion?

This paragraph continues to be challenging to follow:

“The discovery and co-clustering of DNA motifs at occupied CRMs indicate that factor-intrinsic elements include the sequence specificity of DNA binding domains, oligomerization tendencies (e.g. Brachyury binding palindromic motifs due to its propensity to form homodimers²²) and protein-protein interactions, while the developmental context reflects co-expressed TFs (Fig. 2b and Supplementary Fig. 3d,e).”

It seems to me to be saying something obvious about motifs at CRMs (TFs have DNA binding motifs, they bind together at CRMs, and in doing so may interact with other proteins). However, the sentence is making a contrast of this idea to 'developmental context'. It is not clear how the three figure panels referred to support the arguments in this paragraph.

Figure 2a is labelled with “scoring the enrichment of DNA motifs at occupied CRMs however the figure does not have any scores or motifs on it.

The model referred to in Figure 2C is needs to be clarified and would be better suited for the discussion.

In the latter half of this manuscript, the authors focused on maternal factor mPouV/Sox3 and demonstrated how Nodal signalling-initiated transcription depends on pioneer activities of these maternal TFs. However, since Sox3 and Pou5f3 are distributed ubiquitously in early embryos

(according to the author), it does not seem to be a convincing example to show how pioneer factors leads to regionalized transcription. To me, it did not directly answer to the concept raised in introduction. The authors should address this more clearly in their discussion and what is already known in the literature.

The MyoD-HA experiment is interesting although the logic for deploying a lineage-specific pioneer factor MyoD at the MBT stage could be explained a bit more. It's not clear how Fig. 2f and Supplementary Fig. 4b showed that "MyoD-HA binding itself was influenced by the endogenous TF expression profile. The author should elaborate more and explain what an endogenous TF expression profile means. How this connects to the model Figure 6 needs to be clarified. Isn't it expected that MyoD can be a pioneer factor which recognizes its own motifs and can also bind to existing open chromatin with weaker motifs? Also Supp Figure 4E mentions Sox3-HA whereas Sox3-HA is not mentioned in the main text.

The details regarding the intron and exon analyses of RNA-seq data was not clear. The description of the results from the many RNA-seq experiments performed could also be clearer.

The authors perform Capture-C and however the experimental details and explanation of results were minimal.

The $p\Delta$ was used in Figure 5 but I could find little information how it was calculated for the Capture C data. Are the regions shown significant? Providing actual statistics for the specific examples given in the paper would be helpful.

Legend for Figure 5D and E are minimal. What is the definition of the "mPouV/Sox3-dependent ZGA"?

Evidence behind and the relevance of this conclusion written in the results section was not clear. Did the authors look at transcription of CRMs as well as RNAP2 occupancy?:

"At the transcriptional level, lower usage of CRMs coincides with the reduction of coding as well as local non-coding RNA (Fig. 5b,e and Supplementary Fig. 7g). Importantly, the differential

profiling explained how chromatin predetermines signal induction and why, for instance, Nodal-induced transcription of brachyury was strongly affected by the loss of mPouV/Sox3 and that of Eomes less so.”

It seems abstract to say that “differential profiling” can explain a concept. From looking at the data in Figure 5b it seems that CRMs undergoing a (significant?) reduction in accessibility after mPouV/Sox3 LOF coincided with the loss of inductive TF binding. More explanation here would be helpful.

“Sorting of these CRMs according to the significance of lost accessibility reveals that chromatin opening depends on the pioneering activity of mPouV and Sox3 to recognize their canonical motifs in compacted chromatin (Fig. 5c).”

Here is it more accurate to say “lost accessibility suggests that chromatin opening depends on ...”

At the extreme end of affected loci (e.g. brachyury, foxb1, cdc25b and zic1) entire super enhancers became accessible through mPouV/Sox3 activity (Fig. 5f and Supplementary Fig. 7g).

In this case what is the evidence that a super enhancer became accessible through mPouV/Sox3? Also my understanding was that mPouV/Sox3 activity was knocked down (not activated) in this experiment? What factor(s) were used to create the super enhancers? It would be odd to use RNAPII to build super enhancers as clusters of RNAPII would surely indicate stretches of transcribed genes. I may have missed this in the methods but if the authors use RNAP2 to generate super enhancers, more justification is needed.

At the end, the author proposed two models of Nodal gene activation based on their dependency on mPouV/Sox3, with eomes and brachyury as examples. The authors suggest the mechanistic differences underlying their mPouV/Sox3 dependency is whether SOX/POU motifs exist within the enhancer regions of the Nodal genes. It will be more convincing if the author shows this in a systematic way, for example divide the Nodal genes into mPouV/Sox3 dependent and independent categories (as in Fig. 4b), analyzing their enhancer regions to see if one category enrich for SOX/POU

motifs while the other category does not. This analysis may also reveal other pioneer TFs that are important for Nodal signaling.

With all the RNA-seq and ChIP-seq data the author generated, the author should be able to look at the BMP and WNT targeted genes in a similar manner as what they did to the Nodal genes, dividing them into mPouV/Sox3 dependent and independent and characterizing the enhancer motif enrichment for each categories.

All loss of function models were achieved through MO knock-down or small molecular inhibitor, which could lead to non-specific defects. Nevertheless, the author showed good validation of their MOs, including (1) western blot or WMIHC demonstrating MO-mediated protein reduction; (2) RNA rescue at morphological or transcriptional level.

Mutating one or a few POU/SOX motifs that are found within the CRMs of brachyury would be a relevant experiment. If the POU/SOX motif mutation itself leads to loss of accessibility, reduced active histone markers and downregulation of brachyury, should help support the model proposed by the authors.

Supplementary Fig. 6a, The color legend, graphics, and labelling is confusing. For example, in the part of the figure referring to the 'pluripotency' stage how does the 'Opportunist' right branch of the panel relate to the results in the paper? What proportion of maternal binding of Pou5f3/Sox3 is opportunistic? What maternal TFs would be considered to fall in the "other pioneer factor (Developmental context)" during the pluripotency stage? The figure suggests these are bound to open chromatin established prior to Pou5f3/Sox3. Is this correct?

The abstract summary "Our work identifies significant developmental principles that inform our understanding of tissue engineering and tumorigenesis." does not accurately reflect the analysis and work that was done in the manuscript. While this argument was briefly formulated in the discussion, I do not think it fairly represents the content of the work described and I think a summary which is more in line with the data and data analysis is warranted.

The authors note in the abstract that: "The remodeling includes the opening and marking of thousands of regulatory elements, extensive chromatin looping, and the co-recruitment of signal-mediating transcription factors." While Capture-C data is presented in the manuscript, little information is given about chromatin looping changes during early development was given aside from two important example genes.

Reviewer #3 (Remarks to the Author):

The role of maternal pioneer factors in predefining first zygotic responses to inductive signals

Gentsch et al.

In this very nicely written manuscript, the authors use ChIP-Seq of RNA PolII during *Xenopus* development to identify putative regulatory elements driving the maternal to zygotic transition, and then characterise these elements using numerous additional datasets, including ChIP-seq for transcription factors, signal mediators, DNase-seq and analysis of motifs. It's an impressive collection of new datasets which will provide an excellent resource. The authors use these datasets to show that the ubiquitously expressed and maternally Pou5f3 and Sox3 are required for normal embryonic development, including expression of genes which are zygotically expressed in a subset of the embryo. Using loss-of-function experiments, they show that this is likely due to the requirement for these factors to increase chromatin accessibility at their binding sites and allow additional factors to bind in response to the region-specific activity of signalling pathways, and propose a model in which ubiquitously-expressed transcription factors act as pioneer factors for signal mediators.

The key claims are well supported by the data. The provision of code will be very helpful for reproducibility, and the code appears to be well documented and to use appropriate methods. I tested a subset of the provided code and was able to run it with very minor modifications.

Major points

- Figure 2b is provided to support a key claim in the paper, that transcription factor binding can recruit signal mediators, but not vice versa. However, the figure is quite hard to interpret. It would be helpful to have additional explanation of what the figure shows and how this supports the conclusions, either in the text of the figure legend.
- One reason why the figure is confusing is that in this context one would typically expect to see data on co-binding of transcription factors and signal mediators. However, the authors show only enrichment of motifs for signal mediators at regions enriched for TF binding (and vice versa) –

not enrichment for signal mediator binding at regions enriched for TF binding. Enrichment of a motif does not necessarily imply enrichment of binding as measured by ChIP-seq signal. This key claim could be further supported by showing ChIP-seq signal enrichment as well as motif enrichment. It would also be helpful to comment on the extent to which these are concordant, as well as reasons why they might not be, such as the possibility of indirect binding or non-canonical motif usage.

- Although the developmental context of the proposed model, and the specific focus on transcription factors which respond to signalling pathways, are novel, the underlying model of pioneer factors allowing other factors to bind has been previously widely discussed. The authors should add a discussion of previous work on pioneer factors to further place their findings in context.

Minor points

- In general, the figure legends are quite brief and don't allow a full understanding of the figures without also referring back to the text.
- The choice to plot differential gene expression data with a scale of 0-200% of control expression can make these figures harder to interpret for readers who are used to the more typical presentation of such data as log₂ fold changes. It can also be misleading as the upper end of the scale is truncated to only a two-fold change, while the lower end of the scale shows the full range of the data (although that may be appropriate in this case if the majority of differential gene expression is downregulation?).
- I had to make a couple of minor changes to the code to get it to run on my machine. For example, Bioconductor packages, e.g. Diffbind, cannot be installed with `install.packages()` – you could replace this with a call to `biocLite()` or use the new BiocManager package. I had to replace “Arial” with “ArialMT” for the pdf fonts, but this may be OS-specific. Also, it appears that the file name “dnase_test.csv” should be “dnase.csv”.

Elizabeth Ing-Simmons and Juan M Vaquerizas

The role of maternal pioneer factors in predefining first zygotic responses to inductive signals

George E. Gentsch, Thomas Spruce, Nick D.L. Owens, and James C. Smith

Our responses to the reviewers' comments are written in blue.

Reviewer #1:

This paper follows the theme that the competence of embryonic cells to respond to signals depends on whether maternal factors have already opened the chromatin. On initial reads, I got the impression that there was going to be something particularly special about PouV and Sox3, but it would be good to see a more clear discussion of this, since there are other competence factors that can act on different genes from Pou5 and Sox3. Is there really a rational distinction between these sets of genes- especially with regard to pluripotency? It would also be good to put the paper in the context of the frog work on FoxH1, which seems to have a similar message, ie Foxh1 activates different sequences in chromatin to be competent to respond to signaling mediators. While it is clear that this paper has a great deal more data and examines many more TFs, It would be useful to list what is explicitly new here in principle and in detail, and why Pou5, Sox3 and VegT are chosen for emphasis. However, the authors do analyze several maternally encoded factors, including FoxH1, which is covered in this paper, and others, that regulate competence, but these factors do not get as much attention as Sox3/Pou5. Perhaps more could be analyzed in the structure of chromatin near brachyury and eomesodermin- brachyury requires the Pou5 /SoxB pair, so what does Eomes need?

Overall, this is an impressive piece of work, analyzing many variables in chromatin and transcription, and putting them together. It suffers from the volume of data, such that the paper is dense, and not helped by the sometimes odd sentence constructions (Uncharacteristic of the usually clear writing, and grammatical attention of Sir Jim).

It would also be useful to put this work in the context of the fish work, which did not go so far in analysis of chromatin, but analyzed ZGA with respect to loss of Sox, Pou and "Nanog". Indeed, the absence of nanog here is notable- Perhaps there should be some explicit mention here that nanog is not universal, and genes that have been called nanog homologues may sometimes not be true homologues.

As I allude to below, the choice of what is in the paper and what is in the supplement seems artificial, with important points not supported by data in the main text. Also, even in the supplement, where presumably there is no space limit, the opportunity could be taken to head each part of the legend with a brief conclusion, and some minimal description of what has been done.

As an important analysis, and reference work, the paper definitely should be published, since it illustrates the previously documented roles of PouV/SoxB in Zygotic gene activation, but adds the crucial analysis at the level of chromatin. However, it does suffer from its compression, and many of the experiments are not presented for the general audience to understand how the conclusions were drawn. This may be hard to resolve in the compressed format of Nature, (also with its utilitarian but inadequate partial references).

Major and minor detailed comments (no need to publish these I hope!):

- **Lines 30-31:** "Thus, a more plausible way in which tissue-specific competence might arise is through the recruitment of these signal mediators to different genomic sites." What is this more plausible than? Different sites from what?
Other options to regulate signal competence are outlined before this sentence. For instance, competence could be regulated by the loss of signaling components, but this fails to explain the different responses to the same few signals observed in development. Therefore, we consider differences in genome-wide recruitment of the same signal mediators between different tissues a more plausible and common way in regulating signal competence. We have extended the sentence to make clear that the recruitment of the same signal mediators to different gene regulatory sites controls the generation of different cell types.
- **Lines 44-46:** "Second, ENCODE studies highlight the high gene regulatory capacity of maternal or top-level TFs, which makes them more amenable to study"
More so than zygotic TF's? This statement assumes a lot, and doesn't have enough information to know what the author means.
We have rewritten this sentence to clarify our statement: "Second, TFs involved in zygotic (or embryonic) genome activation (ZGA) are likely to occupy top-level positions within gene regulatory networks, and their binding should thus correlate strongly with gene target expression. This makes them more amenable to study". The information is derived from work on the architecture of human regulatory networks as part of the ENCODE project (Gerstein et al., 2012).

- **Lines 33-35:** "... these TFs have not been identified in a systematic fashion, and their modes of action remain largely unknown." But there have been very nice studies that identified such things as Shh competence factors in the limb versus the nervous system. Is the pax6 example really a good one? Because Pax6 is the classic textbook example of a protein affecting signal competence (Gilbert SF. Developmental Biology. 6th edition. Sunderland (MA): Sinauer Associates; 2000. Induction and Competence. Available from: <https://www.ncbi.nlm.nih.gov/books/NBK9993/>), we think it is the best example to mention here.
- **Lines 50-51:** "... using transcriptional, translational and multi-level chromatin profiling, we first identified the earliest active regulatory DNA sequences (cis-regulatory modules, CRMs) in *X. tropicalis*, ..." While these sequences identified from the various methods are likely to be CRMs it would probably be best to say explicitly that these sites are assumed to be CRMs at the beginning, since there is no data reviewed to show that these CRMs are sufficient to support transcriptional control.
We have introduced 'putative' to make this clear to the reader and abbreviated CRMs as putative CRMs (pCRMs).
- **Lines 53-54:** "... effect of TF co-expression on chromatin recruitment in vivo." Don't the authors mean the effect of co-expression of transcription factors on their recruitment to chromatin in vivo? Or do they really just mean generic chromatin recruitment.
Yes, we mean generic chromatin recruitment.
- **Line 91:** "Based on this orthogonal correlation, ..." But the orthogonal correlation does not refer to what was just discussed? i.e CRM conservation. The authors are referring (this) to an earlier comparison and might lose the reader here.
We have corrected this by specifically mentioning the attributes used to select maternal TFs: "Of the most frequently translated TFs with cognate CRM-enriched DNA recognition motifs..."
- **Line 91:** "Based on this orthogonal correlation, we selected Pou5f3, Sox3 and mVegT as potential competence factors of canonical Wnt, Nodal and BMP signaling." It is not clear why the "orthogonal correlation" chose these, and not the others, such as FoxH1. And is this the right use of orthogonal? It would be good to be explicit about what the two correlated measures were, if this really selected PouV/Sox3. Otherwise, it is fine to concentrate on these, but the reader gets the impression that there is something special and different about these factors, and I do not see that there is.
The reviewer makes a good point. We have corrected this (see reply to previous comment).
- **Lines 108-109:** "DNA occupancy levels followed a log-normal distribution with only a few hundred RNAPII-transcribed zygotic genes receiving high and super-enhancer-like input".
The documentation of the characteristics of the TFs and signaling mediators is said to show that they share special characters, but the special characters are not compared to outgroups.
These characteristics are commonly observed among our selected DNA binding proteins from maternal/zygotic TFs to signal mediators and pCRM-associated RNAPII. Outgroup examples like zVegT at late gastrula stage are shown.
- **Lines 541:** Odd construction is used in the supplement where a "Chromatin landscape of [endogenous transcription factor] to super-enhancer of the mesp gene." While a string of code words can be cracked, it might be useful to have a more coherent sentence, maybe with a verb for clarity.
We have changed the figure description to "Snapshots of TF and signal mediator binding to super-enhancers during early embryogenesis"
- **Lines 545-548:** Was Supplementary Fig. 3b early tailbud from dissected embryos? Or is this drawn this way because this is the predominant region of expression in the tailbud? A reference, e.g. to Xenbase or the literature might clarify this.
No, these profiles were generated from whole embryos. Tbx1 and Tbx6 are predominantly expressed in the tailbud as shown in the **Supplementary Fig. 3b**. We have added a reference.
- **Lines 552-556:** The "Super enhancer associated binding correlates with ...DNA occupancy of TFs. Isn't this circular? Maybe this sentence needs to be re-ordered or clarified?
We changed this statement to "The grey box explains the composition of the following dot plots: the x-axis displays zygotic genes (detected by RNAPII profiling from the 32-cell to the late gastrula stage) ranked by the total level of CRM accessibilities or CRM occupancies (normalized to 1 million mapped reads; primary y-axis) that are nearest (≤ 20 kb) to corresponding transcription start sites (TSSs). The secondary y-axis shows the cumulative frequency of gene-associated super-enhancers (≤ 5 kb from TSSs). Genes with super-enhancers are highlighted in red".
- **Lines 552-556:** In Supplementary Figure 4a, It is not immediately clear what the plot shows. What does normalized mean- is this between samples or within samples?

We rewritten and added more information to the legend of **Supplementary Fig. 4a**. The levels are normalized to 1 million mapped reads within samples. See quoted text above.

- **Lines 111-113:** “From the 1,024-cell to the late gastrula stage, frequent chromatin recruitment occurred on average to ~800 super-enhancers which had zygotic genes nearby (<5kb) that are primarily involved in early embryonic processes including germ layer and body axis formation (Supplementary Fig. 2d)’ That shared property was hard work- isn’t the shared property more about the embryonic processes, rather than the frequent chromatin recruitment. Shouldn’t the main issue be in the main sentence rather than a sub clause? And compared to what condition?
We have changed the sentence to “From the 1,024-cell to the late gastrula stage, TF-bound super-enhancers were linked through the gene ontologies of nearby zygotic genes (≤ 5 kb) to early embryonic processes including germ layer and body axis formation (**Fig. 3b**)”. The analysis is based on the overrepresentation of gene ontology terms using standard hypergeometric tests and zygotic genes only as the gene universe (usually conceptualized as the number of balls in an urn).
- **Lines 114-115:** “Among all zygotic genes, promoter-proximal regions were most consistently bound” Compared to what other condition?
We have extended the sentence to clarify that promoter-proximal regions were more consistently bound than other pCRMs.
- **Lines 116-130:** “factor-intrinsic versus developmental context dependent. What is factor intrinsic? Are developmental regulators not controlled by DNA binding proteins such as FoxH1 with intrinsic binding ability? Or does factor intrinsic just mean they are not developmentally regulated?”
We extended this sentence with the procedure leading up to the statement and simplified the latter by talking about protein’s own and other developmental-context dependent attributes:
“Differences in DNA occupancy levels at different developmental stages, compared by pairwise Spearman correlations and principal component (PC) analysis, suggest that the recruitment to chromatin of signal mediators and TFs is driven both by their individual properties and by the developmental stage (**Fig. 3d** and **Supplementary Fig. 5a**). For example, chromatin recruitment of mVegT at stages of pluripotency resembled that of other sequence-specific factors at the same developmental stage but differed from the binding of its zygotic isoform and of related T-box TFs at later developmental stages (highlighted in **Supplementary Fig. 5a**). The importance of developmental stage in driving patterns of chromatin recruitment was further revealed by the changing DNA binding patterns of sequence-nonspecific RNAPII to pCRMs (**Fig. 3e**).
The identification of enriched DNA recognition motifs at occupied pCRMs confirmed known TF/signal mediator properties such as the sequence-specificity of their DNA binding domains, oligomerization tendencies and protein-protein interactions (**Supplementary Fig. 5b**). For example, in contrast to other T-box TFs, Tbx1 recognizes palindromic T motifs due to its propensity to form homodimers²²; and Smad2 chromatin recruitment is frequently associated with the FOXH motif because Smad2 interacts with Foxh1²³ (**Supplementary Fig. 5b**). Other DNA motifs such as the POU or POU-SOX motifs were consistently co-enriched from the 1,024-cell to the late gastrula stage in most binding profiles. This is indicative of the pluripotent state, a developmental context associated with co-expression of Pou5f (Oct4) and SoxB1 (e.g. Sox2 or Sox3) proteins as previously observed *in vitro*²⁴ (**Fig. 3g** and **Supplementary Fig. 5b**).”
- **Lines 375-380:** In the supplement, the legend “PCA of chromatin engagement” is pretty terse. Presumably this is of the target sequences?
We have rewritten the legend to explain the figure (now **Fig. 3d**) in more details: “Biplot of principal component (PC) 1 (accounting for 40% variance) and 2 (28% variance) shows the similarity (or dissimilarity) of TF (Sox3, mVegT, Foxh1^{18,25}, Eomes¹⁹, zVegT¹⁹, TbxT¹⁹ and Tbx6) and signal mediator (β -catenin, Smad1, Smad2) binding levels across ~12,500 highly engaged pCRMs (compiled from the 2,000 pCRMs with the highest DNA occupancy levels detected per protein and developmental stage) over several developmental stages. Note that developmental time (arrow) separates these profiles best.”
- **Lines 134-139:** Context-dependent TF co-expressions affect the dynamics of chromatin recruitment
It took me a long time to find the orange frame, and now that I found it, I do not understand the point being made for orange and black frames. Nor do I understand the basis for “This observation supports context-dependent signal interpretation (top panel in Fig. 2c): “ Perhaps it would be useful to explain how the results would have differed in the boxes if there was not context dependent signal interpretation.
We have highlighted the sections of interest differently. They are now colored and numbered. We have also extended this text passage to explain our observations in more details: “With respect to the chromatin recruitment of TFs versus signal mediators, we note that, first, Smads and/or β -catenin were frequently detected at Sox3, Foxh1 or VegT binding sites (top 2,000 peaks shown in **Fig. 3f**) and, second, Smad- and/or β -catenin-bound pCRMs (top 2,000 peaks) were significantly enriched for SOX, POU-SOX, POU, FOXH and T motifs (red field #1 in **Fig. 3g**) suggesting that corresponding TFs affect the recruitment of these signal mediators. The reverse was much less the case as shown by the low significance of SMAD

and bHSH motif enrichments at TF-bound pCRMs (with the exception of Smad2 interactor Foxh1²³) (blue field #2 in Fig. 3g)".

- Lines 131-133: "The discovery and co-clustering of DNA motifs at occupied CRMs indicate that factor-intrinsic elements include the sequence-specificity of DNA binding domains, oligomerization tendencies (e.g. Brachyury binding palindromic motifs due to its propensity to form homodimers²²) and protein-protein interactions, while the developmental context reflects co-expressed TFs ". There has to be a clearer way of saying this, perhaps starting with better subjects for the sentence that fit with the verbs used? And I don't understand the second part of the sentence.

We have rewritten this section and added examples to make our points clear to the reader: "The hierarchical clustering of DNA occupancy levels for the selected factors at different developmental stages revealed specific DNA motif combinations. For example, pCRMs showing 'unique' binding of mVegT (that is, binding that is not shared with the other profiled factors) show a high frequency of T, OTX and SOX motifs (Supplementary Fig. 6)".

- Lines 142-156: Figure 2e is not described until much later, but even then it is not described in the legend or text enough to know what it means. Furthermore fig 2f appears to only have one significant box- where Sox3 is recruited to E boxes by MyoD. The recruitment of polII is expected from the previously documented effect of MyoD on transcription.

We analyse here the effect of ectopically expressed MyoD on focal RNAPII recruitment at CRMs and show an example of MyoD causing RNAPII elongation at the *actc1* gene. We have rewritten part of this paragraph: "To explore the suggested importance of TF co-expression on pCRM engagement, we ectopically expressed an HA-tagged version of the muscle determinant MyoD (MyoD-HA) in animal cap cells and profiled pCRMs for MyoD-HA as well as for endogenous Sox3 and RNAPII at the early gastrula stage (Fig. 4a and Supplementary Tables 1 and 6). MyoD was chosen because its canonical E-box recognition motif is normally not significantly enriched before or during gastrulation (green field #3 in Fig. 3g and Supplementary Fig. 5b) so its effect on chromatin engagement ought to be clearly discernible, while Sox3 and RNAPII were selected because they are ubiquitously expressed and represent sequence-specific and nonspecific DNA binding factors, respectively. The ectopic expression elevated the Spearman correlations of Sox3 and RNAPII with MyoD-HA (Supplementary Fig. 7a) and shifted the first and second PC of Sox3 and RNAPII toward MyoD-HA (Fig. 4b) suggesting that MyoD altered the binding of both Sox3 and RNAPII. Differential binding analysis on a genome-wide (heatmap in Fig. 4c) and locus-specific (pileup track in Fig. 4d) scale confirms that MyoD-HA co-recruits endogenous Sox3 and RNAPII to its gene targets like *actc1* and *myl1*. The E-box motif of MyoD-HA emerged as a significantly enriched motif of Sox3 and RNAPII binding, while MyoD-HA binding itself seemed to be influenced by endogenous TFs as judged by the developmental stage-characteristic enrichment of SOX, POU-SOX and FOX motifs at MyoD⁺ pCRMs (Fig. 4e). However, this opportunistic recruitment to non-canonical binding sites, such as MyoD-HA to functional Sox3 gene targets (e.g. *otx2* and *sox2* activated by Sox3-HA), did not affect transcription in animal caps (Fig. 4f-h)".

- Lines 152-154: I do not follow why "MyoD-HA binding itself was influenced by the endogenous TF expression profile" though admittedly I expect a major point like this should be in the main text, perhaps it is in the supplement. (Which I read later)- raising the question, Shouldn't the reader be able to follow the main paper without reference to the supplement? And raises the question of why so many essential figures are in the supplement? This one is hard enough to follow without having to jump to the supplement all the time.

We have moved some Supplementary Figures into the main part of the paper and explained the basis of this conclusion in more details. See above.

- Lines 106-107: Would it be good to explain the colors earlier, since people tend to read the legend from the beginning to the end. Is yellow a wise choice in h?

We have added a sentence at the end of the first paragraph to explain that the profiled developmental stages and chromatin factors are both consistently color-coded as illustrated in Fig. 3a: "The profiled developmental stages and chromatin factors are color-coded as illustrated in Fig. 3a and ChIP-Seq peak call coordinates are listed in Supplementary Table 5". We changed the color of the Sox3 pileup track in Fig. 5b from yellow to black.

- Lines 154-156: In this and the MyoD case, where Sox3 is a follower to another protein (MyoD), what impact does this have on the argument that Sox3 is a pioneer that establishes a competent chromatin state for signaling? Signal-induced regionalization of ZGA depends on maternal TFs.

We have not fully explored this yet. We show by means of a few 'opportunistic' gene targets (e.g. *otx2*, *sox2*) that MyoD-HA did not affect transcription of these Sox3 targets in animal caps (Fig. 4f-h).

- Line 348: I'm not seeing any transient expression of TFs in Fig. 2d, as referred to in the text.

We refer here to the translation of *pou5f3.3* and *mVegT* transcripts. Both transcripts are maternal and undergo post-MBT clearance.

- **Line 105:** If brachyury is spelled out, then shouldn't eomesodermin also be spelled out?
We have replaced *brachyury* by its new abbreviation *tbxt*
<http://www.xenbase.org/gene/showgene.do?method=display&genelid=478788&>
- **Lines 229-230:** "By comparison, unaffected CRMs rarely contained canonical POU/SOX motifs, and were strongly enriched for promoter-centric motifs of the Krüppel ZF, bZIP and NFY protein families" It is surprising that the Fox motifs are not in these classes- do all Fox sites coincide with Sox/Pou, or are they just at regions that are less accessible?
As FOXH (and T) motifs are more frequently associated with distal CRMs, they are found slightly enriched in affected CRMs. Interestingly, DNase-hypersensitive sites are not frequently associated with FOXH motifs (see also **Fig. 2b**). We have extended the **Fig. 8e** to include the occurrences of FOX and T motifs.
- **Discussion**
Lines 252-253: "Our results allow us to propose a model (Fig. 6) of pioneer-initiated chromatin remodeling, or priming, that unlocks context-specific CRMs, some of which contain signal responsive elements to enhance and regionalize transcription." Isn't this an enhancement of the previous models, and specifically that for FoxH1.
We are not quite sure to what the reviewer is referring to.
- **Lines 296-300:** It would be nice to have an explicit documentation of the pluripotency gene aspect- are Sox3 and Pou really pluripotency activities in this context, or just two of a set of various maternal pioneer factors that activate the zygotic genome?
Yes, they convey pluripotency. Most prominent example: their activity facilitates Nodal-mediated activation of *tbxt* (*brachyury*), an essential factor to generate neuromesoderm. We think this regulation of signal competence is conserved as *brachyury* induction also requires Sox2/Oct4 in human embryonic stem cells (unpublished data).
- On the competence angle, perhaps it would be worth mentioning the similar principle from the effect of the Zn finger protein that acts to terminate ectodermal competence to respond to mesoderm induction, but only appears to act, as here, through a subset of genes, specifically those that have p53 inputs.
We assume the reviewer is referring to the function of *znf585b* (originally known as XFDL156, *Cell* 2008). We are not sure whether it is a good comparison as *znf585b* is a zygotic gene and restricts *tbxt* expression (i.e. mesoderm induction) by inhibiting p53. Also the role of p53 in regulating mesoderm induction has not been confirmed in any other species than *X. laevis*. In fact, our own experiments in *X. tropicalis* did not find a requirement of p53 in regulating *tbxt* (unpublished results).

Reviewer #2:

Gentsch et al. investigate the role that maternal transcription factors play in establishing the cell type specific responses to inductive signals during early embryonic development. They tackle this important developmental question using the powerful *Xenopus tropicalis* model system. The authors use genomics, bioinformatics and developmental biology approaches to investigate how context-dependent signaling transduction is achieved. Starting with motif analysis within earliest activated CRMs, the authors identified and characterized the roles of maternal TFs, specifically mPouV/Sox3, in unlocking signal-responsive CRMs during *Xenopus* ZGA.

A major strength of the manuscript is the very large number of original and relevant experiments (ChIP-seq, RNA-seq, Capture C, DNase-seq) done in primary *Xenopus* tissues at one or more developmental stages as well as after several perturbation experiments. They anchor their study at the mid-blastula embryo (4096 cells) which shows high levels of homogeneity, little transcription and diversification.

A major weakness of this paper is that it is dense and challenging to read. Specifically, the paper could benefit from a simpler, more systematic approach to data integration and data summary.

I found the manuscript challenging to read and hence review. At an initial and superficial, yet practical level, I would have preferred a more conventional format, rather than a mocked-up publication format with tiny multi-panelled figures. For example, a conventional format makes it easier to identify specific lines, pages for comments (For this reason I will include quotations in my review to highlight the text I am referring to).

While it was clear to me that great effort was made to produce visually appealing figures in both the main text and supplemental material, there are many figure panels where reading the figure legend and looking at the figure was not enough to understand what was being shown.

Moreover, when reading the main text that referred to the figures, it was often challenging to sift through the jargon and understand the underlying logic, details, and caveats that led the authors to so confidently state their conclusions. In my opinion, the authors often seem to state their conclusions and then give reference to a Figure panel or Supplemental figure panel without providing enough details or context to appreciate how their experiments

lead them to their conclusions (I will try and provide some examples below however this occurs throughout the paper).

I found myself having to rely on the methods to understand what analyses were performed. In many cases there was not enough information to assess whether the data the authors presented supported their conclusions. While I appreciated the detailed experimental methodology, I found that the bioinformatics analyses were not sufficiently described.

Specific Comments:

- **Lines 62-81:** At the beginning of the results section the authors mention that RNAPII is a reliable and objective indicator of CRM activity. I agree with this. However, I don't think one can simply identify CRMs using RNAPII ChIP-seq and then say that these RNAPII regions themselves are, by definition, active CRMs. I could not find details about the number of these putative 'CRMs' (i.e. the average length, and evidence of where they are located, their overlap with other data from Mod ENCODE or the literature was not given). How many, and by what criteria, were putative CRMs assigned as being obscured by RNAP2 occupancy at promoters and gene bodies? Since these CRMs form the basis of much of the analyses in the paper I think these details need to be more transparent.

We introduced 'putative' to make clear to the reader that CRM activity has not been tested (putative CRM, pCRM). We agree that 'active' does not portray these pCRMs correctly and replaced it by 'RNAPII⁺'. We have generated new figures (**Supplementary Fig. 1a,d,e**) and extended the main text to address some of the reviewers questions: "In an effort to understand how early chromatin dynamics influence the recruitment of signal mediators to the genome (**Fig. 1b**), we first identified ~27,000 pCRMs from the 32-cell to the late gastrula stage by mapping focal RNA polymerase II (RNAPII) depositions on a genome-wide scale by means of ChIP-Seq (**Fig. 1c** and **Supplementary Tables 1** and **2**). RNAPII has no DNA sequence preference and its chromatin engagement is a reliable and objective indicator of pCRM usage. The number of RNAPII-engaged (RNAPII⁺) pCRMs increased from ~650 at the 32-cell stage to >10,000 at the 1,024-cell and later developmental stages (**Supplementary Fig. 1a**). The largest changes to RNAPII⁺ pCRMs, as calculated by pairwise Spearman correlations, were detected between the 1,024-cell stage and the MBT (**Fig. 1d**), with most pCRMs being engaged only transiently before MBT (e.g., 6,145 from the 128-cell to the 1,024-cell stage) and more persistently after MBT (**Supplementary Fig. 1a**). The analysis of enriched DNA motifs among RNAPII⁺ pCRMs suggests that pre-MBT recruitment is predominantly directed by members of the FOXH, POU, SOX and T domain TF families (**Fig. 1e** and **Supplementary Fig. 1b**).

The discovery of RNAPII⁺ pCRMs is difficult in promoters and gene bodies where extended RNAPII depositions associated with transcript elongation might hamper the correct detection of RNAPII⁺ pCRMs. MBT-staged pCRMs were therefore further characterized for chromatin accessibility and for the enhancer-associated histone mark H3K4me1. In our hands, the high yolk content in early *X. tropicalis* embryos made it impossible to use transposition¹¹ to probe chromatin accessibility. Instead, we used an approach involving DNase I mediated digestion followed by deep sequencing (DNase-Seq), in which we selected small accessible fragments of DNA (see **Methods** and **Supplementary Fig. 1c** for exemplar comparison with other chromatin features). We detected ~17,500 accessible (DNase hypersensitive) pCRMs, ~85% of which showed both RNAPII and flanking H3K4me1 above background (**Fig. 1f**, **Supplementary Fig. 1d** and **Supplementary Tables 1** and **3**). About 29% and 31% of accessible pCRMs (compared with ~16% and ~38% of pCRMs defined by RNAPII peak calling) were found in promoters and gene bodies, respectively (**Supplementary Fig. 1e**)."

Supplementary Table 2 lists all ~27,000 RNAPII⁺ pCRMs detected from the 32-cell to the late gastrula stage. This Table also includes annotation, DNA occupancy levels and DNA motif occurrences.

- **Lines 67-69:** It was mentioned that largest changes in RNAPII recruitment to CRMs occur between 1024 stage and MBT. While the main text and supplemental exemplary genome browser views suggest the data is of good quality, little information is given for the data comparisons made between developmental stages. **Fig. 1d** shows a Spearman correlation matrix of CRM-bound RNAPII from the 32-cell stage to the late gastrula stage. We have generated a new figure (**Supplementary Fig. 1a**) that shows CRM usage in more details.
- **Line 334:** **Fig. 1e** shows a summary of the differential motif enrichments at the different developmental stages. It is not clear to me how this analysis was done. I see that "Z-scores of motif enrichments at RNAPII peaks were calculated using ISMARAv1.2.281 (<https://ismara.unibas.ch/fcgi/mara>) in expert mode (**Fig. 1e**). and "Heatmaps of relative tag densities were hierarchically clustered (Ward's method) using the R package seriations". Without knowing which peaks went into the analysis one cannot reproduce this analysis easily or properly understand what was done. Looking into ISMARA, expert mode requires the user to put in specific intervals/gene names as well as motif matrices. It is not clear what exactly was used to derive the motifs (e.g. were all matrices from a collection used?).

We provide these files as part of the programming code (now available on GitHub at <https://github.com/gegentsch/SignalCompetence>)
/xenTro71/supplementary_files/
pol2_st6to12_crm.bed
pol2_tagCount_ismara.txt.gz
pol2_motifCount_ismara.txt.gz

We also provide our own shortlist of position weight matrices used to find DNA motif occurrences in RNAPII⁺ pCRMs (/xenTro71/supplementary_files/germ_layer_motifs.txt).

For more detailed information read R script "1_pol2_ismara.R".

We have generated a new Supplementary Table that lists all ~27,000 RNAPII⁺ pCRMs and DNA motif occurrences used for the ISMARA analysis (**Supplementary Table 2**).

- **Line 504:** The supporting **Supplementary Fig. 1b** provides additional information. However, **Supplementary Fig. 1b** legend does not appear to explain what the green and black circles mean in terms of number of CRMs and background. I see in the methods section that Top N peaks were used in some analyses however there does not seem to be information about what was done for **Supplementary Fig. 1b**.
We used each 5,000 pCRMs with the highest RNAPII binding levels detected from 32-cell to 1000-cell stage (pre-MBT) and from MBT to late gastrula stage (MBT+), respectively. We have added this information to the figure legend and the online methods. We also added to the figure legend that this analysis included 39,785 (pre-MBT) and 41,898 (MBT+) genomic 'background' regions matching overall GC contents of the selected RNAPII+ pCRMs. The green and black circle represents the percentage of the RNAPII+ pCRMs and background regions containing the indicated motif, respectively. We have changed the in-figure legends to make this clear to the reader.
- The supplementary Table 1 was the only place where the experimental design with the number of biological replicates was presented. I think the authors should mention in the main text as well the number of biological replicates used for each experiment (this is done for the LOF experiments). I did not request/obtain the token to access and verify the GEO submitted data, however given the minimal analyses and description of the data generated, I think it will be helpful for the authors to deposit both raw, and aligned data, as well as the identity of the various CRMs so that others can easily look at the data and reproduce the analyses.
We submitted our computational code to fully disclose our post-sequencing analysis. As requested we also submitted both raw and aligned data to GEO archive. We have generated additional Excel spreadsheets with the genomic coordinates and additional metadata of the various pCRMs (**Supplementary Tables 2, 3, 5, 6, 7, 8, 11 and 12**). We have added the number of biological replicates to the figure legends.
- **Lines 72-81:** In the second paragraph of the results, the authors acknowledge the limitations of using RNAPII for de novo CRM discovery. They go on to use DNase hypersensitivity and H3K4me1 ChIP-seq to show the regions with RNAPII activity within genes at the MBT stage are in accessible chromatin. Are there DNase plus H3K4me1 enriched regions that gain RNAPII later in development? To what extent would performing DNase-seq at the pre-MBT stage inform about the pioneering activity of the maternal TFs (e.g. would the authors expect open chromatin shared between pre-MBT and MBT that shows a gain of RNAPII activity at the MBT stage (or later) to be examples of chromatin opened by maternal pioneer factors?
Yes, 1,951 DNase hypersensitive sites detected at MBT seem to gain RNAPII during gastrulation. However, our chromatin profiling of Pou5f3/Sox3 loss-of-function embryos suggests that chromatin accessibility has an immediate effect on RNAPII recruitment. Thus, we think that additional factors may be involved to attract more RNAPII. Certainly, a temporal profile of chromatin accessibility would reveal whether open chromatin and RNAPII engagement share similar dynamics and may provide an answer to the question of whether simultaneous or sequential gain of chromatin accessibility and RNAPII are prime examples of maternal pioneer activity. More knowledge is required about the efficiency and temporal windows of maternal pioneering activity to dissect the spatio-temporal usage of pCRMs.
- **Line 362:** It would be useful to integrate the ChIP-seq data from the 1024 cell stage (I see β -catenin ChIP, Sox3 ChIP, VegT ChIP data from 1000 cell stage listed in Table 1). To what extent are the pioneer factor bound regions already bound by inductive TFs at the 1000 cell stage? What fraction of Sox3 or VegT bound regions are not considered to be part of RNAP2 defined CRMs at the 1000 cell stage? Do RNAP2 negative Sox3/VegT bound regions at the 1000 cell stage become sites of later TF or RNAP2 usage? The answers to these questions should help support the authors definition of CRMs.
New **Fig. 3c** shows the binding levels of RNAPII at TF⁺ and signal mediator⁺ pCRMs.

- Line 91:** I think the use of ribosome footprinting and proteomics to identify relevant transcription factors was well justified and explained. Given the literature, would the authors have expected to have concluded that Pou5f3, Sox3 and mVegT would be the pioneer factors to study further?
Based on current literature (e.g. Lee *et al.*, 2013, doi:10.1038/nature12632; Zhang *et al.*, 1998, doi.org/10.1016/S0092-8674(00)81592-5), one could speculate that these are relevant pioneer factors to activate the zygotic genome in the frog *Xenopus tropicalis*. However, we provide experimental evidence that these TFs are frequently translated proteins and likely pioneer factors (and potential competence factors) based on ribosome footprinting and chromatin profiling.
- Line 348:** At first glance the model in Fig. 2d is visually appealing. However, for this reviewer the components of the figure were not sufficiently explained. The figure was said to be derived from the authors previously published data derived from 6 references. Which data from each reference were used is not clear. Perhaps this kind of figure would make sense in the discussion or review article, assuming it was properly explained.
We think that this Figure is important as it shows where and when the proteins of interest are detected in the nucleus. We have changed the figure legend to better explain the graphical illustrations, which includes the direct link of references to individual TFs or signal mediators.
- Lines 134-139 and line 362:** The authors tend to present their conclusion/interpretation without clearly describing their observation/results first, which is often confusing. Figures like Fig. 3g appeared multiple times in the manuscript, so it may be worthwhile the authors to take the time to explain it when it first appears. Given the several orange elements in the figure it took me a moment to notice the gold box which I presume is the orange frame that is being referred to. I have a similar comment about the black frame. This may simply be a function of the small figure size. Fig. 3g seems like an expected result where most TF bound regions were enriched for their expected motif. I would also expect that signal mediator TFs would bind regions bound by the other maternal TFs. It would be helpful for the authors to refer to expectations coming from previous studies and highlight what was already known, what was expected and what was novel.
We have relabelled parts of Fig. 3g and rewritten this paragraph to better highlight and explain these findings. "With respect to the chromatin recruitment of TFs versus signal mediators, we note that, first, Smads and/or β -catenin were frequently detected at Sox3, Foxh1 or VegT binding sites (top 2,000 peaks shown in Fig. 3f) and, second, Smad- and/or β -catenin-bound pCRMs (top 2,000 peaks) were significantly enriched for SOX, POU-SOX, POU, FOXH and T motifs (red field #1 in Fig. 3g) suggesting that corresponding TFs affect the recruitment of these signal mediators. The reverse was much less the case as shown by the low significance of SMAD and bHSH motif enrichments at TF-bound pCRMs (with the exception of Smad2 interactor Foxh1²³) (blue field #2 in Fig. 3g)".
In the Discussion, we also refer to two key publications (Mullen *et al.*, 2011; Tromouki *et al.*, 2011) that show *in vitro* the importance of cell lineage regulators in directing the chromatin engagement of signal mediators and thereby generating cell type-specific signal responses.
- Lines 134-139:** For example, for Fig. 3g, which is one of the key figures that support the author's argument, the authors directly brought up their interpretation that "maternal TFs affected the recruitment of signal mediators (see orange frame in Fig. 3g), but (with the exception of Foxh1, which interacts directly with Smad2) the reverse was much less the case". The logic could be clearer if the author provides more description and context for these results. For example, "Enrichment of maternal TF motifs were observed within the binding sites of signal mediators, however the reverse was not seen. This indicates that maternal TFs are likely to affect the recruitment of signal mediators, just as developmental lineage-specific TFs are known to recruit signal mediators later in development (insert representative references from the literature)".
We have rewritten this section and the figure legend to better explain Fig. 3g. See quoted text above.
- Lines 108-115:** This paragraph is dense and hard to follow: "Maternal TFs and signal mediators shared many chromatin characteristics: (1) DNA occupancy levels followed a log-normal distribution with only a few hundred RNAPII-transcribed zygotic genes receiving high and super-enhancer-like²¹ input (Supplementary Fig. 2a-c). (2) From the 1,024-cell to the late gastrula stage, frequent chromatin recruitment occurred on average to ~800 super-enhancers which had zygotic genes nearby (<5kb) that are primarily involved in early embryonic processes including germ layer and body axis formation (Supplementary Fig. 2d). (3) Among all zygotic genes, promoter-proximal regions were most consistently bound (Supplementary Fig. 2e). All these characteristics were also observed for chromatin accessibility and RNAPII engagement at CRMs (Supplementary Fig. 2c,e)."
We have rewritten part of this paragraph: "Maternal and zygotic TFs and signal mediators shared many chromatin characteristics: (1) DNA occupancy levels followed a log-normal distribution with <1,000 RNAPII-transcribed genes receiving high and super-enhancer-like²¹ input (i.e. clusters of occupied pCRMs separated by ≤ 25 kb). Similar distributions were also observed for chromatin accessibility and RNAPII engagement (see examples in Fig. 3a and Supplementary Fig. 3a,b and systematic analysis in Supplementary Fig. 4a). (2) TF-bound super-enhancers were linked through the gene ontologies of

nearby zygotic genes (≤ 5 kb) to early embryonic processes including germ layer and body axis formation (Fig. 3b). (3) The binding sites of TFs and signal mediators were also frequently defined by RNAPII deposition (Fig. 3c). (4) Among all zygotic genes, promoter-proximal regions were more consistently bound than other pCRMs (Supplementary Fig. 4b).”

- Lines 116-133: The following paragraph is also hard to follow. It seems like the findings here are important for the paper but I struggle to make sense of the writing. I think the factor-intrinsic and developmental context-dependent jargon which emerges here generates some ambiguity. Throughout the manuscript TFs are classified using several terms. It would be helpful to have these terms defined clearly once, and then consistently used throughout. For example: pioneer factors, co-expressed maternal TFs, factor-intrinsic TFs, development context-dependent TFs, and Signal TFs. “The comparison of DNA occupancy levels among and between signal mediators and TFs suggests that the recruitment of both depends on factor-intrinsic and developmental context-dependent elements (Supplementary Fig. 3a,b).”....
...Conversely, signal-initiated recruitment fails to generate cell type-specific responses (bottom panel in Fig.2c).

We agree with the reviewer on the ambiguous usage of scientific terms. Thus, we have rephrased these findings. “Differences in DNA occupancy levels at different developmental stages, compared by pairwise Spearman correlations and principal component (PC) analysis, suggest that the recruitment to chromatin of signal mediators and TFs is driven both by their individual properties and by the developmental stage (Fig. 3d and Supplementary Fig. 5a). For example, chromatin recruitment of mVegT at stages of pluripotency resembled that of other sequence-specific factors at the same developmental stage but differed from the binding of its zygotic isoform and of related T-box TFs at later developmental stages (highlighted in Supplementary Fig. 5a). The importance of developmental stage in driving patterns of chromatin recruitment was further revealed by the changing DNA binding patterns of sequence-nonspecific RNAPII to pCRMs (Fig. 3e).

The identification of enriched DNA recognition motifs at occupied pCRMs confirmed known TF/signal mediator properties such as the sequence-specificity of their DNA binding domains, oligomerization tendencies and protein-protein interactions (Supplementary Fig. 5b). For example, in contrast to other T-box TFs, Tbx1 recognizes palindromic T motifs due to its propensity to form homodimers²²; and Smad2 chromatin recruitment is frequently associated with the FOXH motif because Smad2 interacts with Foxh1²³ (Supplementary Fig. 5b). Other DNA motifs such as the POU or POU-SOX motifs were consistently co-enriched from the 1,024-cell to the late gastrula stage in most binding profiles. This is indicative of the pluripotent state, a developmental context associated with co-expression of Pou5f (Oct4) and SoxB1 (e.g. Sox2 or Sox3) proteins as previously observed *in vitro*²⁴ (Fig. 3g and Supplementary Fig. 5b).

The hierarchical clustering of DNA occupancy levels for the selected factors at different developmental stages revealed specific DNA motif combinations. For example, pCRMs showing ‘unique’ binding of mVegT (that is, binding that is not shared with the other profiled factors) show a high frequency of T, OTX and SOX motifs (Supplementary Fig. 6).”

- Line 560: When I look at Supplementary Fig. 5a and the associated legend this conclusion is not immediately apparent. Which data and comparisons of ‘DNA occupancy levels’ went into this conclusion? We have introduced an example in maternal VegT (mVegT) to explain our statement. “For example, chromatin recruitment of mVegT at stages of pluripotency resembled that of other sequence-specific factors at the same developmental stage but differed from the binding of its zygotic isoform and of related T-box TFs at later developmental stages (highlighted in Supplementary Fig. 5a)”. As mentioned in the main text, we have also highlighted (black L-shaped box) the corresponding Spearman correlations between mVegT and all other profiled TFs and signal mediators in the Supplementary Fig. 5a.
- Lines 123-130: This paragraph continues to be challenging to follow: “The discovery and co-clustering of DNA motifs at occupied CRMs indicate that factor-intrinsic elements include the sequence specificity of DNA binding domains, oligomerization tendencies (e.g. Brachyury binding palindromic motifs due to its propensity to form homodimers) and protein-protein interactions, while the developmental context reflects co-expressed TFs (Fig. 2b and Supplementary Fig. 3d,e).” It seems to me to be saying something obvious about motifs at CRMs (TFs have DNA binding motifs, they bind together at CRMs, and in doing so may interact with other proteins). However, the sentence is making a contrast of this idea to ‘developmental context’. It is not clear how the three figure panels referred to support the arguments in this paragraph. This has been rephrased as part of complete overhaul of the entire paragraph: “The identification of enriched DNA recognition motifs at occupied pCRMs confirmed known TF/signal mediator properties such as the sequence-specificity of their DNA binding domains, oligomerization tendencies and protein-protein interactions. For example, in contrast to other T-box TFs, Tbx1 recognizes palindromic T motifs due to its propensity to form homodimers²²; and Smad2 chromatin recruitment is frequently associated with the FOXH motif because Smad2 interacts with Foxh1²³ (Supplementary Fig. 5b). Other DNA motifs such as the POU or POU-SOX motifs were consistently co-enriched from the 1,024-cell to the late gastrula stage in most binding profiles. This is indicative of the pluripotent state, a developmental context associated with co-expression of Pou5f (Oct4) and SoxB1 (e.g. Sox2 or Sox3) proteins as previously observed *in vitro*²⁴ (Fig. 3g and Supplementary Fig. 5b).

The hierarchical clustering of DNA occupancy levels for the selected factors at different developmental stages revealed specific DNA motif combinations. For example, pCRMs showing 'unique' binding of mVegT (that is, binding that is not shared with the other profiled factors) show a high frequency of T, OTX and SOX motifs (**Supplementary Fig. 6**.)”

- **Line 362:** Fig. 3a is labelled with “scoring the enrichment of DNA motifs at occupied CRMs however the figure does not have any scores or motifs on it.
We have removed this from the figure.
- The model referred to in Fig. 2c is needs to be clarified and would be better suited for the discussion. We have removed this figure. In the Discussion, we refer to two key publications (Mullen *et al.*, 2011; Tromouki *et al.*, 2011) that show *in vitro* the importance of cell lineage regulators in directing the chromatin engagement of signal mediators and thereby generating cell type-specific signal responses.
- In the latter half of this manuscript, the authors focused on maternal factor mPouV/Sox3 and demonstrated how Nodal signalling-initiated transcription depends on pioneer activities of these maternal TFs. However, since Sox3 and Pou5f3 are distributed ubiquitously in early embryos (according to the author), it does not seem to be a convincing example to show how pioneer factors leads to regionalized transcription. To me, it did not directly answer to the concept raised in introduction. The authors should address this more clearly in their discussion and what is already known in the literature.
Sox3 and Pou5f3 by themselves do not or only weakly activate signalling targets such as the marginal zone-specific expression of *tbxt*. In this and other instances, they are competence factors and open the chromatin landscape for signal mediators to act on. Thus, they facilitate regionalized transcription. We explain this in the Discussion. To our knowledge, these chromatin dynamics of signal competence have not been explored anywhere else in the literature yet.
- **Lines 142-156:** The MyoD-HA experiment is interesting although the logic for deploying a lineage-specific pioneer factor MyoD at the MBT stage could be explained a bit more.
We have changed some of the sentences and bits of **Fig. 4b** to better explain the logic behind this experiment. “To explore the suggested importance of TF co-expression on chromatin engagement, we ectopically expressed an HA-tagged version of the muscle determinant MyoD (MyoD-HA) in animal cap cells and profiled pCRMs for MyoD-HA as well as for endogenous Sox3 and RNAPII at the early gastrula stage (**Fig. 4a** and **Supplementary Tables 1** and **6**). MyoD was chosen because its canonical E-box recognition motif is normally not significantly enriched before and during gastrulation (green field #3 in **Fig. 3g** and **Supplementary Fig. 5b**) so its effect on chromatin engagement ought to be clearly discernible, while Sox3 and RNAPII were selected because they are ubiquitously expressed and represent sequence-specific and nonspecific DNA binding factors, respectively”
- **Lines 152-156:** It's not clear how **Fig. 4c,e** showed that "MyoD-HA binding itself was influenced by the endogenous TF expression profile. The author should elaborate more and explain what an endogenous TF expression profile means. How this connects to the model **Fig. 10d** needs to be clarified. Isn't it expected that MyoD can be a pioneer factor which recognizes its own motifs and can also bind to existing open chromatin with weaker motifs? Also **Supplementary Fig. 4e** mentions Sox3-HA whereas Sox3-HA is not mentioned in the main text.
We have changed and extended several sentences to elaborate this. “The E-box motif of MyoD-HA emerged as a significantly enriched motif of Sox3 and RNAPII binding, while MyoD-HA binding itself seemed to be influenced by endogenous TFs as judged by the developmental stage-characteristic enrichment of SOX, POU-SOX and FOX motifs at MyoD⁺ pCRMs (**Fig. 4e**). However, this opportunistic recruitment to non-canonical binding sites, such as MyoD-HA to functional Sox3 gene targets (e.g. *otx2* and *sox2* activated by Sox3-HA), did not affect transcription in animal caps (**Fig. 4f-h**).”

Endogenous is a common biological term to point out that the subject originates from within the organism. We use MyoD to substantiate the suggested effect of TF co-expression (based on the detected co-enrichment of several non-canonical DNA motifs such as SOX and OTX at mVegT⁺ pCRMs) on chromatin engagement in general. We explain the connection to our model in **Fig. 10d** in the Discussion.

- The details regarding the intron and exon analyses of RNA-seq data was not clear. The description of the results from the many RNA-seq experiments performed could also be clearer.
The details about intron/exon RNA profiling are outlined in the Online Methods. We have added all the controls for each RNA-Seq experiment to ‘Quantification and statistical analysis’. Their exact computational implementation can be read in the provided R script. We are not sure what else is unclear to the reviewer.
- **Lines 220-238:** The authors perform Capture-C and however the experimental details and explanation of results were minimal.
We have introduced more experimental details about next-generation capture-C and slightly extended the results. “To discover how maternal TFs allow cell type-specific genes to be signal-induced, we compared

various chromatin features from genome-wide accessibility (DNase-Seq; **Fig. 8a** and **Supplementary Fig. 12a**) and DNA occupancy (ChIP-Seq for H3K4me1, RNAPII, Smad2 and β -catenin; **Fig. 8a**) to high-resolution conformation contacts (next-generation capture-C; **Fig. 9a**, **Supplementary Fig. 12b** and **Supplementary Table 10**) of 30 selected promoters (**Fig. 9b**) between control and mPouV/Sox3 LOF embryos at MBT (**Supplementary Table 1**). ... The loss of accessibility triggered further profound changes to chromatin *in situ* ... in contrast to unaffected pCRMs, 32 pCRMs with compromised accessibility—most of which are part of super-enhancers—also showed significantly (FDR $\leq 10\%$) reduced promoter contacts (e.g., *tbxt*, *foxb1*, *cdc25b* and *zic1*) (**Figs. 9c,d** and **10a**, **Supplementary Figs. 13** and **14** and **Supplementary Table 12**).

Supplementary Table 12 lists the changes to promoter contacts upon mPouV/Sox3 LOF. Further details about next-generation capture-C are outlined in the Online Methods.

- **Line 468:** The $p\Delta$ was used in **Fig. 9d** but I could find little information how it was calculated for the Capture C data. Are the regions shown significant? Providing actual statistics for the specific examples given in the paper would be helpful.

The calculation of $p\Delta$ is outlined in the Online Methods 'Differential analysis of chromatin features'. "Raw tag counts in sliding (+200 bp) windows of 400 bp (accessibility, H3K4me1, RNAPII, Smad2 and β -catenin) or 2 kb (RNA and promoter contacts) across the genome were processed to estimate p-values of chromatin changes ($p\Delta$) using DESeq2. Read count dispersions were locally fitted. The shrinkage of \log_2 fold changes, threshold on Cook's distances and independent filtering were all switched off." This has been implemented in the R function `pValueMatrix()` to be found under `R_utils/p_value_matrix.R` on GitHub at <https://github.com/gegentsch/SignalCompetence>. We have provided the statistics for changes in chromatin accessibility and chromatin conformation in new Excel spreadsheets (**Supplementary Table 12**) and added the significance threshold of FDR of $\leq 10\%$ to the main text.

- **Legend for Fig. 8f,g** are minimal. What is the definition of the "mPouV/Sox3-dependent ZGA"?
We have modified the legend of **Fig. 8f** and changed **Fig. 8g**: "**Figure 8** ... (f) Pie charts summarize the distribution of distances (kb) to nearest zygotic TSSs of affected (top pie chart) and unaffected (bottom pie chart) CRMs. (g) Panel compares the effect of mPouV/Sox3 LOF on chromatin accessibility and RNAPII-mediated zygotic gene expression. Scatter plot shows the localization of accessible CRMs (affected, dot colored in orange to red with FDR decreasing from 10%; and unaffected, grey dot) relative to the zygotic TSSs that are active by the MBT and produce enough RNA transcripts to show significant \geq two-fold reductions upon α -amanitin injection (**Fig. 6e**). Gene loci are sorted according to the strongest reduction in transcript level upon mPouV/Sox3 LOF as shown in the log-scaled bar graph to the right."
- **Lines 238-240:** Evidence behind and the relevance of this conclusion written in the results section was not clear. Did the authors look at transcription of CRMs as well as RNAP2 occupancy?: "At the transcriptional level, lower usage of CRMs coincides with the reduction of coding as well as local non-coding RNA
Yes, we looked at both RNA and RNAPII tags that align with pCRMs. "... at the transcriptional level, lower usage of (clustered) pCRMs coincided with the reduction of coding (**Fig. 8g**) as well as local non-coding RNA (last column in **Fig. 8d** and 'low RNA' track in **Fig. 9d** and **Supplementary Figs. 13** and **14**)."
- **Lines 240-244:** "Importantly, the differential profiling explained how chromatin predetermines signal induction and why, for instance, Nodal-induced transcription of brachyury was strongly affected by the loss of mPouV/Sox3 and that of *Eomes* less so." It seems abstract to say that "differential profiling" can explain a concept. From looking at the data in **Fig. 8d** it seems that CRMs undergoing a (significant?) reduction in accessibility after mPouV/Sox3 LOF coincided with the loss of inductive TF binding. More explanation here would be helpful.
We have changed this section: "Importantly, the profiling of LOF embryos revealed how chromatin predetermines signal induction and why, for instance, Nodal-induced transcription of *tbxt* was strongly affected by the loss of mPouV/Sox3 and that of *eomes* less so. In contrast to those of *eomes*, all promoter-tied pCRMs (super-enhancers) of *tbxt* contain canonical POU/SOX motifs, so were not accessible to Smad2 in mPouV/Sox3 LOF embryos (**Fig. 10a,b**). Thus, Smad2 interactions with critical Nodal responsive CRMs of *eomes* remained intact, but those of *tbxt* were impeded by compacted chromatin".
- **Lines 227-229:** "Sorting of these CRMs according to the significance of lost accessibility reveals that chromatin opening depends on the pioneering activity of mPouV and Sox3 to recognize their canonical motifs in compacted chromatin (**Fig. 5c**)." Here is it more accurate to say "lost accessibility suggests that chromatin opening depends on ..."
We have changed this as proposed by the reviewer.
- **Lines 230-232:** "At the extreme end of affected loci (e.g. brachyury, *foxb1*, *cdc25b* and *zic1*) entire super enhancers became accessible through mPouV/Sox3 activity (**Fig. 5f** and **Supplementary Fig. 7g**)." In this

case what is the evidence that a super enhancer became accessible through mPouV/Sox3? Also my understanding was that mPouV/Sox3 activity was knocked down (not activated) in this experiment? What factor(s) were used to create the super enhancers? It would be odd to use RNAPII to build super enhancers as clusters of RNAPII would surely indicate stretches of transcribed genes. I may have missed this in the methods but if the authors use RNAPII to generate super enhancers, more justification is needed.

Super-enhancers were defined by stitching together CHIP-Seq peaks that are ≤ 25 kb apart. Subsequently, super-enhancers ≤ 5 kb from TSS apart were associated with zygotic genes. We used individual profiles of TFs and signal mediators to find super-enhancers. For gene ontology term enrichment analysis, we combined these super-enhancers according to the developmental stage of profiling (see **Supplementary Table 6** for genomic coordinates). The examples listed (e.g. *tbxT*, *foxb1*, *cdc25b* and *zic1*) are associated with such super-enhancers or clusters of CRMs occupied by multiple factors. We have added the definition of a super-enhancer at the first time of use in the main text (line 108: "i.e. clusters of occupied CRMs separated by ≤ 25 kb"). We changed the sentence with regard to the accessibility loss of super-enhancer upon mPouV/Sox3 LOF: "At the extreme end of affected loci (e.g. *tbxt*, *foxb1*, *cdc25b* and *zic1*) entire or large proportions of super-enhancers became inaccessible upon mPouV/Sox3 LOF (**Figs. 9d** and **10a** and **Supplementary Figs. 13** and **14**)."

- At the end, the author proposed two models of Nodal gene activation based on their dependency on mPouV/Sox3, with eomes and brachyury as examples. The authors suggest the mechanistic differences underlying their mPouV/Sox3 dependency is whether SOX/POU motifs exist within the enhancer regions of the Nodal genes. It will be more convincing if the author shows this in a systematic way, for example divide the Nodal genes into mPouV/Sox3 dependent and independent categories (as in Fig. 4b), analyzing their enhancer regions to see if one category enrich for SOX/POU motifs while the other category does not. This analysis may also reveal other pioneer TFs that are important for Nodal signaling.

We have generated new graphs (**Fig. 8g** and **Supplementary Figs. 15-17**) showing that mPouV/Sox3-induced chromatin accessibility is required for the expression of Wnt-, Nodal- or BMP responsive and non-responsive genes. The striking correlation of CRM accessibility loss by mPouV/Sox3 LOF and the occurrence of SOX/POU motifs is shown in **Fig. 8e**. Further dissection of signal responsive CRMs requires more LOF experiments and genetic manipulations, which is beyond the scope of this paper.

- With all the RNA-seq and CHIP-seq data the author generated, the author should be able to look at the BMP and WNT targeted genes in a similar manner as what they did to the Nodal genes, dividing them into mPouV/Sox3 dependent and independent and characterizing the enhancer motif enrichment for each categories.

See answer to previous comment.

- All loss of function models were achieved through MO knock-down or small molecular inhibitor, which could lead to non-specific defects. Nevertheless, the author showed good validation of their MOs, including (1) western blot or WMIHC demonstrating MO-mediated protein reduction; (2) RNA rescue at morphological or transcriptional level.

The pharmacological inhibitors of Nodal and BMP signaling generate morphological phenotypes that are reminiscent of morpholino-mediated depletion of corresponding signal receptor ligands. For example, the depletion of multiple BMP ligands (Reversade *et al.*, 2005) and the treatment with the BMP-inhibitor LDN193189 cause the same axial elongation failure. In addition, treatment with these pharmacological inhibitors and MOs, which were verified previously or we produced at least partial rescues for this paper, did not affect gene groups beyond the biological processes associated with verified targets. For these reasons, and bearing in mind that indirect effects from gene activation cascades are minimal at our developmental stage of interest, we think that off-target gene mis-regulations will be very few.

- Mutating one or a few POU/SOX motifs that are found within the CRMs of brachyury would be a relevant experiment. If the POU/SOX motif mutation itself leads to loss of accessibility, reduced active histone markers and downregulation of brachyury, should help support the model proposed by the authors.

We agree with the reviewer that this experiment could be very informative. We are planning to introduce targeted point mutation in the near future to investigate the code of signal competence. However, because such experiments require at least one year to generate *X. tropicalis* pCRM mutants, we think that this is beyond the scope of this paper.

- Supplementary Fig. 6a, The color legend, graphics, and labelling is confusing. For example, in the part of the figure referring to the 'pluripotency' stage how does the 'Opportunist' right branch of the panel relate to the results in the paper? What proportion of maternal binding of Pou5f3/Sox3 is opportunistic? What maternal TFs would be considered to fall in the "other pioneer factor (Developmental context)" during the pluripotency stage? The figure suggests these are bound to open chromatin established prior to Pou5f3/Sox3. Is this correct?

We presume that the reviewer refers to main Fig. 6 (now **Fig. 10d**). This Figure shows a working model of how we think chromatin remodelling facilitates signal competence. The opportunistic branch is substantiated by our findings that mPouV/Sox3 binds pCRMs without canonical motifs and that, for

instance, the ectopic introduction of MyoD causes Sox3 (and other factors such as RNAPII) to be co-recruited to canonical MyoD⁺ pCRMs. We have added some potential candidates with suspected pioneering activity, e.g. FoxH1 and Klf4. With regard to the timing of pCRM recruitment, we suspect that canonical binding is first required prior to opportunistic binding (see results of ectopic MyoD expression). Nevertheless, it is conceivable that pioneering and opportunistic activity may occur near-simultaneously.

- **Lines 18-19:** The abstract summary “Our work identifies significant developmental principles that inform our understanding of tissue engineering and tumorigenesis.” does not accurately reflect the analysis and work that was done in the manuscript. While this argument was briefly formulated in the discussion, I do not think it fairly represents the content of the work described and I think a summary which is more in line with the data and data analysis is warranted.
We agree with the reviewer and have removed the link to tissue engineering and tumorigenesis in the abstract summary.
- The authors note in the abstract that: “The remodeling includes the opening and marking of thousands of regulatory elements, extensive chromatin looping, and the co-recruitment of signal-mediating transcription factors.” While Capture-C data is presented in the manuscript, little information is given about chromatin looping changes during early development was given aside from two important example genes.
We have moved some of the Supplementary Figures containing capture-C data to main Figures.

Reviewer #3

In this very nicely written manuscript, the authors use ChIP-Seq of RNA PolII during *Xenopus* development to identify putative regulatory elements driving the maternal to zygotic transition, and then characterise these elements using numerous additional datasets, including ChIP-seq for transcription factors, signal mediators, DNase-seq and analysis of motifs. It's an impressive collection of new datasets which will provide an excellent resource. The authors use these datasets to show that the ubiquitously expressed and maternally Pou5f3 and Sox3 are required for normal embryonic development, including expression of genes which are zygotically expressed in a subset of the embryo. Using loss-of-function experiments, they show that this is likely due to the requirement for these factors to increase chromatin accessibility at their binding sites and allow additional factors to bind in response to the region-specific activity of signalling pathways, and propose a model in which ubiquitously-expressed transcription factors act as pioneer factors for signal mediators.

The key claims are well supported by the data. The provision of code will be very helpful for reproducibility, and the code appears to be well documented and to use appropriate methods. I tested a subset of the provided code and was able to run it with very minor modifications.

Major points:

- **Lines 134-139:** Figure 2b is provided to support a key claim in the paper, that transcription factor binding can recruit signal mediators, but not vice versa. However, the figure is quite hard to interpret. It would be helpful to have additional explanation of what the figure shows and how this supports the conclusions, either in the text or the figure legend.
We have rewritten the main text referring to Fig. 2b (now Fig. 3g). We also relabelled specific sections of the figure to clarify our statements: “With respect to the chromatin recruitment of TFs versus signal mediators, we note that, first, Smads and/or β -catenin were frequently detected at Sox3, Foxh1 or VegT binding sites (top 2,000 peaks shown in Fig. 3f) and, second, Smad- and/or β -catenin-bound pCRMs (top 2,000 peaks) were significantly enriched for SOX, POU-SOX, POU, FOXH and T motifs (red field #1 in Fig. 3g) suggesting that corresponding TFs affect the recruitment of these signal mediators. The reverse was much less the case as shown by the low significance of SMAD and bHSH motif enrichments at TF-bound pCRMs (with the exception of Smad2 interactor Foxh1²³) (blue field #2 in Fig. 3g)”.
- One reason why the figure is confusing is that in this context one would typically expect to see data on co-binding of transcription factors and signal mediators. However, the authors show only enrichment of motifs for signal mediators at regions enriched for TF binding (and vice versa) – not enrichment for signal mediator binding at regions enriched for TF binding. Enrichment of a motif does not necessarily imply enrichment of binding as measured by ChIP-seq signal. This key claim could be further supported by showing ChIP-seq signal enrichment as well as motif enrichment. It would also be helpful to comment on the extent to which these are concordant, as well as reasons why they might not be, such as the possibility of indirect binding or non-canonical motif usage.
We have generated new meta-plots (Fig. 3f) of signal mediator binding to the top 2,000 (according to DNA occupancy levels) Sox3⁺, VegT⁺ or FoxH1⁺ pCRMs.
- Although the developmental context of the proposed model, and the specific focus on transcription factors which respond to signalling pathways, are novel, the underlying model of pioneer factors allowing other

factors to bind has been previously widely discussed. The authors should add a discussion of previous work on pioneer factors to further place their findings in context.

We discuss a couple of more publications (Mullen *et al.*, 2011; Trompouki *et al.*, 2011; Lee *et al.*, 2013; Leichsenring *et al.*, 2013; Jacobs *et al.*, 2018) to put our work in the context of current knowledge.

Minor points:

- In general, the figure legends are quite brief and don't allow a full understanding of the figures without also referring back to the text.
We have extended most of the figure legends.
- The choice to plot differential gene expression data with a scale of 0-200% of control expression can make these figures harder to interpret for readers who are used to the more typical presentation of such data as log 2 fold changes. It can also be misleading as the upper end of the scale is truncated to only a two-fold change, while the lower end of the scale shows the full range of the data (although that may be appropriate in this case if the majority of differential gene expression is downregulation?).
We are primarily interested in analysing transcriptional downregulation. We felt that using percentages instead of fold changes generates more compact and readable graphs.
- I had to make a couple of minor changes to the code to get it to run on my machine. For example, Bioconductor packages, e.g. Diffbind, cannot be installed with `install.packages()` – you could replace this with a call to `biocLite()` or use the new BiocManager package. I had to replace “Arial” with “ArialMT” for the pdf fonts, but this may be OS-specific. Also, it appears that the file name “dnase_test.csv” should be “dnase.csv”.
We have corrected this.

REVIEWERS' COMMENTS:

Reviewer #1 (Remarks to the Author):

The paper is still hard work, but clearly important, and it has been clarified extensively, and more explanation has been included. I have only minor suggestions, but recommend that it be published

- Lines 33-35: "... these TFs have not been identified in a systematic fashion, and their modes of action remain largely unknown." But there have been very nice studies that identified such things as Shh competence factors in the limb versus the nervous system. Is the pax6 example really a good one? Because Pax6 is the classic textbook example of a protein affecting signal competence (Gilbert SF. *Developmental Biology*. 6th edition. Sunderland (MA): Sinauer Associates; 2000. Induction and Competence. Available from: <https://www.ncbi.nlm.nih.gov/books/NBK9993/>), we think it is the best example to mention here.

It is a good example, but the reference provided has no content on chromatin analysis, so I would argue still not the best comparable example. The work from the McMahon group showed the dependence on limb versus nervous system competence factors very clearly, and so I would argue that it is much more thorough and apposite than the pax 6 reference provided. But I would support them providing whatever reference they wish, though the reader I am sure would like to see one that provides a similar (or superior) level of analysis to that provided here.

- Lines 53-54: "... effect of TF co-expression on chromatin recruitment in vivo." Don't the authors mean the effect of co-expression of transcription factors on their recruitment to chromatin in vivo? Or do they really just mean generic chromatin recruitment.

Yes, we mean generic chromatin recruitment.

I still don't see what the chromatin is being recruited to. Though now I think it means the effect on recruitment of other TFs to chromatin? Please clarify the meaning.

- Discussion

Lines 252-253: "Our results allow us to propose a model (Fig. 6) of pioneer-initiated chromatin remodeling, or priming, that unlocks context-specific CRMs, some of which contain signal responsive

elements to enhance and regionalize transcription.“ Isn’t this an enhancement of the previous models, and specifically that for FoxH1.

We are not quite sure to what the reviewer is referring to.

Reference 18

And even recently published work recent work.

<https://doi.org/10.1016/j.celrep.2019.05.013>

Reviewer #3 (Remarks to the Author):

This is an impressive piece of work with a complex collection of genomic datasets and a comprehensive analysis. The authors have satisfied our previous requests, and the figures and text are now clearer and easier to understand. I believe the manuscript should be published - however, it still suffers somewhat from being extremely densely packed full of information. This could be improved by expanding the text to more explicitly state the reasoning behind the decisions made at each stage of the experiments / analysis, and to describe the results in clearer language, which would make it much easier for readers to follow.

Here are some specific examples where the text could be made clearer.

- Line 61-81: The authors define at least two sets of putative CRMs (by PolIII and by DNase accessibility) and it’s not clear if these are combined, or which sets or subsets are used for subsequent analyses.
- Line 104: what’s more important for the outgroup status of these TFs, that they are zygotically expressed, or that they are T-box TFs?
- Figure 3d and 3e are confusing. What is the reasoning for showing some datasets on both plots and some on only one? Why are there replicates for Smad1 in early gastrulation in 3e but not in 3d?

- Line 153-154: “developmental stage-characteristic enrichment of SOX, POU-SOX, and FOX motifs” doesn’t appear to be shown in Fig 4e. Specifically, only one developmental stage for MyoD-HA-expressing embryos is shown, SOX motifs appear similar across all stages in this figure, and POU-SOX motifs are not shown here.
- Line 192: “The LOF-mediated reduction of ZGA ranged from ~2% for BMP to ~25% for mPouV/Sox3”. It’s not clear what “2% reduction of ZGA” means – the average reduction in expression, or the % of genes that are affected? Based on the figure S11a, presumably this means that 2% of genes activated at ZGA are significantly affected. The numbers above and below the bars are not explained in the figure legend, however.
- Line 222: “next generation capture-C” – next generation compared to what?
- Line 242 / 478: “promoter-tied” / “level of promoter-tied chromatin conformations” is an unusual and therefore less clear way to refer to what is typically referred to as e.g. “interaction frequency” or “promoter contact frequency”.

REVIEWERS' COMMENTS:

Reviewer #1 (Remarks to the Author):

The paper is still hard work, but clearly important, and it has been clarified extensively, and more explanation has been included. I have only minor suggestions, but recommend that it be published

- Lines 33-35: "... these TFs have not been identified in a systematic fashion, and their modes of action remain largely unknown." But there have been very nice studies that identified such things as Shh competence factors in the limb versus the nervous system. Is the pax6 example really a good one? Because Pax6 is the classic textbook example of a protein affecting signal competence (Gilbert SF. Developmental Biology. 6th edition. Sunderland (MA): Sinauer Associates; 2000. Induction and Competence. Available from: <https://www.ncbi.nlm.nih.gov/books/NBK9993/>), we think it is the best example to mention here.

It is a good example, but the reference provided has no content on chromatin analysis, so I would argue still not the best comparable example. The work from the McMahon group showed the dependence on limb versus nervous system competence factors very clearly, and so I would argue that it is much more thorough and apposite than the pax 6 reference provided. But I would support them providing whatever reference they wish, though the reader I am sure would like to see one that provides a similar (or superior) level of analysis to that provided here.

We have added one more sentence referring to work of the McMahon group on signal competence as suggested by the reviewer (line 34): "For neural patterning, the correct transcriptional response to Sonic hedgehog (Shh) signaling requires the input of the pan-neural TF Sox2 (Peterson et al., 2012)." Reference: Peterson, K. A. et al. Neural-specific Sox2 input and differential Gli-binding affinity provide context and positional information in Shh-directed neural patterning. Genes Dev 26, 2802–2816 (2012).

- Lines 53-54: "... effect of TF co-expression on chromatin recruitment in vivo." Don't the authors mean the effect of co-expression of transcription factors on their recruitment to chromatin in vivo? Or do they really just mean generic chromatin recruitment. Yes, we mean generic chromatin recruitment.

I still don't see what the chromatin is being recruited to. Though now I think it means the effect on recruitment of other TFs to chromatin? Please clarify the meaning.

We have extended the sentence to clarify our findings (lines 70-71): "...to reveal the effect of co-expressed TFs on their recruitment and the recruitment of other transcriptional regulators to chromatin in vivo"

• Discussion

Lines 252-253: "Our results allow us to propose a model (Fig. 6) of pioneer-initiated chromatin remodeling, or priming, that unlocks context-specific CRMs, some of which contain signal responsive elements to enhance and regionalize transcription." Isn't this an enhancement of the previous models, and specifically that for FoxH1. We are not quite sure to what the reviewer is referring to.

Reference 18

And even recently published work recent work.

<https://doi.org/10.1016/j.celrep.2019.05.013>

We have attached these two references (ref. 19 and 36 in our manuscript) to "CRM priming for tissue-specific gene expression" mentioned in the next sentence (lines 285-286): "Our observations are in line with recent reports of CRM priming for tissue-specific gene expression^{19,35,36} and signal interpretation^{37,38}."

Reviewer #3 (Remarks to the Author):

This is an impressive piece of work with a complex collection of genomic datasets and a comprehensive analysis. The authors have satisfied our previous requests, and the figures and text are now clearer and easier to understand. I believe the manuscript should be published - however, it still suffers somewhat from being extremely densely packed full of information. This could be improved by expanding the text to more explicitly state the reasoning behind the decisions made at each stage of the experiments / analysis, and to describe the

results in clearer language, which would make it much easier for readers to follow.

Here are some specific examples where the text could be made clearer.

- Line 61-81: The authors define at least two sets of putative CRMs (by PolII and by DNase accessibility) and it's not clear if these are combined, or which sets or subsets are used for subsequent analyses.

All lists of putative CRMs including those found by RNAPII enrichment, DNase hypersensitivity or TF occupancy are provided as Supplementary Data (Excel spreadsheets). I have made some changes to make clear which set is used for subsequent analysis. Example (line 103) : “Enriched DNA motifs among accessible/RNAPII+/H3K4me+ pCRMs were then correlated with maternally inherited and translated sequence-specific factors identified by egg-staged mass spectrometry¹³ and pre-MBT ribosome footprinting¹⁴ to identify members of the TF families that may play a role in the ZGA.”

- Line 104: what's more important for the outgroup status of these TFs, that they are zygotically expressed, or that they are T-box TFs?

The fact that they are expressed later is the more important attribute to define them as an outgroup. We slightly changed the sentence to make this more apparent to the reader (lines 125-127): “As an outgroup control to these maternal proteins, we selected the binding profiles of the zygotic TFs Eomes (Eomesodermin)²⁰, zVegT²⁰, Tbx1 (Brachyury)²⁰ and Tbx6, all of which contain a T-box DNA binding domain and collectively regulate the neuro-mesodermal cell lineage during and beyond gastrulation²¹.”

- Figure 3d and 3e are confusing. What is the reasoning for showing some datasets on both plots and some on only one? Why are there replicates for Smad1 in early gastrulation in 3e but not in 3d?

For the sake of consistency between figure panels we merged the biological replicates for Smad1 in 3e. We did not want to overcrowd the panels. Fig. 3d shows the genomic binding relationship among maternal and zygotic TFs. Fig. 3e focuses on the relationship between RNAPII and TFs occupancy at pCRMs.

- Line 153-154: “developmental stage-characteristic enrichment of SOX, POU-SOX, and FOX motifs” doesn't appear to be shown in Fig 4e. Specifically, only one developmental stage for MyoD-HA-expressing embryos is shown, SOX motifs appear similar across all stages in this figure, and POU-SOX motifs are not shown here.

We have corrected this: Fig. 4e includes now the developmental stage-specific enrichment of POU-SOX. To make clear that SOX, POU-SOX and FOX motif enrichments are characteristic of the whole developmental period shown (blastula to late gastrula), we have also changed the subsentence (lines 181-183) to “...while MyoD-HA binding itself seemed to be influenced by endogenous TFs as judged by the developmental period (1,024-cell to late gastrula stage) characteristic enrichment of SOX, POU-SOX and FOX motifs at MyoD+ pCRMs (Fig. 4e)”

- Line 192: “The LOF-mediated reduction of ZGA ranged from ~2% for BMP to ~25% for mPouV/Sox3”. It's not clear what “2% reduction of ZGA” means – the average reduction in expression, or the % of genes that are affected? Based on the figure S11a, presumably this means that 2% of genes activated at ZGA are significantly affected. The numbers above and below the bars are not explained in the figure legend, however.

We rephrased this sentence (lines 223-224) to “The percentage of zygotic genes with reduced transcript levels ranged from ~2% to ~25% caused by BMP and mPouV/Sox3 LOFs, respectively”. Explanations were added to the legend of Supplementary Fig. 11a.

- Line 222: “next generation capture-C” – next generation compared to what?

The name “next-generation or NG capture-C” was introduced by the developers in Jim Hughes' lab (Davies et al., 2016). We did not want to change the name as we used and adapted their chromatin conformation capture method.

- Line 242 / 478: “promoter-tied” / “level of promoter-tied chromatin conformations” is an unusual and therefore less clear way to refer to what is typically referred to as e.g. “interaction frequency” or “promoter contact frequency”.

We have replaced “promoter-tied chromatin conformations” with “promoter contact frequencies”.

** See Nature Research's author and referees' website at www.nature.com/authors for information about policies, services and author benefits

This email has been sent through the Springer Nature Tracking System NY-610A-NPG&MTS